# Prioritizing Faithfulness: Efficient Zero-Shot Novel View Synthesis with Adaptive Latent Modulation

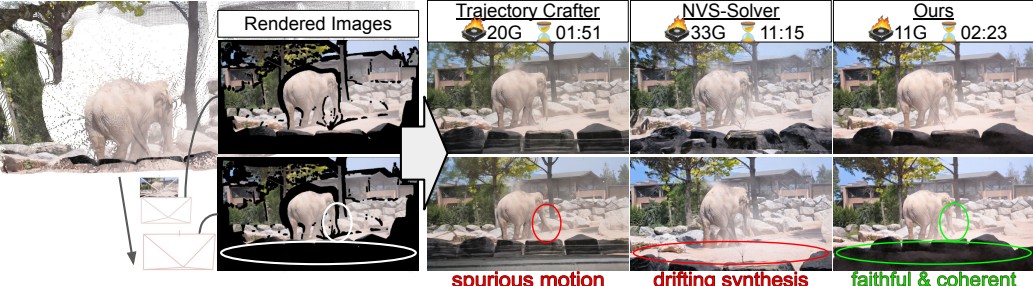

Figure 1: Existing render-and-inpaint NVS methods often sacrifice faithfulness for fidelity, leading to (i) spurious motion, (ii) drifting synthesis: inpainted regions are incoherent with camera motion, and (iii) appearance shifts. In contrast, our **training-free** approach prioritizes faithfulness, yielding globally structure-coherent results while maintaining sufficient fidelity, requiring only 11 GB of memory. A total of 25 frames are generated, with the figure showing the 8th and 16th frames.

## Abstract

The challenge of camera-controlled novel view synthesis (NVS) lies in balancing high visual fidelity with strict faithfulness to the source scene. We argue that current dominant approaches, which rely on finetuning large-scale diffusion models, often over-emphasize fidelity while struggling with faithfulness due to their generative nature. To address this, we propose a zero-shot NVS pipeline that prioritizes faithfulness and efficiency. Our method introduces two key contributions applied during inference: (1) Test-time Latent Homography Deformation, an on-the-fly homography optimization to deform latents for global motion consistency, and (2) Spatially Adaptive RePaint (SA-RePaint), an extension to RePaint that achieves both structural consistency and texture fidelity by introducing a mathematically-grounded, region-wise balancing of these two objectives. Our evaluations demonstrate substantial improvements in faithfulness and camera accuracy with competitive perceptual scores, highlighting a successful integration of faithfulness, quality, and efficiency. This work offers a promising direction for NVS that rebalances the focus towards greater authenticity.

## 1 Introduction

Camera-controlled novel view synthesis (NVS) aims to generate a video along a user-specified camera trajectory from a source image or video. Beyond creative use, NVS is poised to enable practical applications like e-commerce, digital archiving, and virtual architectural walkthrough. The primary challenge is to simultaneously satisfy three key objectives: high visual fidelity, strict faithfulness to the source, and geometric consistency throughout the generated video. While fidelity has often been the central focus, for such practical applications, artifacts like texture changes or color shifts are unacceptable, making faithfulness a priority on par with, or even greater than, visual fidelity.

Many recent methods based on finetuning large-scale video diffusion models (He et al., 2024; Yu et al., 2024), have achieved impressive visual fidelity, but their reliance on strong generative priors

makes consistently maximizing faithfulness a non-trivial challenge, leading to artifacts like *spurious motion* on the primary subject, as seen in Fig. 1. Furthermore, these pipelines are computationally expensive: large datasets and substantial resources for training, with inference also remaining expensive, limiting their broader accessibility and customizability.

In this regard, training-free methods offer a compelling alternative, as they allow for direct faithfulness control without costly retraining. However, to our knowledge, no existing zero-shot method has simultaneously pursued both high faithfulness and lightweight inference. Some, like NVS-Solver (You et al., 2025), achieve a degree of faithfulness through test-time optimization, but still suffer from visual artifacts like *drifting synthesis* in the generated region (failing to follow the camera motion) and prohibitive computational costs (33 GB VRAM, 11:15 inference time, Fig. 1). Others prioritize efficiency but neglect faithfulness as a primary goal (Hou & Wei, 2024). Thus, a solution addressing both challenges remains an open question.

To fill this gap, we propose a novel training-free NVS pipeline that achieves both high faithfulness and efficiency. Our approach is built on a render-and-inpaint scheme: we render images from a 3D point cloud derived from the source image (Yang et al., 2024a); their disoccluded regions are then inpainted by a video diffusion model (Blattmann et al., 2023). For this step, we leverage RePaint (Lugmayr et al., 2022), a technique that repurposes a diffusion model for zero-shot inpainting. This choice proves remarkably effective for faithfulness, yet its naive application exhibits two critical limitations: the aforementioned phenomenon of *drifting synthesis*, and the trade-off between structural consistency and texture fidelity. We tackle these challenges with two key contributions:

1. **Test-time Latent Homography Deformation**, a lightweight optimization that resolves drifting synthesis in inpainted regions. It deforms the latent tensor on the fly to align with the rendered images, ensuring the entire scene moves in coherence with the camera motion.

2. **Spatially Adaptive RePaint (SA-RePaint)**, our solution to the structure-texture trade-off. This issue stems from RePaint's fixed strategy for balancing reliance on the rendered images versus the model's generative freedom. We make this balance spatially auto-adaptive, allowing it to generate globally coherent structures while producing rich new textures.

Our evaluation demonstrates substantial improvements in faithfulness and camera pose accuracy while remaining competitive in perceptual quality, all under 11 GB of VRAM. This outcome shows that significant gains in faithfulness are achievable without disproportionate trade-offs in visual quality or efficiency. By rebalancing these competing objectives towards faithfulness, our work contributes to a more practical and reliable form of NVS, offering a promising direction for applications where authenticity is paramount.

## 2 RELATED WORKS

**Novel View Video Synthesis** Novel view synthesis (NVS) approaches can be broadly categorized into reconstruction-based and generation-based methods. Reconstruction-based methods like NeRF (Mildenhall et al., 2021) and 3DGS (Kerbl et al., 2023) build implicit or explicit 3D scene representations. While faithful, they struggle with novel camera poses. In contrast, recent generation-based methods adapt pretrained video diffusion models (Blattmann et al., 2023; Yang et al., 2024b; HaCohen et al., 2024; Kong et al., 2024; Wan et al., 2025) for NVS, offering superior visual fidelity and generalizability by conditioning on various view-related signals.

**View Conditioning Types** These conditioning signals include camera parameters (He et al., 2024; Zhang et al., 2024a; Zhou et al., 2025; Bai et al., 2025), optical flow (Jin et al., 2025; Burgert et al., 2025), or, most relevant to our work, rendered point clouds derived from depth estimation (Yu et al., 2024; You et al., 2025; Xiao et al., 2025; YU et al., 2025; Ren et al., 2025; Seo et al., 2024; Chen et al., 2025a). Our method adopts the point cloud rendering strategy for its strong geometric prior, maximizing faithfulness. This contrasts with approaches that only use rendered views for positional encoding, citing their unreliability (Seo et al., 2024). We rather argue for a strict separation of concerns, entrusting geometry to the depth estimator and inpainting to the diffusion model, thereby prioritizing faithfulness and scales with improving depth estimators.

**Training-Free Methods** While most generation-based NVS methods rely on finetuning via LoRA (Hu et al., 2022) or ControlNet (Zhang et al., 2023), a few training-free alternatives exist.

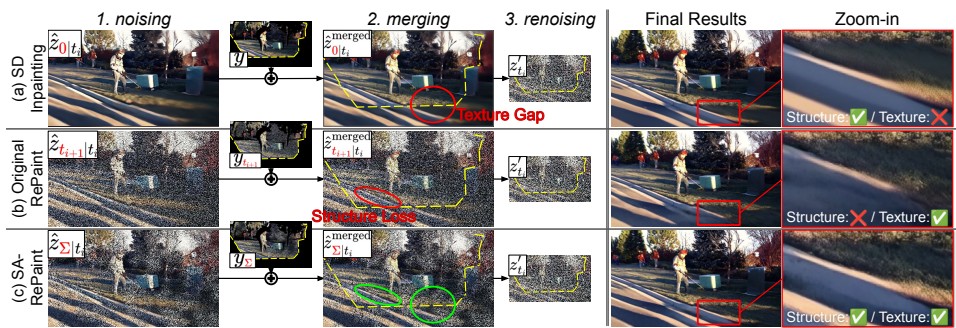

Figure 2: The structure-texture trade-off in RePaint variants. (a, b) Existing RePaint implementations compromise either structural integrity or textural fidelity. (c) Our **SA-RePaint** overcomes this challenge by adopting a spatially variable noise map $\Sigma$. The intermediate steps 1,2,3 are conceptual illustrations; the actual process occurs entirely in the latent space.

The most comparable work, NVS-Solver (You et al., 2025), also uses rendered point clouds but optimizes the latent tensor via gradient descent, incurring significant computational overhead. Other methods either sacrifice faithfulness for efficiency (Hou & Wei, 2024) or require additional 3D reconstruction models like MonST3R (Zhang et al., 2024b), increasing complexity (Park et al., 2025; Zhou et al., 2024). Another line of work employs iterative per-frame RGBD inpainting and 3D lifting (Engstler et al., 2025). Although lightweight, such auto-regressive approaches often suffer from error accumulation and severe temporal drift, unlike batch-processing video diffusion models. Our approach, in contrast, achieves high faithfulness and efficiency without costly backpropagation or auxiliary models, through direct intervention in the inference process of a video diffusion model.

## 3 PRELIMINARIES

### 3.1 STABLE VIDEO DIFFUSION

Stable Video Diffusion (SVD) (Blattmann et al., 2023) is an image-to-video diffusion model built upon the EDM framework (Karras et al., 2022). It operates in a latent space where a video is represented as $z_0$. During training, the forward process corrupts this clean latent by:

$$z_t = \texttt{add\_noise}(z_0, 0 \to t) := z_0 + t\epsilon, \tag{1}$$

where $\epsilon \sim \mathcal{N}(0, I)$ is Gaussian noise and $t \in \mathbb{R}_{\geq 0}$ is a noise level. A U-Net then derives a clean latent $\hat{z}_{0|t} \approx z_0$ from $z_t$, $t$, and the first frame of $z_0$. At inference, one initializes $z_T \sim \mathcal{N}(0, T^2 I)$ and for a decreasing schedule $T = t_0 > \cdots > t_N = 0$ applies the Euler update rule iteratively:

$$z_{t_{i+1}} = \texttt{Euler}(z_{t_i}; \hat{z}_{0|t_i}, t_i \to t_{i+1}) := \hat{z}_{0|t_i} + t_{i+1} \cdot D(z_{t_i}) \tag{2}$$

where $D(z_{t_i}) = (z_{t_i} - \hat{z}_{0|t_i})/t_i$. The final latent $z_{t_N} = z_0$ is then decoded into the output video.

### 3.2 REPAINT VARIANTS AND THEIR IMPLICATIONS

RePaint (Lugmayr et al., 2022) enables diffusion-based zero-shot inpainting by repeatedly pasting the known region of a conditioning image $y$ (defined by a mask $m^{\text{valid}}$) onto intermediate denoised predictions during the sampling process.

In practice, RePaint-style inpainting has been implemented in two prominent ways: **Original RePaint** and **Stable Diffusion (SD) Inpainting**. Let $z_{t_i}$ be the current noisy latent and $\hat{z}_{0|t_i}$ be the prediction of the clean latent. Both variants follow a process of *merging* $y$ and $\hat{z}_{0|t_i}$ based on the mask and *renoising* the result back to the noise level $t_i$, yielding a modified latent $z'_{t_i}$ (Fig. 2 left). This new latent then serves as the input for the standard Euler update step ($z'_{t_i} \to z_{t_{i+1}}$). Their key difference lies in *at which noise level* this merging occurs, as summarized below:

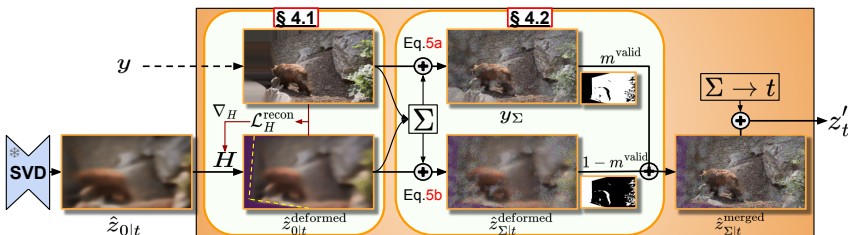

Figure 3a: Overview of our rendering and diffusion inpainting pipeline. Step ① modulates the latent $z_{t_i}$ to $z'_{t_i}$ so that it aligns with $y$ (our core contribution), while ② performs the standard denoising process $z'_{t_i} \mapsto z_{t_{i+1}}$.

Figure 3b: Detailed view of latent modulation. We globally align the latents $\hat{z}_{0|t}$ with the rendered image latents $y$ by test-time optimizing a homography $H$ (Section 4.1), and derive a non-uniform noise map $\Sigma$ (see Fig. 2) for smooth blending (Section 4.2). The merged result $\hat{z}^{\text{merged}}_{\Sigma|t}$ is then further corrupted to reach a uniform noise level $t$. All tensors reside in the latent space; the attached images are for illustration purposes only.

| | **SD Inpainting** (merging at $t = 0$) | **Original RePaint** (merging at $t = t_{i+1}$) |
|---|---|---|
| *Noising* | None | $\begin{cases} y_{t_{i+1}} = \texttt{add\_noise}(y, 0 \to t_{i+1}) \\ z_{t_{i+1}} = \texttt{Euler}(z_{t_i}; \hat{z}_{0|t_i}, t_i \to t_{i+1}) \end{cases}$ |
| *Merging* | $z^{\text{merged}}_{0|t_i} = m^{\text{valid}} y + (1 - m^{\text{valid}}) \hat{z}_{0|t_i}$ | $z^{\text{merged}}_{t_{i+1}} = m^{\text{valid}} y_{t_{i+1}} + (1 - m^{\text{valid}}) z_{t_{i+1}}$ |
| *Renoising* | $z'_{t_i} = \texttt{add\_noise}(z^{\text{merged}}_{0|t_i}, 0 \to t_i)$ | $z'_{t_i} = \texttt{add\_noise}(z^{\text{merged}}_{t_{i+1}}, t_{i+1} \to t_i)$ |

The choice of merging noise level, despite both variants renoising to the same level $t_i$, leads to distinct outcomes (Fig. 2 right). Merging at the lowest noise level $t = 0$ (SD Inpainting) maintains structural alignment but results in overly smooth, textureless outputs. Conversely, merging at a high noise level $t_{i+1}$ (Original RePaint) enhances texture fidelity but compromises structural consistency.

This trade-off motivates us to develop a method that adaptively selects the optimal noise level for merging per region. Ideally, such a method would use lower noise for textured regions to preserve structure, while applying higher noise to smoother regions for better inpainting fidelity (Fig. 2c). The mathematical formulation of this intuition is discussed in Section 4.2.

## 4 METHODOLOGY

Given a source image $I_0$ and a specified camera path $\{C_f\}_{f=0}^{F-1}$, our goal is to synthesize a novel view video $\{I'_f\}_{f=0}^{F-1}$ ($I'_0 = I_0$) following this camera path. Although our method easily extends to video inputs (see Appendix P), we focus on the single-image setting: this creates a "bullet-time" effect where the scene must remain static, providing a stringent test for faithfulness as even minor motion artifacts are highly perceptible. Our pipeline (Fig. 3a) consists of two main stages: rendering via 3D projection and diffusion-based inpainting. Our key contributions are introduced within the latter inpainting stage to resolve the inherent challenges of this framework.

**Rendering via 3D Projection** First, we lift the input image $I_0$ to a 3D point cloud using Depth Anything V2 (Yang et al., 2024a), and render it along the camera path to produce a sequence of views $\{I_f\}_{f=0}^{F-1}$. These views contain empty regions corresponding to disocclusions. To prevent black pixels in these regions from contaminating valid regions during VAE encoding, we pre-fill them using a classical completion method (Bertalmio et al., 2001). These pre-filled images are VAE-encoded to form a latent tensor $y \in \mathbb{R}^{F \times C \times H \times W}$, with corresponding valid-region masks $m^{\text{valid}} \in [0, 1]^{F \times 1 \times H \times W}$ derived by resizing their pixel-space counterparts.

**Diffusion Inpainting**   We inpaint the disoccluded regions in $y$ using a RePaint-style (Lugmayr et al., 2022) iterative process with Stable Video Diffusion (SVD) (Blattmann et al., 2023). The process begins at an intermediate step $i_0 \in [0, N-1]$, where we initialize the latent by noising $y$:

$$z_{t_{i_0}} = y + t_{i_0}\,\epsilon, \quad \epsilon \sim \mathcal{N}(0, I). \tag{3}$$

SVD then iteratively denoises this latent, with RePaint intervening at each step as in Sec. 3.2.

However, a naive application of RePaint suffers from two key issues: (i) *drifting synthesis*, where the inpainted region fails to follow the camera motion (as seen in Fig. 1, NVS-Solver), and (ii) the *structure-texture trade-off* discussed in Sec. 3.2 (Fig. 2). To resolve these issues, we introduce two lightweight strategies that respectively address each problem by modulating the clean prediction $\hat{z}_{0|t_i}$ before the merging step. For simplicity, we hereafter drop the denoise step subscript $i$.

## 4.1   TEST-TIME LATENT HOMOGRAPHY DEFORMATION

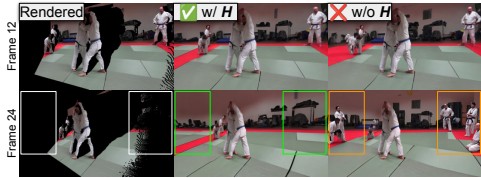

Our baseline approach, a naive application of Re-Paint, often suffers from *drifting synthesis*: the inpainted regions fail to follow the camera motion, appearing static and disconnected from the rendered area (an issue also seen in methods like NVS-Solver, Fig. 1). We hypothesize that this stems from an inherent static bias in image-to-video diffusion models, which prioritize texture stability over consistent motion (Tian et al., 2025; Choi et al., 2025). This issue is particularly pronounced in SVD, which lacks text prompts that could otherwise guide motion.

Figure 4: Comparison before and after introducing homography deformation. The orange boxes indicate that the background of "w/o $H$" doesn't follow the camera rotation.

To address this *drifting synthesis*, we introduce a test-time latent homography deformation. Our goal is to find a set of homographies $\{H_f\}_{f=0}^{F-1}$, where each $H_f \in \mathbb{R}^{3\times3}$ warps the clean prediction $\hat{z}_{0|t}$'s $f$-th frame to align with the rendered image's latent $y[f]$. We formulate this as an optimization problem solved at each denoise step, minimizing two losses: a reconstruction loss $\mathcal{L}_H^{\text{recon}}$ enforcing the alignment, and a temporal smoothing loss $\mathcal{L}_H^{\text{smooth}}$ encouraging constant velocity:

$$\mathcal{L}_H^{\text{recon}} = \sum_{f=0}^{F-1} \left\| \left( y[f] - H_f\,\hat{z}_{0|t}[f] \right) \cdot m^{\text{valid}}[f] \right\|_1, \quad \mathcal{L}_H^{\text{smooth}} = \sum_{f=1}^{F-2} \left\| H_{f+1} - 2H_f + H_{f-1} \right\|_1. \tag{4}$$

The total loss is $\mathcal{L}_H = \mathcal{L}_H^{\text{recon}} + \lambda_H \mathcal{L}_H^{\text{smooth}}$. At each denoise step, $H_f$ is initialized to identity and optimized by Adam (Kingma & Ba, 2014) with a learning rate of 0.01 for 100 steps. After $\{H_f\}_{f=0}^{F-1}$ has converged, we replace $\hat{z}_{0|t}$ with $\hat{z}_{0|t}^{\text{deformed}} = \texttt{stack}(\{H_f\,\hat{z}_{0|t}[f]\}_{f=0}^{F-1})$ for subsequent process.

Since homographies perform a global transformation, they effectively propagate the motion from the rendered regions into the inpainting areas, ensuring spatio-temporal consistency across frames (Fig. 4). Visualizations of this warping process are in Appendix E, where the homography is shown to gradually converge to the identity over denoising steps. However, homographies cannot model complex, depth-induced parallax. We therefore disable this deformation during later denoise steps, allowing the model to establish finer geometric details unconstrained by the global warp.

## 4.2   SPATIALLY ADAPTIVE REPAINT (SA-REPAINT)

As discussed in Sec. 3.2, RePaint's fixed merge noise level creates a structure-texture trade-off. We propose to resolve this with a spatially adaptive merge noise level, computed as a per-pixel map $\Sigma \in [0, t]^{F \times C \times H \times W}$. To derive a criterion for computing $\Sigma$, we first analyze the source of this trade-off, which is guided by the conceptual visualizations in Fig. 2 left.

**Merging at low noise** ($t = 0$, SD Inpainting) Merging at the clean level (Fig. 2a) creates a stark "texture gap" between the sharp, pasted region $y$ and the smoother prediction $\hat{z}_{0|t}$. We hypothesize that this visible gap is also present in the actual merging in the latent space. Consequently, in the subsequent denoise step, the model may misinterpret this discrepancy as a genuine scene feature, thereby inhibiting texture generation in the inpainted region and impairing fidelity.

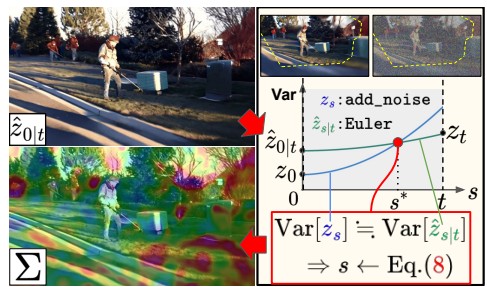
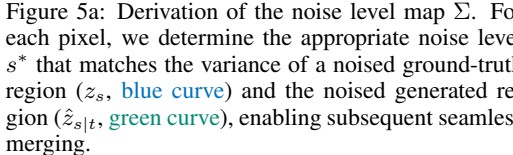

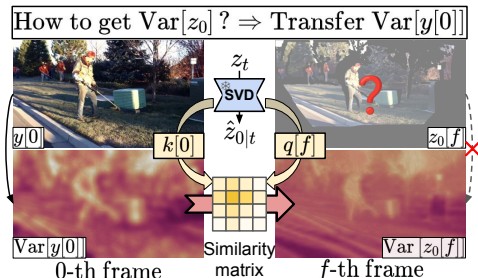

Figure 5a: Derivation of the noise level map $\Sigma$. For each pixel, we determine the appropriate noise level $s^*$ that matches the variance of a noised ground-truth region ($z_s$, blue curve) and the noised generated region ($\hat{z}_{s|t}$, green curve), enabling subsequent seamless merging.

Figure 5b: Estimating $\text{Var}[z_0]$ via Variance Transfer. Since the ground truth $z_0$ is unavailable, its variance $\text{Var}[z_0]$, required in (a), must be estimated. Our approach transfers the variance map of the first frame $y[0]$, leveraging the attention $qk$-similarity between the first and subsequent frames ($f = 1, \ldots, F - 1$).

**Merging at high noise** ($t = t_{i+1}$, Original RePaint) Conversely, at high noise levels, the texture gap seems to vanish (Fig. 2b), yet the resulting loss of structural consistency suggests an underlying statistical imbalance. We posit that the noising applied to $y$ is more destructive to its overall geometric structures than the one applied to $\hat{z}_{0|t}$[1]. This could obscure geometric cues in the known region, hindering their propagation to inpainting regions and compromising coherence.

This analysis suggests that our goal should be to define a noise level map $\Sigma$ such that $y$ and $\hat{z}_{0|t}$ noised by $\Sigma$ are statistically coherent and merge seamlessly. We expect such a merged tensor to be perceived as in-domain, allowing the model to exert its original generation capability and resolve the structure-texture trade-off. Let's first extend the noising formulation from Sec. 3.2 to be pixelwise:

$$y_\Sigma := \texttt{add\_noise}(y, 0 \to \Sigma) = y + \Sigma \cdot \epsilon, \quad \epsilon \sim \mathcal{N}(0, I) \tag{5a}$$

$$\hat{z}_{\Sigma|t} := \texttt{Euler}(z_t; \hat{z}_{0|t}, t \to \Sigma) = (1 - \Sigma/t) \cdot \hat{z}_{0|t} + (\Sigma/t) \cdot z_t \tag{5b}$$

To make $y_\Sigma$ and $\hat{z}_{\Sigma|t}$ statistically coherent, matching their full statistical distributions is intractable. We therefore propose to match their *local pixel-value variance* as a tractable proxy for this purpose.

For a tensor $z \in \mathbb{R}^{F \times C \times H \times W}$, let $p = (p_f, p_y, p_x)$ be a spatio-temporal coordinate, and $\mathcal{W}_p$ be a local spatio-temporal window (e.g., of size $3 \times 3 \times 3$) centered at $p$. We define the *local pixel-value variance* $\text{Var}[z] \in \mathbb{R}^{F \times C \times H \times W}$, computed independently for each channel, as follows:

$$\text{Var}[z](p_f, :, p_y, p_x) := \frac{1}{|\mathcal{W}_p|} \sum_{q \in \mathcal{W}_p} z(q_f, :, q_y, q_x)^2 - \left( \frac{1}{|\mathcal{W}_p|} \sum_{q \in \mathcal{W}_p} z(q_f, :, q_y, q_x) \right)^2. \tag{6}$$

Our task is thus to find a map $\Sigma$ that satisfies $\text{Var}[y_\Sigma](p) \approx \text{Var}[\hat{z}_{\Sigma|t}](p)$ for all $p$.

### 4.2.1 Deriving the Per-Pixel Noise Level

Let $s = \Sigma(p) \in [0, t]$ be the target noise level at pixel $p$. We focus on its local window $\mathcal{W}_p$ and consider how to deduce $s$. Accordingly, we rewrite $y_\Sigma$ and $\hat{z}_{\Sigma|t}$ as $y_s$ and $\hat{z}_{s|t}$, respectively. For simplicity, we treat $\text{Var}[\hat{z}_{s|t}]$ and $\text{Var}[y_s]$ as scalars, implicitly referring to their values at pixel $p$.

If the conditioning image $y$ is available on $\mathcal{W}_p$, the objective is straightforward: find the noise level $s$ that minimizes the variance difference between the noised conditioning image $y_s$ and the noised prediction $\hat{z}_{s|t}$: $s^* := \arg\min_{s \in [0,t]} \left\| \text{Var}[\hat{z}_{s|t}] - \text{Var}[y_s] \right\|_1$.

However, this formulation is confined to known regions, as $y$ offers no guidance in areas requiring inpainting. To create a unified objective, we need a reference signal that is valid across all pixels. The most logical candidate is the variance of the final, ideal output $z_0$. We therefore generalize the objective by replacing $y$ with the (hypothetical) ground-truth $z_0$.

$$s^* := \arg\min_{s \in [0,t]} \left\| \text{Var}[\hat{z}_{s|t}] - \text{Var}[z_s] \right\|_1, \tag{7}$$

---

[1]Due to the asymmetric noising mechanisms (`add_noise` for $y$ versus `Euler` for $\hat{z}_{0|t}$), the initial textural superiority of $y$ can be reversed at high noise levels, leaving it more degraded than the prediction $\hat{z}_{0|t}$.

where $z_s := z_0 + s\epsilon$. This objective is well-defined for all pixels and consistently reduces to our initial objective in known regions where we can assume $z_0 \approx y$. Fortunately, this generalized problem is a quadratic minimization, thus it admits a closed-form solution.

**Theorem 1.** *The noise level $s^* \in [0, t]$ that minimizes the objective in Eq. 7 is an element of the set:*

$$s^* \in \{\eta_+, \eta_-, 0, t\}, \quad \text{where } \eta_\pm := \text{clip}\left(\left(-B \pm \sqrt{\max(0, B^2 - AC)}\right)/A; 0, t\right). \quad (8)$$

*The coefficients $A$, $B$, and $C$ are given below, where $D(z_t) := (z_t - \hat{z}_{0|t})/t$. $\text{Var}[\cdot]$ and $\text{Cov}[\cdot, \cdot]$ denote the pixel-value variance and covariance on the local window:*

$$A = \text{Var}[D(z_t)] - 1, \quad B = \text{Cov}[\hat{z}_{0|t}, D(z_t)], \quad C = \text{Var}[\hat{z}_{0|t}] - \text{Var}[z_0]. \quad (9)$$

Theorem 1 reveals that the optimal noise level $s^*$ is one of at most four candidate values. We find the true optimum by evaluating the objective in Eq. 7 for each candidate. This provides a direct path to computing the entire noise map $\Sigma$ pixel by pixel.

However, a critical prerequisite remains. The computation of the coefficient $C$ requires $\text{Var}[z_0]$ as an input. While we can use $\text{Var}[y]$ in known regions, the core challenge is to estimate $\text{Var}[z_0]$ for unknown regions before $z_0$ is generated. We tackle this circular problem using cross-frame attention.

### 4.2.2 ESTIMATING $\text{Var}[z_0]$ WITH CROSS-FRAME ATTENTION

To estimate the unknown variance $\text{Var}[z_0]$, we propagate texture information from a known source: the input first frame $y[0]$. We leverage the cross-frame attention mechanism within the SVD U-Net, positing that attention similarity reflects textural similarity. Let $q_t$ and $k_t$ be the attention queries and keys from a U-Net upsample block. We approximate $\text{Var}[z_0]$ as follows:

$$\text{Var}[z_0] \approx \lambda_{\text{var}} \cdot A_t \text{Var}[y[0]], \quad \text{where } A_t := \text{softmax}\left(q_t(k_t[0])^\top/\sqrt{d}\right). \quad (10)$$

Here, $A_t$ is the attention similarity map between all the frames' queries $q_t$ and the first frame's key $k_t[0]$, scaled by the channel dimension $d$. The hyperparameter $\lambda_{\text{var}}$ scales the transferred variance, and we set it to 1.5 to enhance fidelity (see Appendix J for ablation).

As visualized in Fig. 5b, this method robustly estimates a smooth variance map for $\text{Var}[z_0]$ even in unobserved regions. The resulting $\Sigma$ map accurately identifies areas requiring more texture synthesis (e.g., the foliage on the right) by assigning higher noise levels, while keeping noise low in simpler or rendered regions (e.g., the road). The full algorithm is provided in Appendix D.

### 4.2.3 ENSURING RELIABLE VARIANCE TRANSFER WITH KEY WEIGHTING

Our variance estimation (Eq. 10) presumes meaningful textural correspondence between the source $y[0]$ and the generated content. This assumption breaks down when novel objects are generated in unknown regions, invalidating the variance transfer since there is no source texture to draw from.

To enforce reliable correspondence, we employ time-varying key weighting, inspired by Sun et al. (2025); Jia et al. (2024). We modulate the key $k_{t_i}$ with a weight mask $w_i$ that suppresses its influence in invalid regions, especially during early denoising steps:

$$k_{t_i}^{w_i} := \phi(w_i) \cdot k_{t_i}, \quad \text{where } w_i := i/N + m^{\text{valid}} \cdot (1 - i/N). \quad (11)$$

Here, $\phi$ is a function to align the shape of $w_i$ with the key token $k_{t_i}$. This guides the model to rely on valid regions in $y$, ensuring a reliable basis for variance transfer by discouraging object hallucination. As denoising progresses, we gradually relax this weighting ($w_i \rightarrow 1$), allowing the model to shift its focus from source-reliance to the generation of coherent internal details within the inpainted regions, which results in improved fidelity.

## 5 EXPERIMENTS

We evaluate our proposed method on several standard benchmarks for novel view synthesis. Detailed implementation settings, including hyperparameters, are provided in Appendix G. Notably, our pipeline also incorporates Smoothed Energy Guidance (SEG) (Hong, 2024), which enhances perceptual quality with negligible computational overhead.

## 5.1 Benchmark Settings

**Metrics.** We adopt a dual-faithfulness evaluation strategy to rigorously assess distinct aspects of the synthesis. First, we introduce **Input-Faithfulness**, which measures the pixel-wise alignment between the output and the projected input rendering. This effectively serves as a metric for *controllability*: any deviation from valid rendered images indicates failure issues as reported in Fig. 1. Second, we report **GT-Faithfulness**, a conventional metric to compare the output against ground-truth target views. For both metrics, we compute PSNR, SSIM, and LPIPS *exclusively on the valid rendering regions*. This masking is essential to decouple the evaluation of faithfulness (preserving visible content) from fidelity (hallucinating occluded content), ensuring that plausible inpainting is not penalized for differing from the ground truth. The quality of these hallucinated regions is assessed by **Fidelity** measures: FID, KID, FVD, and VBench scores (Huang et al., 2024). Additionally, we compute **Camera Pose Errors** (ATE, RRE, RTE) between GLOMAP-estimated poses (Pan et al., 2024) and the ground truth. Lastly, we assess **Static Geometry Compliance** using Thresholded Symmetric Epipolar Distance (TSED) (Yu et al., 2023) with thresholds of 0.25 and 0.5, and MEt3R (Asim et al., 2024) with images resized so that the longer side is 256 while the aspect ratio is retained.

**Tasks and Datasets.** We evaluate our method on two different settings. **(i) Scripted Camera Motion**: we use DAVIS (Perazzi et al., 2016) and Tanks and Temples (T&T) (Knapitsch et al., 2017). To generate our test sequences, we sample one frame at a 25-frame interval from each source video, apply Depth Anything V2 (Yang et al., 2024a), and synthesize a 25-frame clip using a predefined camera trajectory chosen uniformly from a set of 10 motions comprising rotation and zoom. In this setting, we don't have access to ground truth target images, so we skip the gt-faithfulness evaluation. **(ii) Real Camera Motion**: We use the Mannequin Challenge (MC) validation set (Li et al., 2019) and DL3DV-Evaluation (Ling et al., 2024). We cut out 25-frame clips from the video, apply Depth Anything 3 (Lin et al., 2025) to infer corresponding per-frame depth and camera pose, and synthesize 25-frame clips based on these poses. We have found that SED scores are too low in other baselines to compare meaningfully, so we remove them from the evaluation.

Note that DAVIS and MC contain movable objects, posing a challenge to maintain faithfulness to rendered images against the video diffusion model's strong motion priors. Filtering out MC videos below 1080p and random half of DL3DV-Evaluation, the respective dataset contains a total of 288 (DAVIS), 314 (T&T), 268 (MC), and 360 (DL3DV) sequences. We report aggregated scores across all datasets; detailed per-dataset results are in Appendix H.

**Baselines.** We compare our method against four rendering or trajectory-based methods: Trajectory Attention (Xiao et al., 2025), Trajectory Crafter (YU et al., 2025), Diffusion As Shader (Gu et al., 2025), and NVS-Solver (You et al., 2025). The former three are fine-tuned models, whereas NVS-Solver, like our method, is training-free. Other training-free works (Hou & Wei, 2024; Zhou et al., 2024) are excluded due to the unavailability of public implementations. We also evaluated Invisible Stitch (Engstler et al., 2025) and Stable Virtual Camera (Zhou et al., 2025); however, due to fundamental methodological differences (e.g., iterative inpainting or lack of explicit 3D guidance) that hinder a strictly fair comparison, their results are reported separately in Appendix M.

## 5.2 Comparison Results: Scripted Camera Motion

As illustrated in Fig. 6, our method demonstrates superior faithfulness to the rendered images and globally consistent yet high-fidelity generation, avoiding misalignment artifacts common in prior work, such as spurious foreground motion, static/drifting background synthesis, or color shifts.

The quantitative results in Tables 1 reveal a dramatic improvement in faithfulness (PSNR, SSIM, LPIPS) and geometric consistency (camera pose, TSED), substantially outperforming all baselines. This high faithfulness is achieved in tandem with competitive perceptual quality, as reflected in our FID/KID[3] and VBench scores. While some of these scores do not rank highest, this is partly because certain baselines fail to generate motion aligned with the camera trajectory, resulting in near-static videos. This failure mode can artificially inflate inter-frame consistency metrics. In contrast, our ap-

---

[2]See Appendix I for the evaluation details.

[3]Trajectory Crafter's unexpectedly high FID/KID likely stems from a domain shift, as it tends to synthesize novel backgrounds not present in the ground truth. This hypothesis is plausible given its strong VBench scores.

Figure 6: Qualitative comparison with other methods. Our method achieves the highest faithful geometric/appearance alignment to the rendered images (green boxes). In contrast, other methods exhibit severe inconsistency (red boxes) or subtle yet noticeable appearance shifts (orange boxes). Gray labels indicate finetuning-based methods. Other results are in Appendix K.

Table 1: Quantitative results of Scripted Camera Motion. Top: Comparison on standard metrics. Bottom: VBench evaluation. Gray rows denote training-based methods. Here, KID' denotes $KID \times 10^3$, $T_{.25}$ ($T_{.50}$) represents TSED with a threshold of 0.25 (0.50), and M3R stands for MEt3R.

| Method | Input-Faithfulness | | | Fidelity ↓ | | | Pose ↓ | | | Geometry | | | Efficiency[2] ↓ | |
|---|---|---|---|---|---|---|---|---|---|---|---|---|---|---|
| | PSNR↑ | SSIM↑ | LPIPS↓ | FID | KID' | FVD | ATE | RRE | RTE | $T_{.25}$↑ | $T_{.50}$↑ | M3R↓ | GB | Time |
| Traj. Attention | 23.01 | 0.731 | 0.175 | 18.36 | 1.737 | 640.1 | 0.266 | 0.063 | 0.100 | 0.378 | 0.916 | 0.031 | 12.7 | **1:15** |
| Traj. Crafter | 24.11 | 0.804 | 0.114 | 18.65 | 1.586 | 699.9 | 0.132 | 0.048 | 0.074 | 0.472 | 0.965 | **0.030** | 19.4 | 1:51 |
| Diff. As Shader | 14.92 | 0.453 | 0.396 | 19.43 | 1.671 | **497.5** | 1.543 | 0.491 | 1.285 | 0.040 | 0.130 | 0.037 | 30.8 | 7:30 |
| NVS-Solver | 21.91 | 0.713 | 0.188 | 16.57 | 1.039 | 640.0 | 0.593 | 0.161 | 0.311 | 0.318 | 0.764 | 0.037 | 32.7 | 11:15 |
| Ours | **29.27** | **0.868** | **0.068** | **16.43** | **0.763** | 648.1 | **0.056** | **0.022** | **0.028** | **0.656** | **0.966** | **0.030** | 10.8 | 2:23 |

| Method | Subject Consis.↑ | Background Consis.↑ | Temporal Flicker↑ | Motion Smooth.↑ | Overall Consis.↑ | Aesthetic Quality↑ | Imaging Quality↑ |
|---|---|---|---|---|---|---|---|
| Traj. Attention | 95.49 | 94.98 | **96.45** | 98.87 | 23.79 | 52.36 | 67.68 |
| Traj. Crafter | 95.38 | **95.44** | 96.10 | 99.02 | 24.32 | 53.05 | 69.66 |
| Diff. As Shader | 94.99 | 94.62 | 96.08 | 98.65 | **24.33** | 53.19 | 66.40 |
| NVS-Solver | 95.18 | 94.43 | 94.50 | 98.26 | 23.98 | 52.38 | **72.19** |
| Ours | **95.69** | 94.99 | 95.97 | **99.05** | 23.91 | **53.61** | 69.36 |

Table 2: Quantitative results of Real Camera Motion. Top: Comparison on standard metrics. Bottom: VBench evaluation. Gray rows denote training-based methods.

| Method | Input-Faithfulness | | | GT-Faithfulness | | | Fidelity ↓ | | | Pose ↓ | | | Geometry |
|---|---|---|---|---|---|---|---|---|---|---|---|---|---|
| | PSNR↑ | SSIM↑ | LPIPS↓ | PSNR↑ | SSIM↑ | LPIPS↓ | FID | KID' | FVD | ATE | RRE | RTE | M3R↓ |
| Traj. Attention | 21.12 | 0.735 | 0.215 | 18.29 | 0.617 | 0.281 | 20.32 | 5.069 | 156.6 | 0.186 | 0.749 | 0.089 | 0.062 |
| Traj. Crafter | 22.77 | 0.793 | 0.164 | 18.71 | 0.645 | 0.269 | 19.30 | 5.437 | 127.4 | 0.394 | 0.771 | 0.190 | **0.057** |
| Diff. As Shader | 12.92 | 0.452 | 0.523 | 12.70 | 0.431 | 0.559 | 26.50 | 5.380 | 304.0 | 0.349 | 1.085 | 0.182 | 0.066 |
| NVS-Solver | 20.51 | 0.720 | 0.223 | 16.91 | 0.572 | 0.300 | 15.80 | **1.789** | 128.9 | 0.392 | 0.915 | 0.620 | 0.064 |
| Ours | **28.95** | **0.893** | **0.074** | **18.97** | **0.652** | **0.241** | **15.56** | 1.792 | **106.1** | **0.114** | **0.637** | **0.053** | 0.060 |

| Method | Subject Consis.↑ | Background Consis.↑ | Temporal Flicker↑ | Motion Smooth.↑ | Overall Consis.↑ | Aesthetic Quality↑ | Imaging Quality↑ |
|---|---|---|---|---|---|---|---|
| Traj. Attention | 92.11 | 93.19 | **94.38** | **97.82** | 24.02 | 48.81 | 59.43 |
| Traj. Crafter | **92.86** | **93.60** | 93.33 | 97.43 | **24.78** | 50.87 | 63.51 |
| Diff. As Shader | 90.61 | 92.32 | 94.27 | 97.39 | 24.52 | 50.13 | 55.17 |
| NVS-Solver | 92.81 | 92.75 | 91.23 | 96.55 | 24.45 | 50.19 | **70.78** |
| Ours | 92.53 | 93.16 | 92.84 | 97.16 | 24.23 | **51.81** | 66.65 |

proach successfully generates videos that adhere to the camera path. Therefore, our VBench scores reflect the quality of a genuinely challenging, motion-consistent generation, confirming a superior balance between adherence and visual quality. Moreover, our zero-shot approach is significantly more efficient than NVS-Solver and rivals, if not surpasses, training-based methods in efficiency.

## 5.3 COMPARISON RESULTS: REAL CAMERA MOTION

From Table 2, we observe a clear positive correlation between Input-Faithfulness and GT-Faithfulness. Notably, our method, which strictly prioritizes Input-Faithfulness, also leads in GT-

Table 3: Ablation study on DAVIS. B: Baseline, SAR: SA-RePaint; KID' denotes KID$\times 10^3$, $T_{.25}$ ($T_{.50}$) represents TSED with a threshold of 0.25 (0.50), and M3R stands for MEt3R.

| Method | Input-Faithfulness | | | Fidelity ↓ | | | Camera Pose ↓ | | | Geometry | | |
|---|---|---|---|---|---|---|---|---|---|---|---|---|
| | PSNR↑ | SSIM↑ | LPIPS↓ | FID | KID' | FVD | ATE | RRE | RTE | $T_{.25}$↑ | $T_{.50}$↑ | M3R↓ |
| B (RePaint+SEG) | 29.89 | 0.866 | 0.076 | 30.19 | 1.410 | 714.1 | 0.067 | 0.025 | 0.033 | 0.674 | 0.958 | 0.032 |
| w/o Homography | 29.50 | 0.864 | 0.075 | 28.01 | 0.862 | 702.1 | 0.108 | 0.031 | 0.043 | 0.641 | 0.959 | 0.033 |
| w/o SAR ($\Sigma = 0$) | 29.95 | 0.866 | 0.075 | 30.35 | 1.460 | 709.2 | 0.050 | 0.021 | 0.027 | 0.713 | 0.968 | 0.032 |
| w/o SAR ($\Sigma = t_{i+1}$) | 29.65 | 0.864 | 0.077 | 28.99 | 1.003 | 699.0 | 0.049 | 0.021 | 0.027 | 0.683 | 0.967 | 0.032 |
| All | 29.58 | 0.864 | 0.074 | 28.14 | 0.816 | 705.2 | 0.051 | 0.022 | 0.027 | 0.672 | 0.964 | 0.033 |

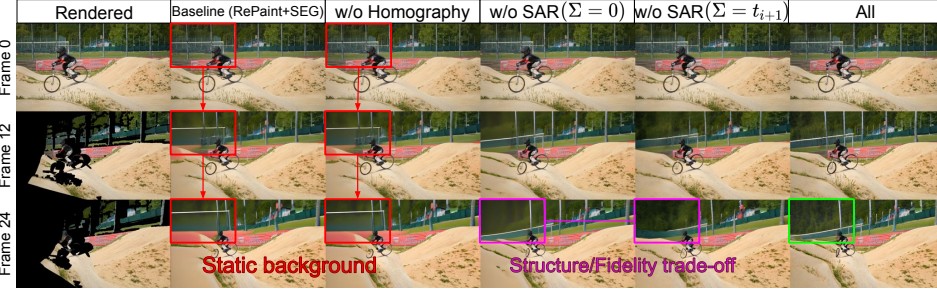

Figure 7: Visualization of each component's role. Without homography, the background remains static (red boxes). Without SA-RePaint, either global structure or texture fidelity is compromised (purple boxes). 'All' successfully overcomes all these issues (green box).

**Faithfulness.** This result validates our stance: by encouraging diffusion models to adhere to the input guidance as closely as possible, we not only resolve the practically prominent issues as pointed in Fig. 1 but also contribute to higher accuracy in terms of "conventional" faithfulness.

On the other hand, we observe a general decline in VBench metrics such as Motion Smoothness and Temporal Flicker compared to the Scripted setting. This trend is attributable to the inherent complexity of real-world trajectories, which often contain handheld jitter and irregular velocities, posing greater challenges for temporal consistency than smooth, synthesized paths.

### 5.4 ABLATION STUDY

In Table 3, our RePaint-based baseline (B) validates its design selection by achieving exceptional faithfulness. However, its low fidelity and pose accuracy scores highlight the two critical limitations this approach introduces: geometric inconsistency (*drifting synthesis*) and a poor *structure-texture trade-off*. Our components respectively target these limitations: Homography Deformation (Sec.4.1) is crucial for geometric consistency; its removal severely degrades camera pose accuracy, though this strict enforcement comes at a slight cost to perceptual fidelity (FID). In contrast, SA-RePaint (Sec.4.2) primarily enhances texture fidelity. Its inclusion markedly improves FID/KID by generating richer details while preserving structural coherence (Fig. 7). The slight decrease in TSED with SAR is expected. The realistic details SAR adds to formerly untextured regions enable more stringent feature matching in TSED, penalizing SAR more heavily for minor geometric deviations, thereby lowering the score. Our full model ('All') synergizes these strengths, trading a negligible decrease in faithfulness for notable gains in fidelity and pose accuracy to achieve a superior overall balance. Further hyperparameter ablations are in Appendix J.

### 6 CONCLUSION

We introduced a zero-shot novel view synthesis pipeline to address the trade-off between faithfulness, fidelity, and efficiency. Our method, featuring Test-Time Latent Homography Deformation and Spatially Adaptive RePaint (SA-RePaint), demonstrates that significant gains in faithfulness are achievable without disproportionate trade-offs in perceptual quality. By rebalancing the NVS objectives towards faithfulness while maintaining computational efficiency (under 11 GB VRAM), our work offers a promising and accessible direction for applications where authenticity is paramount.

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

## A    ADDITIONAL SUPPLEMENTARY MATERIALS

Our anonymized code is uploaded to `https://anonymous.4open.science/r/FaithfulNVS-9EDF`. Also, please refer to the supplementary videos for more visualization.

## B  "ADD_NOISE" FORMULATION IN SVD

**Lemma 1.** *Let $z_0$ be the clean latent tensor. Let $s$ and $t$ be real numbers with $0 < s < t$. Under the EDM (Karras et al., 2022) framework, the operation* add_noise$(\cdot, s \to t)$ *to add an i.i.d. Gaussian noise on $z_s \sim \mathcal{N}(z_0, s^2 I)$ to generate a new random variable $z_t \sim \mathcal{N}(z_0, t^2 I)$ is given by*

$$z_t = \text{add\_noise}(z_s, s \to t) := z_s + \sqrt{t^2 - s^2}\,\epsilon, \quad \epsilon \sim \mathcal{N}(0, I). \tag{12}$$

*Proof.* Since $z_t$ is an affine transformation of independent Gaussian variables $z_s$ and $\epsilon$, it also follows a Gaussian distribution. We proceed by computing its mean and variance. The mean is given by:

$$E[z_t] = E[z_s + \sqrt{t^2 - s^2}\,\epsilon] = E[z_s] + \sqrt{t^2 - s^2}\,E[\epsilon] = z_0 + \sqrt{t^2 - s^2} \cdot 0 = z_0. \tag{13}$$

As $z_s$ and $\epsilon$ are independent, the variance is the sum of their variances:

$$\text{Var}[z_t] = \text{Var}[z_s + \sqrt{t^2 - s^2}\,\epsilon] \tag{14}$$
$$= \text{Var}[z_s] + \text{Var}[\sqrt{t^2 - s^2}\,\epsilon] \tag{15}$$
$$= \text{Var}[z_s] + (t^2 - s^2)\,\text{Var}[\epsilon] \tag{16}$$
$$= s^2 I + (t^2 - s^2)I = t^2 I. \tag{17}$$

Thus, we have shown that $z_t \sim \mathcal{N}(z_0, t^2 I)$. $\qquad\square$

## C  PROOF OF THEOREM 1

**Theorem 1.** *The optimal $s^* \in [0, t]$ satisfying*

$$s^* = \underset{0 \le s \le t}{\arg\min} \left\| \text{Var}[\hat{z}_{s|t}] - \text{Var}[z_s] \right\|_1 \tag{18}$$

*is an element of the set:*

$$s^* \in \{\eta_+, \eta_-, 0, t\}, \quad \text{where } \eta_\pm := \text{clip}\left(\frac{-B \pm \sqrt{\max(0, B^2 - AC)}}{A}; 0, t\right). \tag{19}$$

*Here, A, B, and C are given by the following:*

$$A = \text{Var}[D(z_t)] - 1 \tag{20}$$
$$B = \text{Cov}[\hat{z}_{0|t}, D(z_t)] \tag{21}$$
$$C = \text{Var}[\hat{z}_{0|t}] - \text{Var}[z_0] \tag{22}$$

*where $D(z_t) := (z_t - \hat{z}_{0|t})/t$, and $z_0$ is the ground truth clean latent. $\text{Cov}[\hat{z}_{0|t}, D(z_t)]$ stands for the covariance between $\hat{z}_{0|t}$ and $D(z_t)$.*

*Proof.* Recall the definitions of $z_s$ and $\hat{z}_{s|t}$:

$$z_s = z_0 + s\epsilon, \quad \epsilon \sim \mathcal{N}(0, I) \tag{23}$$
$$\hat{z}_{s|t} = \left(1 - \frac{s}{t}\right)\hat{z}_{0|t} + \left(\frac{s}{t}\right)z_t \tag{24}$$

Taking the variance of Eq. 23 (assuming the independence between $z_0$ and $\epsilon$), we get:

$$\text{Var}[z_s] = \text{Var}[z_0] + s^2 \tag{25}$$

Next, we simplify the expression for $\hat{z}_{s|t}$ by defining $D(z_t) := (z_t - \hat{z}_{0|t})/t$.

$$\hat{z}_{s|t} = \hat{z}_{0|t} + \frac{s}{t}(z_t - \hat{z}_{0|t}) = \hat{z}_{0|t} + sD(z_t) \tag{26}$$

The variance of Eq. 24 is then a quadratic function of $s$:

$$\text{Var}[\hat{z}_{s|t}] = \text{Var}[\hat{z}_{0|t} + sD(z_t)] = \text{Var}[D(z_t)]s^2 + 2\text{Cov}[\hat{z}_{0|t}, D(z_t)]s + \text{Var}[\hat{z}_{0|t}] \tag{27}$$

The objective function in Eq. 18 is the absolute value of the difference between these two variances. Let this difference be $f(s)$:

$$f(s) := \text{Var}[\hat{z}_{s|t}] - \text{Var}[z_s] \tag{28}$$

$$= \underbrace{(\text{Var}[D(z_t)] - 1)}_{A} s^2 + 2 \underbrace{\text{Cov}[\hat{z}_{0|t}, D(z_t)]}_{B} s + \underbrace{(\text{Var}[\hat{z}_{0|t}] - \text{Var}[z_0])}_{C} \tag{29}$$

$$= As^2 + 2Bs + C \tag{30}$$

Our goal is to find $\arg\min_{s \in [0,t]} |f(s)|$. In an unconstrained setting $s \in \mathbb{R}$, the minimizers of $|f(s)|$ are given by the roots of $f(s) = 0$ if they are real, or by the vertex of the parabola $s = -B/A$ if the roots are complex. Both cases are compactly represented by:

$$s = \frac{-B \pm \sqrt{\max(0, B^2 - AC)}}{A} \tag{31}$$

When considering the constraint $s \in [0,t]$, the optimal value $s^*$ must be found within the set of candidates comprising the unconstrained solutions that lie in $[0,t]$ and the interval's boundaries, $0$ and $t$. Therefore, $s^*$ must be an element of the set given in the theorem statement. One can determine the true minimum by evaluating $|f(s)|$ for each candidate. $\square$

**Notes for implementation:** The equation $f(s) = 0$ can have two real roots, both of which may lie in $[0,t]$. In this case, we select the root using the following formula:

$$s^* = \frac{-B + \text{sign}(B)\sqrt{\max(0, B^2 - AC)}}{A}. \tag{32}$$

This choice is motivated by the ideal case where the model's prediction is perfect, i.e., $\hat{z}_{0|t} = z_0$. Given the construction $z_t = z_0 + t\epsilon$ where $\epsilon \sim \mathcal{N}(0, I)$, we have:

$$D(z_t) = ((z_0 + t\epsilon) - z_0)/t = \epsilon \tag{33}$$

$$A = \text{Var}[\epsilon] - 1 = 0 \tag{34}$$

$$B = \text{Cov}[z_0, \epsilon] = 0 \tag{35}$$

$$C = \text{Var}[z_0] - \text{Var}[z_0] = 0 \tag{36}$$

In this ideal scenario, all coefficients are zero. In practice, this means that $A$ can be a small value, making the solution for $s^*$ sensitive to division by $A$. To mitigate the risk of large perturbations in $s^*$ due to a small $A$, we select the root whose numerator has a smaller absolute value, leading to Eq. 32.

## D  SA-REPAINT IMPLEMENTATION

Algorithm 1 describes our implementation of Section 4.2. Note that the operations to deduce $\Sigma$ are pixelwise. As a specific detail of the process, we apply guided filtering (He et al., 2012) to the computed covariances with $\text{Var}[z_0]$ as a guide. This step serves two purposes: firstly, to smooth the coefficients $A$, $B$, and $C$, which are prone to noise due to being close to zero (see the proof of Appendix C); and secondly, to guide the solution such that $\Sigma$ correlates with the image's structure, a behavior we expect, where it is lower in textureless areas and higher in high-frequency, complex regions. Moreover, particular care must be taken with the division $\texttt{nunom}/A$. Although both $\texttt{nunom}$ and $A$ are spatio-temporally smooth, the coefficient map $A$ is expected to be close to $0$ as shown in the note of C. Therefore, pixel-wise division $\texttt{nunom}/A$ is very susceptible to the sign flips and the slight fluctuation on $A$, resulting in a highly unstable noise level map $\Sigma = \text{clip}(\texttt{nunom}/A; 0, t)$. We circumvent this instability by employing a simple yet effective technique that reformulates the division as solving a local least-squares problem.

**Definition of** $\texttt{safe\_division}$  For simplicity, let $P, Q \in \mathbb{R}^{H \times W}$ be two single-channel images. Our goal is to compute a ratio map $R$ that is more robust to noise and spatially coherent than the pointwise division $P/Q$. To achieve this, we assume that the ratio is locally constant within a small

---

**Algorithm 1:** SA-RePaint (Section 4.2)

---

**Input:** Current noisy latent $z_t \in \mathbb{R}^{F \times C \times H \times W}$, one-step denoised latent $\hat{z}_{0|t} \in \mathbb{R}^{F \times C \times H \times W}$,
    rendered image latent $y \in \mathbb{R}^{F \times C \times H \times W}$

**Hyperparameters:** Variance scaler $\lambda_{\text{var}} \in \mathbb{R}$, local spatial/temporal window radius $r_s, r_t \in \mathbb{R}$

**Output:** Noise level map $\Sigma \in [0, t]^{F \times C \times H \times W}$

1  $\text{Var}[z_0] \leftarrow \lambda_{\text{var}} \cdot \texttt{get\_var\_data}(y_{0:1})$ ;                                    /\* Eq. 10 \*/
2  $D(z_t) \leftarrow (z_t - \hat{z}_{0|t})/t$ ;
3  $\text{Var}[\hat{z}_{0|t}] \leftarrow \texttt{guided\_filter}\left(\texttt{local\_covariance}(\hat{z}_{0|t}, \hat{z}_{0|t}, r_s, r_t), \text{Var}[z_0]\right)$ ;
4  $\text{Var}[D(z_t)] \leftarrow \texttt{guided\_filter}\left(\texttt{local\_covariance}(D(z_t), D(z_t), r_s, r_t), \text{Var}[z_0]\right)$ ;
5  $\text{Cov}[\hat{z}_{0|t}, D(z_t)] \leftarrow \texttt{guided\_filter}\left(\texttt{local\_covariance}(\hat{z}_{0|t}, D(z_t), r_s, r_t), \text{Var}[z_0]\right)$ ;
6  $A \leftarrow \text{Var}[D(z_t)] - 1;\ B \leftarrow \text{Cov}[\hat{z}_{0|t}, D(z_t)];\ C \leftarrow \text{Var}[\hat{z}_{0|t}] - \text{Var}[z_0]$ ;      /\* Eq. 9 \*/
7  $\texttt{nunom\_pos} \leftarrow -B + \text{sign}(B)\sqrt{\max(0, B^2 - AC)}$ ;
8  $\texttt{nunom\_neg} \leftarrow -B - \text{sign}(B)\sqrt{\max(0, B^2 - AC)}$ ;
9  $\Sigma_{\text{pos}} \leftarrow \text{clip}(\texttt{safe\_division}(\texttt{nunom\_pos}, A); 0, t)$ ;                    /\* (See below) \*/
10  $\Sigma_{\text{neg}} \leftarrow \text{clip}(\texttt{safe\_division}(\texttt{nunom\_neg}, A); 0, t)$ ;
11  $\Sigma \leftarrow (A\Sigma_{\text{pos}}^2 + B\Sigma_{\text{pos}} + C \leq A\Sigma_{\text{neg}}^2 + B\Sigma_{\text{neg}} + C)\ ?\ \Sigma_{\text{pos}} : \Sigma_{\text{neg}}$ ;      /\* Eq. 8 \*/
12  $\Sigma \leftarrow (A\Sigma_{\text{pos}}^2 + B\Sigma_{\text{pos}} + C \leq A \cdot 0^2 + B \cdot 0 + C)\ ?\ \Sigma_{\text{pos}} : 0$ ;
13  $\Sigma \leftarrow (A\Sigma_{\text{pos}}^2 + B\Sigma_{\text{pos}} + C \leq A \cdot t^2 + B \cdot t + C)\ ?\ \Sigma_{\text{pos}} : \texttt{t}$ ;
14  **return** $\Sigma$

---

window $W(p)$ around each pixel $p$. We then find the optimal ratio $R(p)$ that minimizes the sum of squared errors within this window:

$$R(p) = \underset{r \in \mathbb{R}}{\arg\min} \sum_{q \in W(p)} (P(q) - r \cdot Q(q))^2 \tag{37}$$

This is a convex quadratic minimization problem. By setting the derivative of the objective function with respect to $r$ to zero, we obtain the analytical solution:

$$R(p) = \frac{\sum_{q \in W(p)} P(q)Q(q)}{\left(\sum_{q \in W(p)} Q(q)^2\right) + \epsilon} \tag{38}$$

where $\epsilon > 0$ is a small constant added for numerical stability to prevent division by zero.

This method can be readily extended to multi-channel images by applying it channel-wise, or to videos by defining $W(p)$ as a 3D spatio-temporal window. In summary, `safe_division` is defined as follows:

$$\frac{\texttt{nunom}}{A} \approx \texttt{safe\_divison}(\texttt{nunom}, A) := \text{clip}\left(\frac{\texttt{box\_blur}(\texttt{nunom} \cdot A)}{\texttt{box\_blur}(A^2) + \epsilon}; 0, t\right) \tag{39}$$

## E   TEST-TIME HOMOGRAPHY IMPLEMENTATION

Algorithm 2 delineates the test-time homography optimization described in Section 4.1. We empirically observed that $N_{\text{iter}} = 100$ and $\eta = 0.01$ were enough for $\{H_j\}_{j=0}^{F-1}$ to converge, and the iteration completes in a second.

Figure 9 shows the process of how homographies rectify the intermediate denoised images. At the beginning of the denoising steps, the image color and structure are not well established, so the deduced homographies $\{H_j\}_{j=0}^{F-1}$ strongly act on the denoised latents $\hat{z}_{0|t}$. Over time, the latents comply with the global motion, and the homography deformation magnitude is reduced. This deformation magnitude is quantified by calculating the

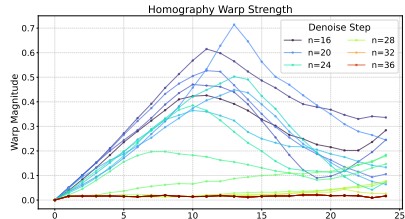

Figure 8: Evolution of homography deformation strength over denoising steps.

---

**Algorithm 2:** Test-time Latent Homography Deformation (Section 4.1)

---

**Input:** One-step denoised latent $\hat{z}_{0|t} \in \mathbb{R}^{F \times C \times H \times W}$, rendered image latent $y \in \mathbb{R}^{F \times C \times H \times W}$,
       rendered image valid mask $m^{\text{valid}} \in \{0, 1\}^{F \times 1 \times H \times W}$

**Hyperparameters:** Regularization weight $\lambda_H$, learning rate $\eta$, Max iterations $N_{iter}$

**Output:** Aligned latent $\hat{z}_{0|t}^{\text{deformed}} \in \mathbb{R}^{F \times C \times H \times W}$

**1** Initialize the homographies $\{H_j \in \text{PGL}(3, \mathbb{R})\}_{j=0}^{F-1}$ such that $H_j = [1, 0, 0; 0, 1, 0; 0, 0, 1]$ ;

**2** Initialize $\texttt{optimizer} = \texttt{Adam}([H'_0, \ldots, H'_{F-1}], \texttt{lr} = \eta)$, where $H'_j \in \mathbb{R}^8 = \texttt{flatten}(H_j)[: 8]$
  ;

**3 for** $i \leftarrow 1$ **to** $N_{iter}$ **do**

**4**     Warp latents: $z_{0|t}^{\text{deformed}}[j] \leftarrow H_j \, \hat{z}_{0|t}[j]$ ;

**5**     Reconstruction loss: $\mathcal{L}_H^{\text{reconst}} \leftarrow \sum_{j=1}^{F-1} \left\| \left(y[j] - \hat{z}_{0|t}^{\text{deformed}}[j]\right) \cdot m^{\text{valid}}[j] \right\|_1$ ;

**6**     Smoothness loss: $\mathcal{L}_H^{\text{smooth}} \leftarrow \sum_{j=1}^{F-2} \left\| H_{j+1} - 2H_j + H_{j-1} \right\|_1$ ;

**7**     Sum up the losses: $\mathcal{L} \leftarrow \mathcal{L}_H^{\text{reconst}} + \lambda_H \mathcal{L}_H^{\text{smooth}}$ ;

**8**     Update parameters: $H' \leftarrow \texttt{optimizer}(\nabla_H \mathcal{L}, \eta)$.

**9** Warp latents: $z_{0|t}^{\text{deformed}}[j] \leftarrow H_j \, \hat{z}_{0|t}[j]$ ;

**10 return** $\hat{z}_{0|t}^{\text{deformed}}$

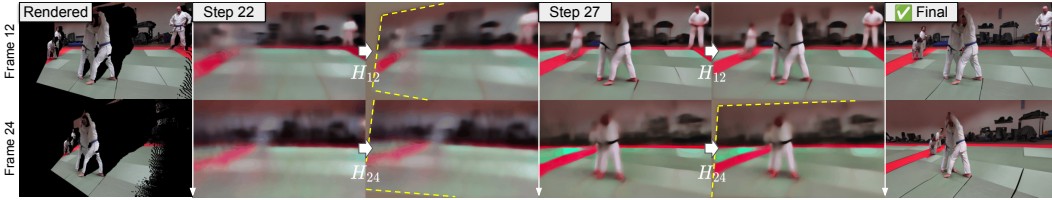

Figure 9: Homography deformation at 22th and 27th denoising steps, with 12th and 24th frames aligned vertically. We see that homographies try to align the latents $\hat{z}_{0|t}$ to the rendered images.

average of the distances four image corners move. Figure 8 shows this result. Note that the deformation peaks at around frame 12, and then decreases in later frames. This is because the rendered images contain fewer valid regions as the camera moves more, resulting in a weaker guiding signal for homography optimization. In practice, this doesn't cause any problems; as we see in the graph, the homography eventually converges to the identity.

## F    SMOOTHED ENERGY GUIDANCE (SEG) WITH BLUR WEIGHTING

The methods introduced in Sec. 4 primarily enhance faithfulness while maintaining fidelity, but the fidelity itself remains bounded by the underlying model capabilities. Indeed, Latent Homography Deformation (Section 4.1) promotes global structural consistency, while SA-RePaint (Section 4.2) manages the fidelity–faithfulness tradeoff.

To purely improve fidelity in a zero-shot manner, we employ Smoothed Energy Guidance (SEG) (Hong, 2024), a variant of classifier-free guidance (CFG) (Ho & Salimans, 2022). SEG replaces the unconditional prediction in CFG with a conditional one with attention maps blurred:

$$\text{Attn}(q_{t_i}, k_{t_i}^{w_i}, v_{t_i}; G_\sigma) = \text{softmax}\left(\frac{(G_\sigma * q_{t_i}) k_{t_i}^{w_i \top}}{\sqrt{d}}\right) v_{t_i} \tag{40}$$

where $G_\sigma$ denotes a Gaussian blur kernel with standard deviation $\sigma$.

However, directly applying SEG in our task results in unstable neon artifacts in invalid regions. We attribute this to texture inconsistencies: since the invalid regions are already blurry, further blurring reduces guidance effectiveness, leading to nearly unconditional generation. To address this, we propose **spatially adaptive blurring**: we set a lower blur sigma $\sigma_{\text{invalid}}$ for invalid pixels and a higher one $\sigma_{\text{valid}}$ for valid pixels, ensuring $0 < \sigma_{\text{invalid}} < \sigma_{\text{valid}}$. This mitigates neon effects in inpainted regions while sharpening object boundaries around valid/invalid area borders.

We use $\sigma_{\text{invalid}} = 2$ and $\sigma_{\text{valid}} = 4$. To encourage novel structure generation in invalid regions, we alternate between standard CFG and SEG at a $2 : 1$ ratio.

## G  IMPLEMENTATION DETAILS

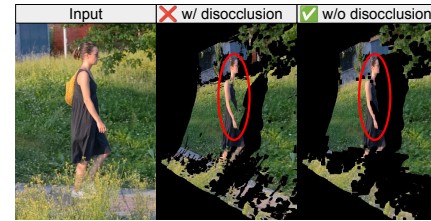

Our pipeline operates on a $1024 \times 576$ resolution. During rendering, we discard disoccluded pixels, as they potentially lead to see-through artifacts as illustrated in Fig. 10. We use $N = 50$ denoising steps, with inference starting at step $i_0 = 16$. For Homography Deformation (Sec. 4.1), the regularization weight is $\lambda_H = 0.5$, and this process is disabled after step 30. For SA-RePaint (Sec. 4.2), we set $\Sigma = 0$ for the first 25 steps for stability, then make it adaptive with a local window radius of $r = 1$ for variance computation.

Figure 10: We mask out the disoccluded region (right) to prevent potential see-through artifacts (middle).

To further boost perceptual quality, we incorporate Smoothed Energy Guidance (SEG; Appendix F). In the final 10 denoising steps, we disable all our proposed components to allow the model to harmonize the boundaries between valid and inpainted regions. We empirically found that these final 10 steps of free generation have a negligible impact on overall faithfulness.

## H  DETAILED QUANTITATIVE RESULTS

Tables 6-5 report the quantitative metrics and the VBench scores for Mannequin Challenge, DAVIS, and Tanks and Temples, respectively. KID' denotes KID$\times 10^3$, $T_{.25}$ ($T_{.50}$) represents TSED with a threshold of 0.25 (0.50), and M3R stands for MEt3R.

## I  EFFICIENCY EVALUATION

To determine the minimal required memory, we identify the lowest memory limit at which our inference pipeline can run without an Out-of-Memory (OOM) error. This is achieved by progressively lowering the maximum allocatable memory using `torch.cuda.set_per_process_memory_fraction`. We chose this method over querying `torch.cuda.max_memory_allocated` because the latter fails to account for memory reserved by the CUDA context and the PyTorch caching allocator. Consequently, our approach provides a more realistic measure of the total memory footprint in an actual execution environment. Inference time, conversely, was measured without any memory cap, focusing specifically on the iterative process of the denoising U-Net. We run each model three times and report the median processing time. All experiments were conducted on a single NVIDIA A6000 GPU with 48 GB of VRAM.

## J  FURTHER ABLATION STUDIES

### J.1  EFFECT OF $\lambda_H$ IN LATENT HOMOGRAPHY DEFORMATION

In Section 4.1, we defined the test-time loss function of homography deformation as follows: $\mathcal{L}_H = \mathcal{L}_H^{\text{recon}} + \lambda_H \mathcal{L}_H^{\text{smooth}}$. This section ablates the effect of $\lambda_H$, i.e., how the smoothness regularization affects the final generation results. Table 8 shows that the camera pose errors for $\lambda_H = 0$ are notably higher than those for $\lambda_H > 0$. We attribute this to the homography overfitting to each frame independently, leading to a loss of temporal consistency and producing a non-smooth motion trajectory distinct from the one used for rendering.

Table 4: Quantitative comparison on DAVIS (Scripted Camera Motion). Top: Comparison on standard metrics. Bottom: VBench evaluation. Gray rows are training-based.

| Method | Input-Faithfulness | | | Fidelity ↓ | | | Camera Pose ↓ | | | Geometry | | |
|---|---|---|---|---|---|---|---|---|---|---|---|---|
| | PSNR↑ | SSIM↑ | LPIPS↓ | FID | KID' | FVD | ATE | RRE | RTE | T.25↑ | T.50↑ | M3R↓ |
| Traj. Attention | 22.93 | 0.725 | 0.195 | 30.73 | 1.855 | 647.8 | 0.273 | 0.066 | 0.108 | 0.406 | 0.894 | 0.035 |
| Traj. Crafter | 24.24 | 0.811 | 0.119 | 32.03 | 1.906 | 704.7 | 0.139 | 0.047 | 0.071 | 0.499 | 0.958 | 0.032 |
| Diff. As Shader | 14.79 | 0.434 | 0.432 | 32.60 | 2.036 | 492.5 | 1.782 | 0.543 | 1.409 | 0.038 | 0.092 | 0.044 |
| NVS-Solver | 21.78 | 0.695 | 0.209 | 27.92 | 1.108 | 665.9 | 0.909 | 0.200 | 0.394 | 0.326 | 0.715 | 0.041 |
| Ours | 29.58 | 0.864 | 0.074 | 28.14 | 0.816 | 705.2 | 0.051 | 0.022 | 0.027 | 0.672 | 0.964 | 0.033 |

| Method | Subject Consis.↑ | Background Consis.↑ | Temporal Flicker↑ | Motion Smooth.↑ | Overall Consis.↑ | Aesthetic Quality↑ | Imaging Quality↑ |
|---|---|---|---|---|---|---|---|
| Traj. Attention | 95.00 | 94.80 | 96.64 | 98.89 | 23.94 | 51.12 | 62.73 |
| Traj. Crafter | 94.92 | 95.28 | 96.20 | 99.05 | 24.44 | 51.93 | 65.97 |
| Diff. As Shader | 94.26 | 94.34 | 95.92 | 98.51 | 24.77 | 52.50 | 62.25 |
| NVS-Solver | 94.68 | 94.27 | 94.66 | 98.21 | 24.11 | 51.07 | 68.52 |
| Ours | 95.33 | 94.92 | 96.03 | 99.05 | 24.07 | 52.35 | 65.39 |

Table 5: Quantitative comparison on Tanks and Temples (Scripted Camera Motion). Top: Comparison on standard metrics. Bottom: VBench evaluation. Gray rows are training-based.

| Method | Input-Faithfulness | | | Fidelity ↓ | | | Camera Pose ↓ | | | Geometry | | |
|---|---|---|---|---|---|---|---|---|---|---|---|---|
| | PSNR↑ | SSIM↑ | LPIPS↓ | FID | KID' | FVD | ATE | RRE | RTE | T.25↑ | T.50↑ | M3R↓ |
| Traj. Attention | 23.08 | 0.737 | 0.157 | 21.71 | 1.985 | 1052 | 0.258 | 0.061 | 0.093 | 0.353 | 0.936 | 0.028 |
| Traj. Crafter | 23.99 | 0.798 | 0.109 | 21.57 | 1.884 | 1146 | 0.126 | 0.050 | 0.077 | 0.446 | 0.971 | 0.027 |
| Diff. As Shader | 15.04 | 0.470 | 0.362 | 23.44 | 1.938 | 891.8 | 1.304 | 0.439 | 1.161 | 0.041 | 0.165 | 0.030 |
| NVS-Solver | 22.04 | 0.729 | 0.169 | 19.96 | 1.254 | 1067 | 0.310 | 0.124 | 0.221 | 0.310 | 0.808 | 0.034 |
| Ours | 28.98 | 0.872 | 0.063 | 19.95 | 0.941 | 1025 | 0.058 | 0.022 | 0.029 | 0.641 | 0.967 | 0.027 |

| Method | Subject Consis.↑ | Background Consis.↑ | Temporal Flicker↑ | Motion Smooth.↑ | Overall Consis.↑ | Aesthetic Quality↑ | Imaging Quality↑ |
|---|---|---|---|---|---|---|---|
| Traj. Attention | 95.99 | 95.16 | 96.26 | 98.84 | 23.65 | 53.60 | 72.63 |
| Traj. Crafter | 95.84 | 95.61 | 96.01 | 99.00 | 24.20 | 54.17 | 73.35 |
| Diff. As Shader | 95.73 | 94.90 | 96.23 | 98.80 | 23.90 | 53.88 | 70.54 |
| NVS-Solver | 95.68 | 94.58 | 94.34 | 98.32 | 23.86 | 53.69 | 75.86 |
| Ours | 96.05 | 95.06 | 95.90 | 99.04 | 23.74 | 54.87 | 73.33 |

Table 6: Quantitative comparison on Mannequin Challenge (Real Camera Motion). Top: Comparison on standard metrics. Bottom: VBench evaluation. Gray rows are training-based.

| Method | Input-Faithfulness | | | GT-Faithfulness | | | Fidelity ↓ | | | Camera Pose ↓ | | | Geometry |
|---|---|---|---|---|---|---|---|---|---|---|---|---|---|
| | PSNR↑ | SSIM↑ | LPIPS↓ | PSNR↑ | SSIM↑ | LPIPS↓ | FID | KID' | FVD | ATE | RRE | RTE | M3R↓ |
| Traj. Attention | 21.96 | 0.783 | 0.168 | 18.43 | 0.639 | 0.250 | 31.49 | 4.371 | 230.2 | 0.187 | 0.509 | 0.091 | 0.049 |
| Traj. Crafter | 24.48 | 0.855 | 0.119 | 18.96 | 0.662 | 0.243 | 30.16 | 4.302 | 221.4 | 0.196 | 0.552 | 0.092 | 0.044 |
| Diff. As Shader | 13.06 | 0.468 | 0.500 | 12.79 | 0.448 | 0.538 | 40.30 | 5.824 | 598.2 | 0.162 | 0.620 | 0.082 | 0.052 |
| NVS-Solver | 21.95 | 0.775 | 0.184 | 17.20 | 0.591 | 0.274 | 29.93 | 3.733 | 251.7 | 0.193 | 0.528 | 0.093 | 0.054 |
| Ours | 30.51 | 0.925 | 0.056 | 19.03 | 0.663 | 0.226 | 26.18 | 1.865 | 187.1 | 0.061 | 0.424 | 0.031 | 0.047 |

| Method | Subject Consis.↑ | Background Consis.↑ | Temporal Flicker↑ | Motion Smooth.↑ | Overall Consis.↑ | Aesthetic Quality↑ | Imaging Quality↑ |
|---|---|---|---|---|---|---|---|
| Traj. Attention | 93.92 | 93.43 | 95.00 | 98.56 | 24.10 | 51.22 | 63.44 |
| Traj. Crafter | 94.58 | 93.97 | 94.58 | 98.49 | 24.75 | 52.17 | 64.88 |
| Diff. As Shader | 93.52 | 92.90 | 94.93 | 98.02 | 25.13 | 52.37 | 57.75 |
| NVS-Solver | 93.81 | 92.50 | 92.15 | 97.69 | 24.35 | 50.86 | 71.82 |
| Ours | 94.43 | 93.31 | 94.07 | 98.36 | 24.17 | 53.60 | 67.48 |

Table 7: Quantitative comparison on DL3DV-Evaluation (Real Camera Motion). Top: Comparison on standard metrics. Bottom: VBench evaluation. Gray rows are training-based.

| Method | Input-Faithfulness | | | GT-Faithfulness | | | Fidelity ↓ | | | Camera Pose ↓ | | | Geometry |
|---|---|---|---|---|---|---|---|---|---|---|---|---|---|
| | PSNR↑ | SSIM↑ | LPIPS↓ | PSNR↑ | SSIM↑ | LPIPS↓ | FID | KID' | FVD | ATE | RRE | RTE | M3R↓ |
| Traj. Attention | 20.28 | 0.687 | 0.262 | 18.15 | 0.595 | 0.313 | 25.16 | 6.965 | 234.6 | 0.186 | 0.990 | 0.088 | 0.076 |
| Traj. Crafter | 21.05 | 0.732 | 0.209 | 18.46 | 0.627 | 0.296 | 23.17 | 7.467 | 188.6 | 0.593 | 0.990 | 0.287 | 0.071 |
| Diff. As Shader | 12.78 | 0.437 | 0.547 | 12.61 | 0.413 | 0.580 | 33.93 | 7.244 | 362.8 | 0.536 | 1.550 | 0.282 | 0.079 |
| NVS-Solver | 19.07 | 0.666 | 0.261 | 16.63 | 0.553 | 0.327 | 18.12 | 2.471 | 179.8 | 0.592 | 1.305 | 0.294 | 0.075 |
| Ours | 27.39 | 0.861 | 0.093 | 18.92 | 0.641 | 0.255 | 20.27 | 3.064 | 163.3 | 0.168 | 0.850 | 0.075 | 0.074 |

| Method | Subject Consis.↑ | Background Consis.↑ | Temporal Flicker↑ | Motion Smooth.↑ | Overall Consis.↑ | Aesthetic Quality↑ | Imaging Quality↑ |
|---|---|---|---|---|---|---|---|
| Traj. Attention | 90.30 | 92.95 | 93.76 | 97.08 | 23.93 | 46.40 | 55.43 |
| Traj. Crafter | 91.13 | 93.22 | 92.09 | 96.38 | 24.82 | 49.57 | 62.14 |
| Diff. As Shader | 87.70 | 91.74 | 93.62 | 96.76 | 23.90 | 47.89 | 52.58 |
| NVS-Solver | 91.81 | 93.00 | 90.31 | 95.41 | 24.55 | 49.53 | 69.74 |
| Ours | 90.64 | 93.01 | 91.61 | 95.96 | 24.28 | 50.01 | 65.83 |

Table 8: Quantitative comparison with different $\lambda_H$ on DAVIS. Top: Comparison on standard metrics. Bottom: VBench evaluation.

| | Input-Faithfulness | | | Fidelity ↓ | | | Camera Pose ↓ | | | TSED ↑ | |
|---|---|---|---|---|---|---|---|---|---|---|---|
| $\lambda_H$ | PSNR ↑ | SSIM ↑ | LPIPS ↓ | FID | KID $\times 10^3$ | FVD | ATE | RRE | RTE | @.25 | @.50 |
| 0.0 | 29.58 | 0.864 | 0.074 | 28.13 | 0.834 | 705.2 | 0.086 | 0.023 | 0.037 | 0.658 | 0.966 |
| 0.5 | 29.58 | 0.864 | 0.074 | 28.14 | 0.816 | 705.2 | 0.051 | 0.022 | 0.027 | 0.672 | 0.964 |
| 1.0 | 29.58 | 0.864 | 0.074 | 28.17 | 0.861 | 704.1 | 0.066 | 0.022 | 0.030 | 0.660 | 0.965 |
| 1.5 | 29.57 | 0.864 | 0.074 | 28.12 | 0.872 | 702.4 | 0.055 | 0.022 | 0.028 | 0.651 | 0.953 |

| $\lambda_H$ | Subject Consis.↑ | Background Consis.↑ | Temporal Flicker↑ | Motion Smooth.↑ | Overall Consis.↑ | Aesthetic Quality↑ | Imaging Quality↑ |
|---|---|---|---|---|---|---|---|
| 0.0 | 95.35 | 94.81 | 95.99 | 99.03 | 24.05 | 52.08 | 65.54 |
| 0.5 | 95.33 | 94.92 | 96.03 | 99.05 | 24.07 | 52.35 | 65.39 |
| 1.0 | 95.34 | 94.81 | 96.00 | 99.03 | 24.06 | 52.09 | 65.53 |
| 1.5 | 95.35 | 94.87 | 96.00 | 99.03 | 24.06 | 52.13 | 65.53 |

## J.2 EFFECT OF $\lambda_{\text{var}}$ IN SA-REPAINT

In Section 4.2.2, we proposed a method to approximate the latent variance $\text{Var}[z_0]$ using the latent variance map of the input image latent, $\text{Var}[y_{0:1}]$, and the attention correspondence matrix:

$$\text{Var}[z_0] \approx \lambda_{\text{var}} \cdot \text{softmax}\left(\frac{q_t \left(k_t^{w_i}[0]\right)^\top}{\sqrt{d}}\right) \text{Var}[y[0]] \qquad (41)$$

Here, we analyze the effect of the global scalar $\lambda_{\text{var}}$ on the model's qualitative and quantitative performance.

A higher value of $\lambda_{\text{var}}$ increases the overall variance $\text{Var}[z_0]$, which encourages the model to generate novel views with high-frequency textures, particularly in uncertain or occluded regions. This effect is corroborated in Figure 11, which shows that increasing $\lambda_{\text{var}}$ leads to richer low-level textures and the generation of new semantic structures.

However, this increased generation capability comes at the cost of reduced faithfulness to the source view. As shown in Tables 9, key metrics such as PSNR, camera pose accuracy, and TSED degrade as $\lambda_{\text{var}}$ increases. We also observe that temporal consistency and motion smoothness are diminished.

## J.3 EFFECT OF PRE-FILLING

In Figure 3a, we introduced the process of filling the void black regions in the rendered images by a classical inpainting algorithm (Bertalmio et al., 2001). This is a crucial step to avoid artifacts at the valid-invalid boundary. Figure 12 shows the difference with or without this prefilling step. The gray artifacts in the "w/o prefilling" appear exactly in the same position as the valid-invalid borders in

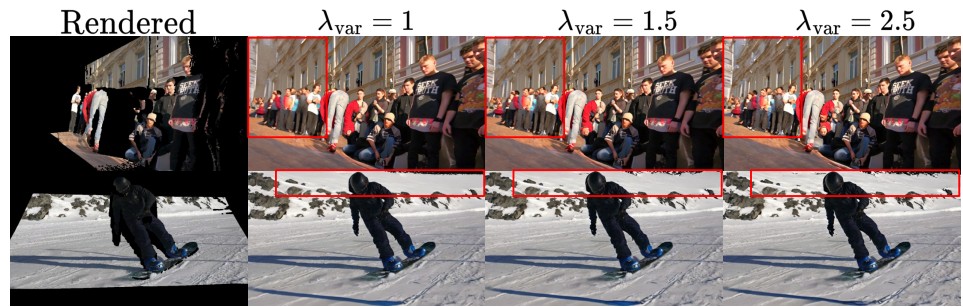

Figure 11: Fidelity enhancement by increasing $\lambda_{\mathrm{var}}$.

Table 9: Quantitative comparison with different $\lambda_{\mathrm{var}}$ on DAVIS. Top: Comparison on standard metrics. Bottom: VBench evaluation.

| $\lambda_{\mathrm{var}}$ | Input-Faithfulness | | | Fidelity ↓ | | | Camera Pose ↓ | | | TSED ↑ | |
| | PSNR ↑ | SSIM ↑ | LPIPS ↓ | FID | KID $\times 10^3$ | FVD | ATE | RRE | RTE | @.25 | @.50 |
|---|---|---|---|---|---|---|---|---|---|---|---|
| 1.0 | 29.66 | 0.864 | 0.074 | 28.32 | 0.876 | 703.8 | 0.081 | 0.023 | 0.032 | 0.684 | 0.966 |
| 1.5 | 29.58 | 0.864 | 0.074 | 28.14 | 0.816 | 705.2 | 0.051 | 0.022 | 0.027 | 0.672 | 0.964 |
| 2.0 | 29.54 | 0.864 | 0.074 | 28.03 | 0.831 | 705.1 | 0.063 | 0.022 | 0.030 | 0.649 | 0.957 |
| 2.5 | 29.51 | 0.864 | 0.074 | 27.99 | 0.838 | 703.6 | 0.061 | 0.022 | 0.030 | 0.662 | 0.962 |
| 3.0 | 29.48 | 0.864 | 0.074 | 27.95 | 0.832 | 701.9 | 0.061 | 0.022 | 0.029 | 0.645 | 0.964 |

| $\lambda_{\mathrm{var}}$ | Subject Consis.↑ | Background Consis.↑ | Temporal Flicker↑ | Motion Smooth.↑ | Overall Consis.↑ | Aesthetic Quality↑ | Imaging Quality↑ |
|---|---|---|---|---|---|---|---|
| 1.0 | 95.33 | 94.91 | 96.13 | 99.08 | 24.06 | 52.36 | 65.16 |
| 1.5 | 95.33 | 94.92 | 96.03 | 99.05 | 24.07 | 52.35 | 65.39 |
| 2.0 | 95.33 | 94.78 | 95.94 | 99.01 | 24.06 | 52.09 | 65.65 |
| 2.5 | 95.31 | 94.87 | 95.93 | 99.01 | 24.07 | 52.31 | 65.60 |
| 3.0 | 95.30 | 94.75 | 95.85 | 98.98 | 24.10 | 52.04 | 65.80 |

the rendered image. This strongly indicates that the artifact is caused by contamination of the black region during the VAE encoding process. Since our main focus is the faithfulness to the rendered images, the conditioning VAE latent must be prepared with care, unlike other previous works.

To ensure a fair comparison, Table 10 evaluates the effect of prefilling on other baselines. As TrajectoryAttention (Xiao et al., 2025) and DiffusionAsShader (Gu et al., 2025) do not directly operate on rendered RGB images, our evaluation focuses on TrajectoryCrafter (YU et al., 2025) and NVS-Solver (You et al., 2025) using the DAVIS dataset. An interesting contrast emerges: prefilling slightly impairs TrajectoryCrafter (except for faithfulness scores) yet benefits NVS-Solver. The impairment to TrajectoryCrafter can be attributed to the train-test domain gap. In contrast, NVS-Solver, being a zero-shot method like ours, is immune to this issue. Nevertheless, its failure to consistently enforce faithfulness to the input results in scores lower than those of our method.

## K  ADDITIONAL QUALITATIVE COMPARISON

Figs. 13, 14, 15, and 16 show additional qualitative comparison results. The common failure cases observed in previous methods include (i) unintended foreground object motion, (ii) color shifts, (iii) texture washout, and (iv) background motion inconsistency between rendered images and generated frames. Trajectory Attention (Xiao et al., 2025) tends to exhibit (iii), whereas NVS-Solver (You et al., 2025) and DiffusionAsShader (Gu et al., 2025) are likely to suffer from (iv). In addition, (ii) is often prominent in NVS-Solver. Although Trajectory Crafter (YU et al., 2025) is less susceptible to these failure cases, (i) can still occasionally be observable, especially when the input image contains animals. Our training-free method, which is designed to explicitly maintain faithfulness, effectively overcomes these issues and achieves competitive perceptual fidelity comparable to Trajectory Crafter.

| Rendered w/o Prefill | Result w/o Prefill | Rendered w/ Prefill | Result w/ Prefill |

Figure 12: The benefit of applying the prefilling operation. Without it, the gray border is present in the final result.

Table 10: Quantitative comparison by applying prefilling on different baselines with the DAVIS dataset. (+) applies prefilling, whereas (-) doesn't. KID' indicates KID$\times 10^3$.

| Method | Input-Faithfulness | | | Fidelity ↓ | | | Camera Pose ↓ | | | TSED ↑ | |
|---|---|---|---|---|---|---|---|---|---|---|---|
| | PSNR↑ | SSIM↑ | LPIPS↓ | FID | KID' | FVD | ATE | RRE | RTE | @.25 | @.50 |
| - Traj. Crafter | 24.24 | 0.811 | 0.119 | 32.03 | 1.906 | 704.7 | 0.139 | 0.047 | 0.071 | 0.499 | 0.958 |
| + Traj. Crafter | 24.34 | 0.812 | 0.117 | 31.75 | 1.914 | 706.7 | 0.144 | 0.048 | 0.075 | 0.498 | 0.957 |
| - NVS-Solver | 21.78 | 0.695 | 0.209 | 27.92 | 1.108 | 665.9 | 0.909 | 0.200 | 0.394 | 0.326 | 0.715 |
| + NVS-Solver | 23.16 | 0.729 | 0.185 | 27.87 | 1.017 | 698.3 | 0.480 | 0.140 | 0.250 | 0.427 | 0.808 |
| + Ours | 29.58 | 0.864 | 0.074 | 28.14 | 0.816 | 705.2 | 0.051 | 0.022 | 0.027 | 0.672 | 0.964 |

## L  ADDITIONAL EXPERIMENTS WITH DYCHECK DATASET

Our faithfulness evaluation is primarily performed on the non-void (valid) regions of the rendering. Specifically, we compute pixel-wise metrics between the generated images and the rendered images only at pixels where the rendering is valid (i.e., non-black). However, for the sake of comparison with prior work, it will be desirable to evaluate the generated images against complete ground-truth images over the entire image domain. This is feasible using datasets with multiple time-synchronized cameras, such as DyCheck (Gao et al., 2022). Nevertheless, we argue that this evaluation protocol is inherently flawed for two reasons:

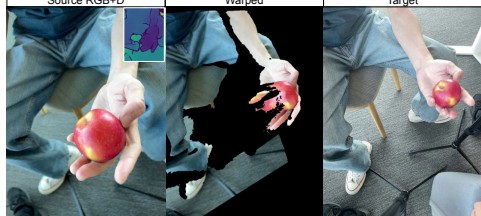

Figure 17: Visual comparison between the warped source image and the ground-truth target image ("apple" scene in DyCheck), highlighting misalignment of object edges and color shifts.

1. **Misalignment between ground-truth and rendered images**: Figure 17 compares the result of warping the source image to a target camera view (using the source's LiDAR depth) against the actual image captured by the target camera. We observe significant spatial misalignment and appearance shifts. Since our task is to faithfully complete the missing pixels in the warped image, even a perfectly inpainted image would not strictly match the ground truth due to these geometric and photometric errors.

2. **Entanglement of faithfulness and fidelity**: Evaluating the entire image without differentiating valid/invalid regions leads to the conflation of two distinct objectives: the valid rendered region should be preserved (faithfulness), while the invalid region should be filled with realistic, plausible textures (fidelity). Furthermore, there are an infinite number of valid ways to fill the invalid regions. A perfectly realistic inpainting could be penalized simply for deviating from the specific ground-truth texture, which is unreasonable.

Despite these limitations, we conduct this evaluation to facilitate a fair comparison with existing methods. We report the scores on the entire image using the DyCheck dataset. Following TrajectoryCrafter (YU et al., 2025), we use five scenes: *'apple', 'block', 'teddy', 'paper-windmill', and 'spin'*. We treat the first handheld camera as the source and the first fixed camera as the target.

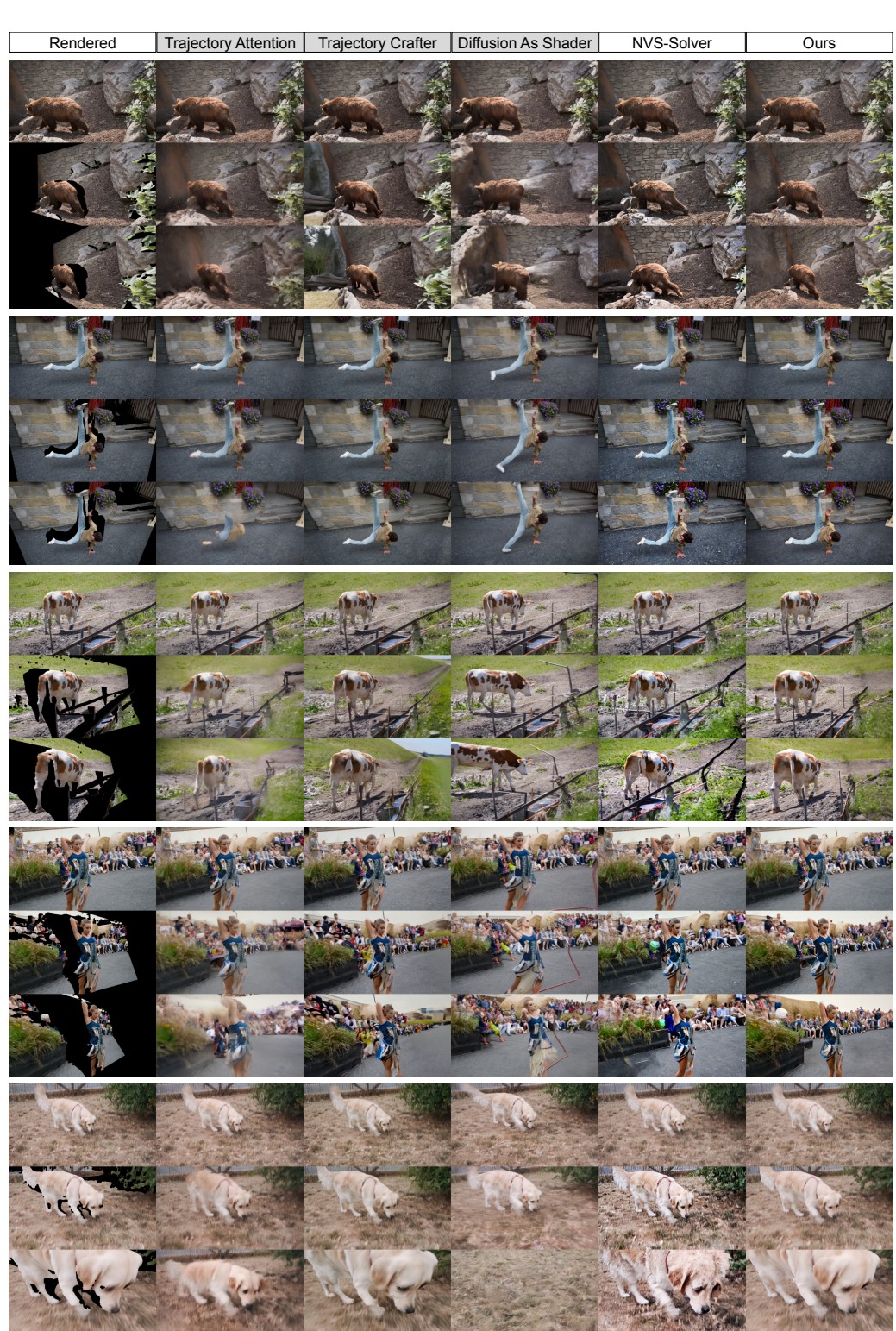

Figure 13: Additional qualitative comparison on DAVIS.

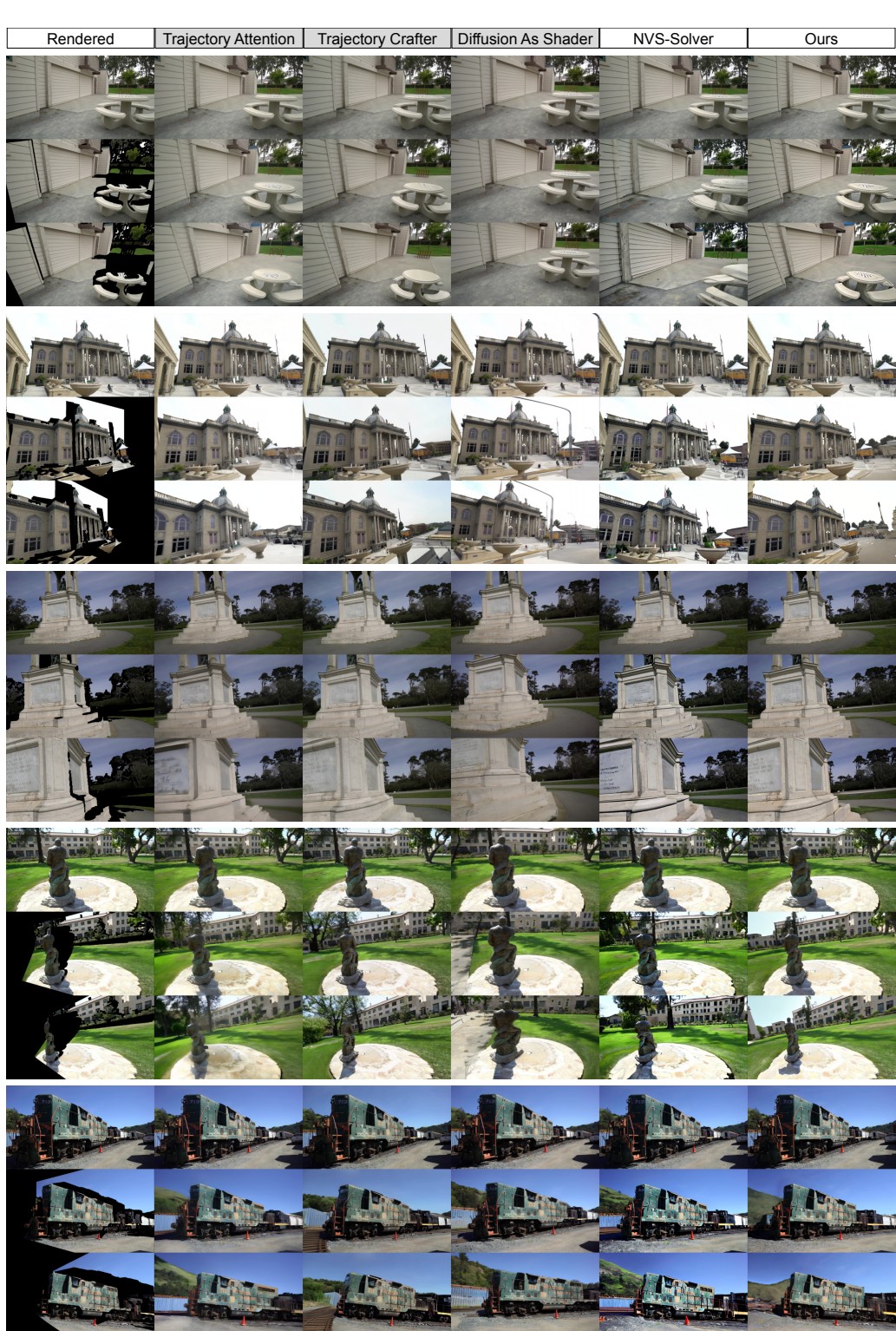

Figure 14: Additional qualitative comparison on Tanks and Temples.

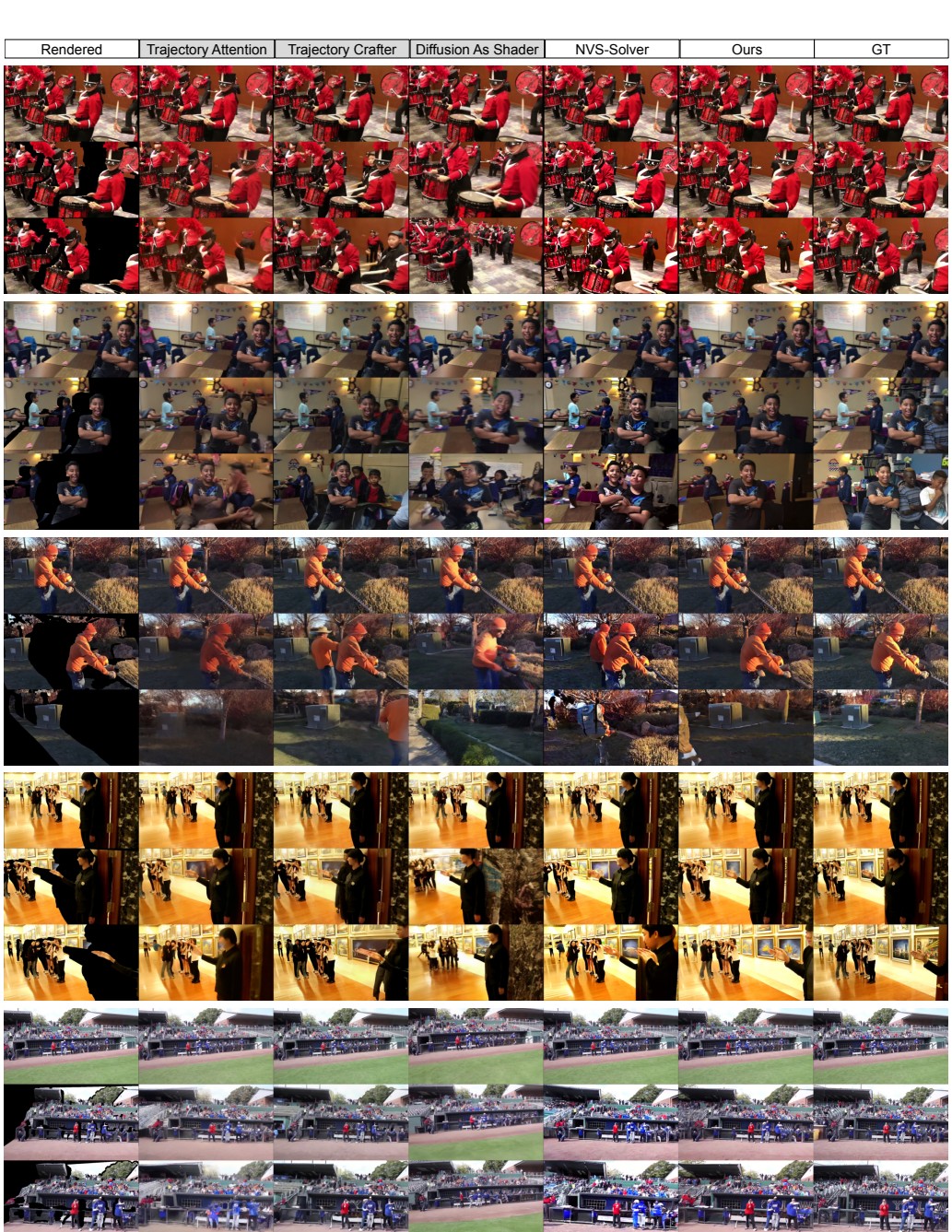

Figure 15: Additional qualitative comparison on Mannequin Challenge.

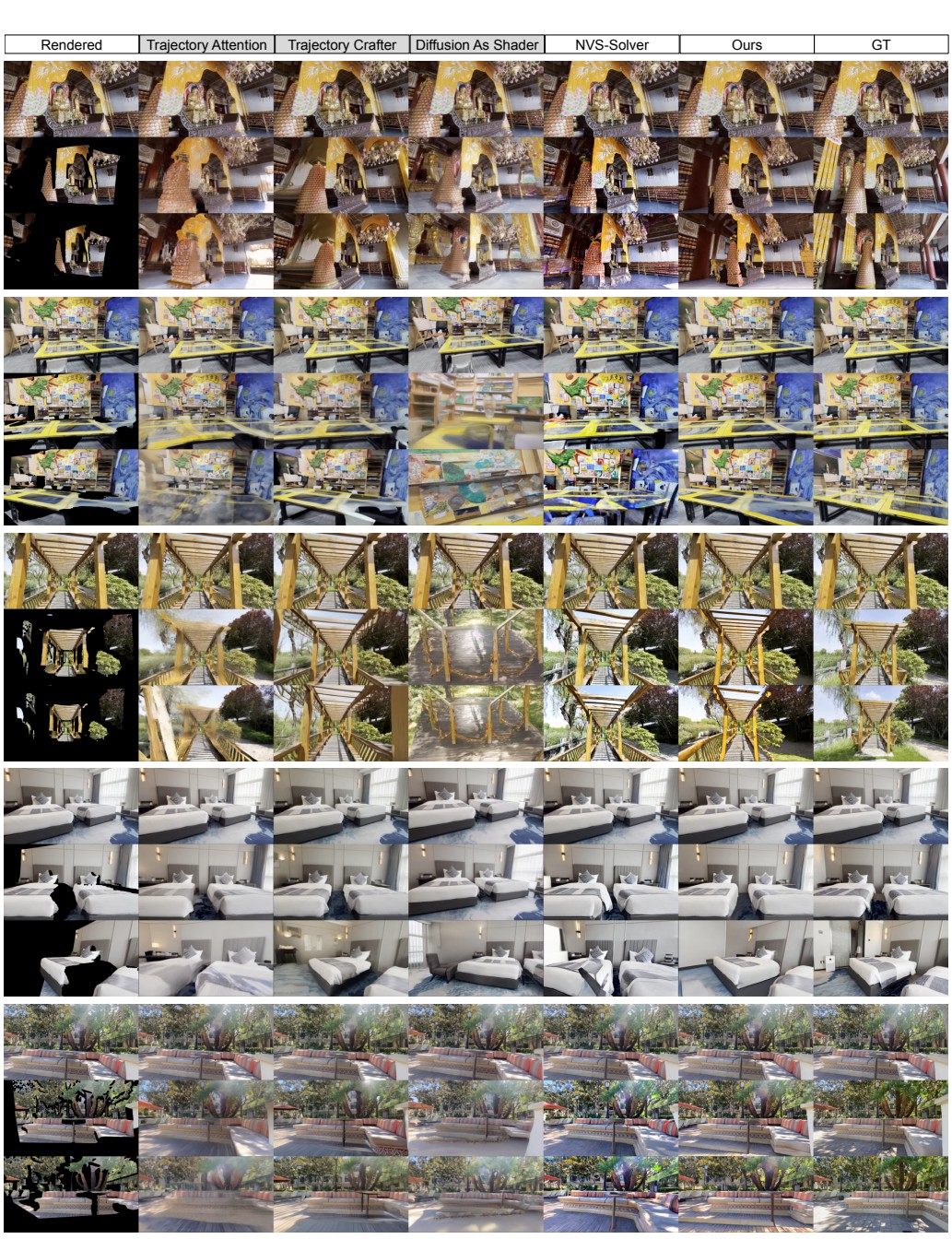

Figure 16: Additional qualitative comparison on DL3DV-Evaluation.

Table 11: Quantitative evaluation on the "full" image domain using the DyCheck iPhone dataset. Gray rows denote training-based methods.

| Method | PSNR ↑ | | | | | | SSIM ↑ | | | | | | LPIPS ↓ | | | | | |
|---|---|---|---|---|---|---|---|---|---|---|---|---|---|---|---|---|---|---|
| | Apple | Block | Paper | Spin | Teddy | **Mean** | Apple | Block | Paper | Spin | Teddy | **Mean** | Apple | Block | Paper | Spin | Teddy | **Mean** |
| Traj. Attention | 10.78 | 10.73 | 12.60 | 11.70 | 9.556 | 10.90 | 0.205 | 0.422 | 0.219 | 0.286 | 0.234 | 0.285 | 0.920 | 0.750 | 0.576 | 0.720 | 0.783 | 0.751 |
| Traj. Crafter | 11.96 | 14.76 | 16.83 | 15.16 | 13.49 | **14.42** | 0.234 | 0.522 | 0.382 | 0.334 | 0.372 | **0.382** | 0.820 | 0.473 | 0.396 | 0.499 | 0.595 | **0.547** |
| NVS-Solver | 11.10 | 13.54 | 14.56 | 13.00 | 12.02 | 12.82 | 0.197 | 0.476 | 0.267 | 0.237 | 0.301 | 0.310 | 0.838 | 0.481 | 0.390 | 0.556 | 0.620 | 0.571 |
| Ours | 12.25 | 14.10 | 16.79 | 14.90 | 13.46 | 14.23 | 0.240 | 0.524 | 0.375 | 0.321 | 0.365 | 0.377 | 0.833 | 0.500 | 0.390 | 0.499 | 0.616 | 0.561 |

Specifically, we sample every tenth frame for evaluation. For each source-target pair, we first apply VGGT (Wang et al., 2025a) to obtain depth maps[4] and camera parameters. During this process, the images are center-cropped and resized to $518 \times 518$. Then, we construct a camera trajectory from the source to the target pose using Spherical Linear Interpolation (SLERP) for rotation and Linear Interpolation (LERP) for translation. Warped images are rendered along this trajectory, resized to the model's input resolution, and fed into the model to inpaint void regions. The outputs are then resized back to $518 \times 518$. Finally, the last frame of the generated video is compared with the ground-truth target image.

Table 11 presents the quantitative results. Despite the inherent limitations of this evaluation protocol, our method demonstrates performance highly competitive with the state-of-the-art training-based method, TrajectoryCrafter. Considering our superior faithfulness demonstrated in the main paper, this score gap here is primarily driven by the fidelity in void regions, where TrajectoryCrafter benefits from its domain-specific training.

## M    COMPARISON WITH OTHER BASELINES

In the main paper, we prioritized comparisons with other methods that share the "render-and-inpaint" paradigm and can be fairly evaluated under a unified depth estimation backbone. Here, we provide additional comparisons with two relevant works: **Invisible Stitch** (Engstler et al., 2025) and **Stable Virtual Camera (SEVA)** (Zhou et al., 2025). We excluded these methods from the main experimental results because their fundamental methodological differences hinder a strictly fair quantitative comparison. We detail the specific reasons below, followed by the reference results.

**Invisible Stitch.**    Although Invisible Stitch falls into the category of 3D-aware generation, several factors make a direct comparison problematic:

- **Iterative Error Accumulation:** Unlike our video inpainting approach, Invisible Stitch relies on a recursive loop of rendering, inpainting, and unprojecting on a per-frame basis. This iterative nature is highly prone to error accumulation, in which minor artifacts in early inpainted frames are permanently baked into the 3D representation, progressively degrading the quality of subsequent frames.

- **Heuristic Rendering and Optimization:** The method's performance heavily depends on heuristic parameters for point cloud rendering (e.g., point size determination) and requires complex hyperparameter tuning for keyframe selection for inpainting and 3DGS optimization. These heuristics introduce ambiguity, complicating the establishment of a standardized evaluation setting.

- **Backbone Incompatibility:** The original method relies on a specific fine-tuned ZoeDepth (Bhat et al., 2023) model for depth inpainting. Substituting this with our standardized backbone (Depth Anything V2) and using external depth inpainting models, e.g., Prior Depth Anything (Wang et al., 2025b), creates a domain gap that inevitably penalizes its performance, making it difficult to isolate the method's true capability from the backbone's influence.

- **Task Mismatch:** Designed primarily for panoramic expansion, the method often struggles with the large disocclusions and parallax effects typical in our forward-facing camera motion benchmarks.

---

[4]We found that the depth maps from VGGT warp the source image to the target view more accurately than the provided LiDAR depth.

Table 12: Reference quantitative comparison with Invisible Stitch and Stable Virtual Camera on Mannequin Challenge (Real Camera Motion). Note that Input-Faithfulness does not apply (N/A) to Stable Virtual Camera, as it does not utilize rendered images as guidance.

| Method | Input-Faithfulness | | | GT-Faithfulness | | | Fidelity ↓ | |
|---|---|---|---|---|---|---|---|---|
| | PSNR↑ | SSIM↑ | LPIPS↓ | PSNR↑ | SSIM↑ | LPIPS↓ | FID | KID' |
| Invisible Stitch | 23.18 | 0.804 | 0.215 | 18.63 | 0.633 | 0.306 | 62.89 | 13.79 |
| Stable Virtual Camera | N/A | N/A | N/A | 13.69 | 0.497 | 0.393 | 33.59 | 4.331 |
| Ours | **30.51** | **0.925** | **0.056** | **19.03** | **0.663** | **0.226** | **26.18** | **1.865** |

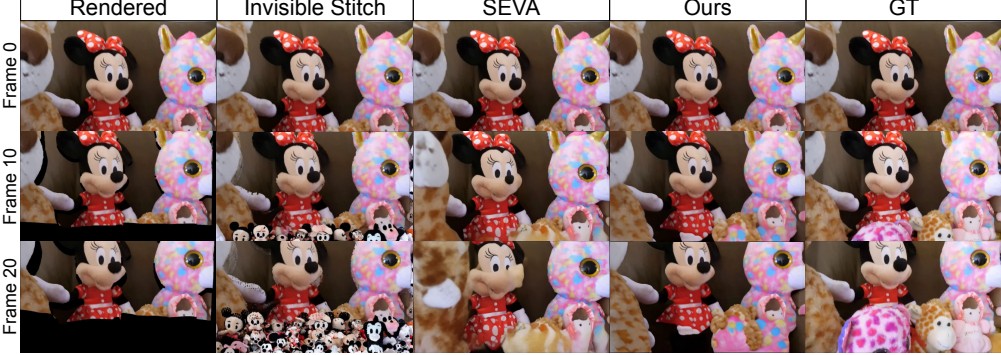

Figure 18: Qualitative results of Invisible Stitch and Stable Virtual Camera (SEVA). Invisible Stitch suffers from error propagation, while SEVA exhibits camera drift (see Minnie's eyebrow position) and structural degradation due to domain shift.

**Stable Virtual Camera (SEVA).** Comparison with SEVA is structurally challenging due to its lack of explicit 3D guidance:

- **Inapplicability of Input-Faithfulness:** Since SEVA generates novel views without explicit 3D conditioning (i.e., it does not use rendered images as input), our primary metric, *Input-Faithfulness*, which measures adherence to the geometric guidance, is structurally inapplicable. While *GT-Faithfulness* could technically be computed, comparing a geometry-free method against geometry-guided ones on this metric yields little meaningful insight, especially since our method is designed to improve Input-Faithfulness.

- **Scale Ambiguity and Drifting:** In the single-image setting without explicit 3D priors, SEVA suffers from severe scale ambiguity. We observed that the model frequently misinterprets the scene scale, resulting in excessive camera movement relative to the target trajectory. To mitigate this, we followed the authors' recommendation to manually rescale the camera trajectory (e.g., by a factor of 0.1) during evaluation. This manual intervention fundamentally makes a fair comparison difficult.

- **Domain Gap:** SEVA is trained primarily on static scenes. Consequently, it exhibits significant performance degradation when applied to our datasets containing humans, such as Mannequin Challenge, often failing to preserve the structure of foreground subjects.

Due to these limitations, the quantitative and qualitative results presented in this section should be interpreted as reference values rather than a direct competitive benchmark.

For Invisible Stitch, we select every 5 frames as keyframes for RGBD inpainting, and skip the 3DGS training part to avoid complications. Therefore, the evaluation is conducted only on these keyframes. Since the frame interval changes, which can negatively affect camera pose and geometry scores, we limit evaluation to per-frame faithfulness and frame-level fidelity for fairness. The RGBD inpainting part consists of Stable Diffusion 2 (Blattmann et al., 2023) and Prior Depth Anything (Wang et al., 2025b). For Stable Virtual Camera, we follow the original SEVA paper and test the camera scales [0.1, 0.2, ..., 1.0], and report the result of scale 0.1 because it recorded the best scores.

As shown in Table 12 and Figure 18, Invisible Stitch struggles to generate spatiotemporally coherent videos, as reflected in its poor fidelity scores. Conversely, SEVA's low GT-Faithfulness scores

highlight the difficulty of achieving precise camera motion control from a single image without explicit 3D guidance. These results underscore the critical role of explicit rendering guidance and non-iterative generation for robust zero-shot novel view synthesis.

# N   WAN2.2 AS THE BASE VIDEO DIFFUSION MODEL

Our proposed homography deformation and SA-RePaint both manipulate latents directly with the denoiser treated intact. Therefore, our method naturally extends to more recent DiT-based architectures (Peebles & Xie, 2023). This section demonstrates the extensibility of our method by replacing SVD with the latest Wan2.2-TI2V-5B model (Wan et al., 2025).

## N.1   PRELIMINARIES OF FLOW MATCHING

Wan2.2 differs from SVD in that it is built on the flow matching framework: given a clean video latent $z_0$ and a continuous time $t \in [0, 1]$, the forward noising process is defined by:

$$z_t = (1 - t)z_0 + t\,\epsilon, \quad \epsilon \sim \mathcal{N}(0, I) \tag{42}$$

The model is tasked to predict the flow $v := \epsilon - x_0$ from the noisy input $z_t$, timestep $t$, and conditioning signals of the first frame and the text prompt describing the video. Let $v_\theta^{(t)}(z_t)$ be the predicted flow. During inference, we start from a pure Gaussian noise $z_1 \sim \mathcal{N}(0, I)$ and gradually denoise it with the predefined timestep sequence $1 = t_N > t_{N-1} > \cdots > t_1 > t_0 = 0$ following the Euler update:

$$\hat{z}_{0|t_i} = z_{t_i} - t_i \cdot v_\theta^{(t_i)}(z_{t_i}) \tag{43}$$

$$z_{t_{i-1}} = \hat{z}_{0|t_i} + t_{i-1} \cdot v_\theta^{(t_i)}(z_{t_i}) \tag{44}$$

## N.2   REFORMULATION OF SA-REPAINT FOR FLOW MATCHING

Due to the fundamental mathematical difference, we need a slight modification in our SA-RePaint process. Let $y$ be the latent tensor of rendered images, $m^{\mathrm{valid}}$ be the mask tensor representing the valid region of $y$, and $z_t$ be the current noisy latent under generation. Recapitulating Section 3.2, SA-RePaint consists of three steps:

**(i) Noising**   Both $y$ and $\hat{z}_{0|t}$ are noised to a certain level $s \in [0, t]$ so that the resulting latents $y_s$ and $\hat{z}_{s|t}$ seamlessly blends. Based on Eq. 42,

$$y_s := (1 - s)y + s\,\epsilon, \quad \epsilon \sim \mathcal{N}(0, I) \tag{45}$$

$$\hat{z}_{s|t} := \left(1 - \frac{s}{t}\right)\hat{z}_{0|t} + \frac{s}{t}z_t \tag{46}$$

**(ii) Merging**   This is a simple mask-based blending:

$$\hat{z}_{s|t}^{\mathrm{merged}} = m^{\mathrm{valid}}\, y_s + \left(1 - m^{\mathrm{valid}}\right)\hat{z}_{s|t} \tag{47}$$

**(iii) Renoising**   The merged latent $\hat{z}_{s|t}^{\mathrm{merged}}$ is noised back to the level of timestep $t$. The actual noise strength to add needs derivation specifically for the flow matching formulation:

**Lemma 2.** *Let $z_0$ be the clean latent tensor. Let $s$ and $t$ be real numbers with $0 < s < t$. Under the flow matching (Lipman et al., 2022) framework, the operation* `add_noise`$(\cdot, s \to t)$ *to add an i.i.d. Gaussian noise on $z_s \sim \mathcal{N}\left((1 - s)z_0, s^2 I\right)$ to generate a new random variable $z_t \sim \mathcal{N}\left((1 - t)z_0, t^2 I\right)$ is given by*

$$z_t = \mathtt{add\_noise}(z_s, s \to t) := \frac{1 - t}{1 - s}z_s + \sqrt{t^2 - s^2\left(\frac{1 - t}{1 - s}\right)^2}\,\epsilon, \quad \epsilon \sim \mathcal{N}(0, I). \tag{48}$$

*Proof.* Since $z_t$ is an affine transformation of independent Gaussian variables $z_s$ and $\epsilon$, it also follows a Gaussian distribution. Therefore, the proof concludes by showing the mean and the variance of the resulting Gaussian distribution. Let $A := \frac{1-t}{1-s}$. Then the means is:

$$E[z_t] = E\left[A\, z_s + \sqrt{t^2 - s^2 A^2}\, \epsilon\right] = A\, E[z_s] + \sqrt{t^2 - s^2 A^2}\, E[\epsilon] = A\,(1-s)\, E[z_0] + 0 = (1-t)\, E[z_0]. \tag{49}$$

Noting that $z_s$ and $\epsilon$ are independent, the variance is:

$$\mathrm{Var}[z_t] = \mathrm{Var}\left[A\, z_s + \sqrt{t^2 - s^2 A^2}\, \epsilon\right] \tag{50}$$

$$= A^2\, \mathrm{Var}[z_s] + (t^2 - s^2 A^2)\, \mathrm{Var}[\epsilon] \tag{51}$$

$$= A^2 s^2 I + (t^2 - s^2 A^2)I = t^2 I. \tag{52}$$

Therefore, we conclude that $z_t \sim \mathcal{N}((1-t)z_0, t^2 I)$. $\qquad\square$

### N.3 DEDUCE THE PER-PIXEL NOISE LEVEL

Similarly to Section 4.2.1, we aim to find appropriate $0 \le s \le t$ locally so that $y_s$ and $\hat{z}_{s|t}$ blend seamlessly along their borders. Let $p$ be a pixel position where $y$ is valid, and we temporarily focus on the local window around $p$. We define the optimal $s$ on this window as:

$$s^* := \underset{0 \le s \le t}{\arg\min} \left\|\mathrm{Var}[\hat{z}_{s|t}] - \mathrm{Var}[y_s]\right\|_1. \tag{53}$$

Because $p$ is in the valid region of $y$, the ground-truth clean latent $z_0$ should satisfy $z_0 = y$ within this local window. Therefore, we can rewrite the above equation as:

$$s^* := \underset{0 \le s \le t}{\arg\min} \left\|\mathrm{Var}[\hat{z}_{s|t}] - \mathrm{Var}[z_s]\right\|_1 \tag{54}$$

where $z_s := (1-s)\, z_0 + s\, \epsilon$.

**Theorem 2.** *The optimal $s^*$ satisfying Eq. 54 is an element of the set:*

$$s^* \in \{\eta_+, \eta_-, 0, t\}, \quad \text{where } \eta_\pm := \mathrm{clip}\left(\frac{-B \pm \sqrt{\max(0, B^2 - AC)}}{A}; 0, t\right). \tag{55}$$

*Here, $A$, $B$, and $C$ are given by the following:*

$$A = \mathrm{Var}[v_\theta^{(t)}(z_t)] - \mathrm{Var}[z_0] - 1 \tag{56}$$

$$B = \mathrm{Cov}[\hat{z}_{0|t}, v_\theta^{(t)}(z_t)] + \mathrm{Var}[z_0] \tag{57}$$

$$C = \mathrm{Var}[\hat{z}_{0|t}] - \mathrm{Var}[z_0] \tag{58}$$

*Proof.* From the definition of $z_s$ and $\hat{z}_{s|t}$, we have

$$\mathrm{Var}[z_s] = \mathrm{Var}\left[(1-s)\, z_0 + s\, \epsilon\right] \tag{59}$$

$$= (1-s)^2\, \mathrm{Var}[z_0] + s^2 \tag{60}$$

$$\mathrm{Var}[\hat{z}_{s|t}] = \mathrm{Var}[(1 - s/t)\, \hat{z}_{0|t} + (s/t)\, z_t] \tag{61}$$

$$= \mathrm{Var}\left[\hat{z}_{0|t} + s\, v_\theta^{(t)}(z_t)\right] \tag{62}$$

$$= \mathrm{Var}\left[\hat{z}_{0|t}\right] + 2s\, \mathrm{Cov}\left[\hat{z}_{0|t}, v_\theta^{(t)}(z_t)\right] + s^2\, \mathrm{Var}\left[v_\theta^{(t)}(z_t)\right] \tag{63}$$

Therefore, the objective function is rewritten as $\left\|\mathrm{Var}[\hat{z}_{s|t}] - \mathrm{Var}[z_s]\right\|_1 = \|As^2 + 2Bs + C\|_1$, where

$$A = \mathrm{Var}[v_\theta^{(t)}(z_t)] - \mathrm{Var}[z_0] - 1 \tag{64}$$

$$B = \mathrm{Cov}[\hat{z}_{0|t}, v_\theta^{(t)}(z_t)] + \mathrm{Var}[z_0] \tag{65}$$

$$C = \mathrm{Var}[\hat{z}_{0|t}] - \mathrm{Var}[z_0] \tag{66}$$

The rest is the same as the proof of Theorem 1 except that the representations of coefficients $A$, $B$, and $C$ have changed. Therefore, we can directly refer to its solution with the coefficients replaced by the above. $\qquad\square$

### N.4 IMPLEMENTATION DETAILS

We use WAN2.2-TI2V-5B as our base model due to its affordable inference cost. Since it is fundamentally different from Stable Video Diffusion in terms of theoretical framework, model architecture, and inference capability, we have altered several implementation details as listed below:

- The input image size is $1280 \times 704$, and we generate 25 frames to match our SVD baseline.
- Wan2.2's VAE compression rate is $4 \times 16 \times 16$; except for the first frame's independent encoding, every 4-frame chunk is encoded into a single latent frame with $16\times$ spatial size reduction. We accordingly area-resize the rendering mask to this size. Although Wan2.2 VAE uses the previous chunk's information to condition the current frame chunk's encoding, we found that our per-chunk independent resizing performs relatively well.
- We empirically found that WAN2.2's intermediate outputs become sharp and clear much faster than SVD. Therefore, we halt homography deformation at step 10 out of 50 total denoising steps.
- The attention q/k tensors for $\text{Var}[z_0]$ estimation (Sec. 4.2.2) are extracted from the 15th DiT self attention block.
- We remove attention key weighting (Sec. 4.2.3) because its inclusion leads to blurry inpainting results. This does not undermine our variance transfer logic, since we can naturally expect similar texture generation around the valid/invalid borders even without attention key weighting.
- Also, we remove SEG (Hong, 2024) because its integration led to worse results.
- Similar to our SVD baseline, we apply free generation without any intervention after step 40 for smooth blending between valid and invalid (inpainted) regions.

### N.5 EVALUATION

Tables 13, 14, 15, and 16 show drastic improvements from our SVD baseline. Especially pronounced is the much higher and diverse inpainting fidelity without compromising the faithfulness to the rendered areas, as evidenced by the competitive VBench scores against Trajectory Crafter. We can see these superior traits also in Fig. 19.

However, we have identified several limitations of our current method when applied to Wan2.2: (1) Dynamic Motion Prior: Although our method strictly enforces consistency with the rendered images, the inpainted regions are completely up to the model's prior. Since Wan2.2 tries to inject dynamics in its generated videos, we sometimes observe unintended motion in the inpainted areas, such as water splashes. (2) Drifting Synthesis: We observed cases where drifting synthesis occurs even when homography deformation is applied. We hypothesize that this is due to Wan2.2's stronger generative capability, which can produce diverse scenes including mildly unrealistic video effects, possibly making drifting synthesis a plausible output. Additionally, as Wan2.2 is a flow-based model, the global structure of the video is determined in the earlier denoising stages than diffusion-based counterparts, so iterative homography compensation may not fully take effect. (3) Stripe Artifacts: In some cases, grid-pattern artifacts emerge in inpainted regions. We suspect this may stem from the direct mask operation in the spatiotemporally compressed latents, but further investigation is necessary.

Nevertheless, the above results demonstrate a significant potential for our method's generalizability. Note that we have slightly modified the Wan2.2 VAE encoder so that it doesn't cache the previous frame chunk's data on the GPU for next-chunk conditioning. This change enables the entire model to run within a 24 GB memory budget, achieving our end goal of faithfulness-first, low-cost, zero-shot NVS for wider community accessibility.

### N.6 ABLATION

To verify the effectiveness of our proposed homography deformation and SA-RePaint in this Wan2.2 setting, we conduct an ablation study similar to Sec. 5.4: we discard each module from our full pipeline and observe the metric shifts. Table 17 shows similar numerical changes as in Table 3, corroborating that our proposed modules are functioning as expected. More specifically, removing

Table 13: Quantitative comparison on DAVIS (Scripted Camera Motion). Top: Comparison on standard metrics. Bottom: VBench evaluation.

| Method | Input-Faithfulness | | | Fidelity ↓ | | | Camera Pose ↓ | | | Geometry | | |
|---|---|---|---|---|---|---|---|---|---|---|---|---|
| | PSNR↑ | SSIM↑ | LPIPS↓ | FID | KID' | FVD | ATE | RRE | RTE | T.25↑ | T.50↑ | M3R↓ |
| Traj. Crafter | 24.24 | 0.811 | 0.119 | 32.03 | 1.906 | 704.7 | 0.139 | 0.047 | 0.071 | 0.499 | 0.958 | **0.032** |
| Ours (SVD) | 29.58 | 0.864 | 0.074 | 28.14 | 0.816 | 705.2 | **0.051** | 0.022 | **0.027** | 0.672 | **0.964** | 0.033 |
| Ours (Wan2.2) | **32.67** | **0.942** | **0.054** | **26.35** | **0.486** | 699.2 | 0.066 | **0.021** | 0.032 | **0.856** | 0.961 | 0.033 |

| Method | Subject Consis.↑ | Background Consis.↑ | Temporal Flicker↑ | Motion Smooth.↑ | Overall Consis.↑ | Aesthetic Quality↑ | Imaging Quality↑ |
|---|---|---|---|---|---|---|---|
| Traj. Crafter | 94.92 | 95.28 | **96.20** | 99.05 | **24.44** | 51.93 | **65.97** |
| Ours (SVD) | **95.33** | 94.92 | 96.03 | 99.05 | 24.07 | 52.35 | 65.39 |
| Ours (Wan2.2) | 95.25 | **95.69** | 96.06 | **99.14** | 24.20 | **52.70** | 65.65 |

Table 14: Quantitative comparison on Tanks and Temples (Scripted Camera Motion). Top: Comparison on standard metrics. Bottom: VBench evaluation.

| Method | Input-Faithfulness | | | Fidelity ↓ | | | Camera Pose ↓ | | | Geometry | | |
|---|---|---|---|---|---|---|---|---|---|---|---|---|
| | PSNR↑ | SSIM↑ | LPIPS↓ | FID | KID' | FVD | ATE | RRE | RTE | T.25↑ | T.50↑ | M3R↓ |
| Traj. Crafter | 23.99 | 0.798 | 0.109 | 21.57 | 1.884 | 1146 | 0.126 | 0.050 | 0.077 | 0.446 | **0.971** | **0.027** |
| Ours (SVD) | 28.98 | 0.872 | 0.063 | 19.95 | 0.941 | 1025 | **0.058** | **0.022** | **0.029** | 0.641 | 0.967 | **0.027** |
| Ours (Wan2.2) | **31.54** | **0.937** | **0.046** | **18.52** | **0.626** | 1012 | 0.063 | 0.023 | 0.033 | **0.878** | 0.962 | **0.027** |

| Method | Subject Consis.↑ | Background Consis.↑ | Temporal Flicker↑ | Motion Smooth.↑ | Overall Consis.↑ | Aesthetic Quality↑ | Imaging Quality↑ |
|---|---|---|---|---|---|---|---|
| Traj. Crafter | 95.84 | 95.61 | **96.01** | 99.00 | **24.20** | 54.17 | 73.35 |
| Ours (SVD) | **96.05** | 95.06 | 95.90 | 99.04 | 23.74 | 54.87 | 73.33 |
| Ours (Wan2.2) | 95.92 | **95.90** | 95.89 | **99.11** | 23.87 | **55.74** | **74.19** |

Table 15: Quantitative comparison on Mannequin Challenge (Real Camera Motion). Top: Comparison on standard metrics. Bottom: VBench evaluation.

| Method | Input-Faithfulness | | | GT-Faithfulness | | | Fidelity ↓ | | | Camera Pose ↓ | | | Geometry |
|---|---|---|---|---|---|---|---|---|---|---|---|---|---|
| | PSNR↑ | SSIM↑ | LPIPS↓ | PSNR↑ | SSIM↑ | LPIPS↓ | FID | KID' | FVD | ATE | RRE | RTE | M3R↓ |
| Traj. Crafter | 24.48 | 0.855 | 0.119 | 18.96 | 0.662 | 0.243 | 30.16 | 4.302 | 221.4 | 0.196 | 0.552 | 0.092 | **0.044** |
| Ours (SVD) | 30.51 | 0.925 | 0.056 | 19.03 | 0.663 | 0.226 | 26.18 | 1.865 | 187.1 | **0.061** | 0.424 | **0.031** | 0.047 |
| Ours (Wan2.2) | **33.48** | **0.958** | **0.027** | **19.11** | **0.667** | **0.211** | **22.64** | **0.761** | 191.3 | 0.063 | 0.593 | 0.033 | 0.047 |

| Method | Subject Consis.↑ | Background Consis.↑ | Temporal Flicker↑ | Motion Smooth.↑ | Overall Consis.↑ | Aesthetic Quality↑ | Imaging Quality↑ |
|---|---|---|---|---|---|---|---|
| Traj. Crafter | 94.58 | **93.97** | **94.58** | **98.49** | **24.75** | 52.17 | 64.88 |
| Ours (SVD) | 94.43 | 93.31 | 94.07 | 98.36 | 24.17 | 53.60 | 67.48 |
| Ours (Wan2.2) | **94.72** | 93.93 | 94.05 | 98.24 | 24.61 | **54.30** | **68.19** |

Table 16: Quantitative comparison on DL3DV-Evaluation (Real Camera Motion). Top: Comparison on standard metrics. Bottom: VBench evaluation.

| Method | Input-Faithfulness | | | GT-Faithfulness | | | Fidelity ↓ | | | Camera Pose ↓ | | | Geometry |
|---|---|---|---|---|---|---|---|---|---|---|---|---|---|
| | PSNR↑ | SSIM↑ | LPIPS↓ | PSNR↑ | SSIM↑ | LPIPS↓ | FID | KID' | FVD | ATE | RRE | RTE | M3R↓ |
| Traj. Crafter | 21.05 | 0.732 | 0.209 | 18.46 | 0.627 | 0.296 | 23.17 | 7.467 | 188.6 | 0.593 | 0.990 | 0.287 | 0.071 |
| Ours (SVD) | **27.39** | 0.861 | 0.093 | **18.92** | 0.641 | 0.255 | 20.27 | 3.064 | **163.3** | **0.168** | 0.850 | 0.075 | 0.074 |
| Ours (Wan2.2) | 26.57 | **0.877** | **0.079** | 18.90 | **0.645** | **0.253** | **16.66** | **2.581** | 192.3 | 0.238 | 1.341 | 0.100 | **0.069** |

| Method | Subject Consis.↑ | Background Consis.↑ | Temporal Flicker↑ | Motion Smooth.↑ | Overall Consis.↑ | Aesthetic Quality↑ | Imaging Quality↑ |
|---|---|---|---|---|---|---|---|
| Traj. Crafter | 91.13 | 93.22 | 92.09 | **96.38** | 24.82 | 49.57 | 62.14 |
| Ours (SVD) | 90.64 | 93.01 | 91.61 | 95.96 | 24.28 | 50.01 | 65.83 |
| Ours (Wan2.2) | **91.60** | **93.90** | **92.56** | 96.16 | **24.84** | **51.15** | **67.31** |

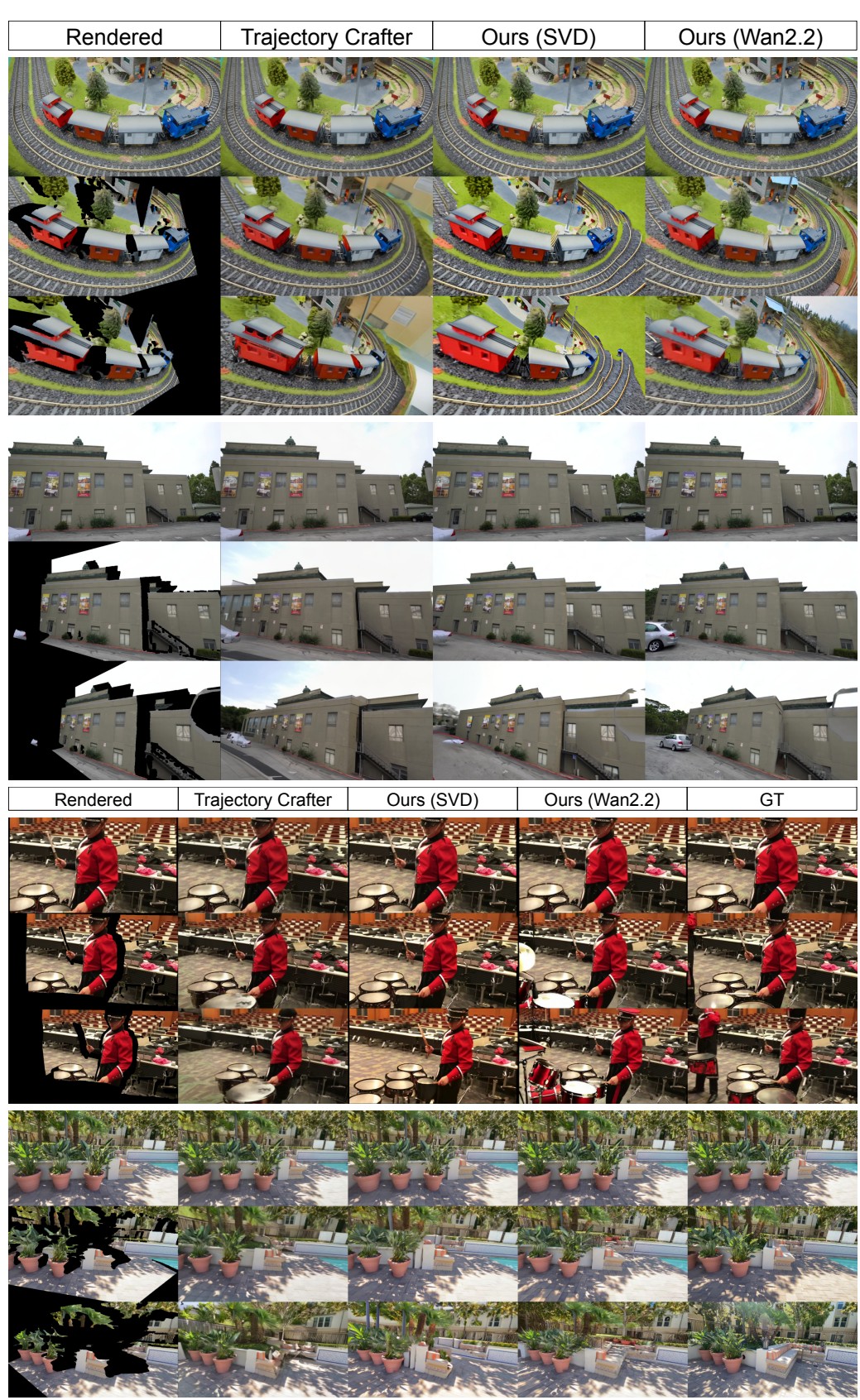

Figure 19: Qualitative comparison between Trajectory Crafter and ours (SVD-based and Wan2.2-based). Each image is from DAVIS, Tanks and Temples, Mannequin Challenge, and DL3DV-Evaluation.

Table 17: Ablation study on DAVIS with the Wan2.2 backbone. SAR: SA-RePaint; KID' denotes KID$\times 10^3$, $T_{.25}$ ($T_{.50}$) represents TSED with a threshold of 0.25 (0.50), and M3R stands for MEt3R.

| Method | Input-Faithfulness | | | Fidelity ↓ | | | Camera Pose ↓ | | | Geometry | | |
| --- | --- | --- | --- | --- | --- | --- | --- | --- | --- | --- | --- | --- |
| | PSNR↑ | SSIM↑ | LPIPS↓ | FID | KID' | FVD | ATE | RRE | RTE | $T_{.25}$↑ | $T_{.50}$↑ | M3R↓ |
| w/o Homography | 31.62 | 0.939 | 0.057 | 26.34 | 0.519 | 698.3 | 0.194 | 0.046 | 0.079 | 0.759 | 0.906 | 0.031 |
| w/o SAR ($\Sigma = 0$) | 33.01 | 0.943 | 0.054 | 27.32 | 0.710 | 704.6 | 0.098 | 0.026 | 0.038 | 0.865 | 0.967 | 0.032 |
| All | 32.67 | 0.942 | 0.054 | 26.35 | 0.486 | 699.2 | 0.066 | 0.021 | 0.032 | 0.856 | 0.961 | 0.033 |

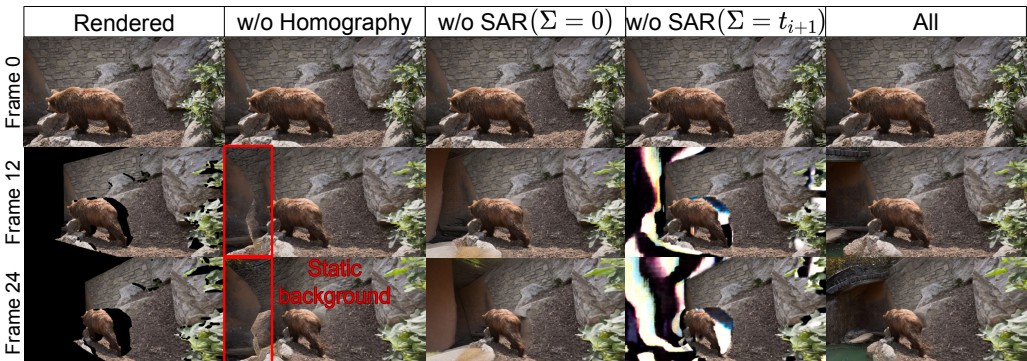

Figure 20: Ablation of our components with Wan2.2 backbone. Removing each component results in expected degradation (drifting synthesis, textureless generation) except for w/o SAR ($\Sigma = t_{i+1}$, original RePaint), which was unsuccessful in generation itself.

homography deformation leads to an obvious decline in camera pose accuracy, indicating more severe drifting synthesis. Replacing SA-Repaint with Stable Diffusion-type RePaint ($\Sigma = 0$) results in higher Input-Faithfulness at the expense of a notable fidelity drop, indicating that the inpainting quality is compromised because of the unresolved texture gap (cf. Fig. 2) during RePaint. Lastly, replacing it with the original RePaint ($\Sigma = t_{t+1}$) shows different results: it completely fails in inpainting (cf. Fig. 20). This may be attributable to the faster generation characteristics of flow matching models as briefly mentioned in Section N.4, which can be at odds with strong renoising of the original RePaint. Our SA-RePaint doesn't suffer from this collapse while achieving much higher texture fidelity than RePaint ($\Sigma = 0$).

## O  SA-RePaint for General Image Diffusion Models

Our proposed SA-RePaint is not constrained to the novel view synthesis task. To demonstrate the same generality as the original RePaint, we apply it to a general image inpainting task with Stable Diffusion 2 (SD2).

### O.1  Preliminary of DDIM

Contrary to Stable Video Diffusion (SVD) built on the EDM framework (Karras et al., 2022), Stable Diffusion 2 (SD2) inference works under the variance-preserving DDIM scheduling (Song et al., 2020). Therefore, we first reformulate SA-RePaint under this setting. Let $z_0$ be a clean latent tensor. Both EDM and DDIM predefine a decreasing timestep sequence $T = t_0 > t_1 > \cdots > t_N = 0$ where $N$ is the total number of denoise steps. However, DDIM (more precisely, its predecessor DDPM (Ho et al., 2020)) is built on a discrete Markov chain formulation, so $\{t_i\}_{i=0}^{N}$ are all integers. Instead, it introduces a real number sequence $0 < \alpha_{t_0} < \alpha_{t_1} < \cdots < \alpha_{t_N} = 1$ such that the forward noising process is defined as:

$$z_{t_i} = \sqrt{\alpha_{t_i}} z_0 + \sqrt{1 - \alpha_{t_i}}\epsilon, \quad \epsilon \sim \mathcal{N}(0, I) \tag{67}$$

The model is trained to predict $\epsilon$ from the noisy latent $z_{t_i}$ conditioned on the timestep $t_i$. Let's call the predicted noise $\epsilon_\theta^{(t_i)}(z_{t_i})$. Then, the backward denoise path is defined as follows:

$$\hat{z}_{0|t_i} = \frac{z_{t_i} - \sqrt{1 - \alpha_{t_i}} \epsilon_\theta^{(t_i)}(z_{t_i})}{\sqrt{\alpha_{t_i}}} \tag{68}$$

$$z_{t_{i+1}} = \sqrt{\alpha_{t_{i+1}}} \hat{z}_{0|t_i} + \sqrt{1 - \alpha_{t_{i+1}}} \epsilon_\theta^{(t_i)}(z_{t_i}) \tag{69}$$

where $\hat{z}_{0|i}$ is the one-step denoised result similar to what we defined in the SVD setting.

## O.2 REFORMULATION OF SA-REPAINT FOR DDIM

We now focus on a particular denoising step, so we drop the subscript $i$. Let $y$ be a latent tensor of rendered images, and $m^{\text{valid}}$ be the mask representing which part of $y$ is valid. The core idea is the same: we want to paste $y$ onto the intermediate denoised result so that the pasting border is unnoticeable. Therefore, we consider (i) noising both $y$ and $\hat{z}_{0|t}$ to a certain middle noise level, (ii) pasting them together, and (iii) further noising the merged result to the noise level at timestep $t$ (c.f. the table in Sec. 3.2 for comparison with SVD).

**(i) Noising**   Based on Eq. 67 and Eq. 69, we define the noised $y$ and $\hat{z}_{0|t}$ at timestep $0 < s \le t$ as follows:

$$y_s := \sqrt{\alpha_s} y + \sqrt{1 - \alpha_s} \epsilon, \quad \epsilon \sim \mathcal{N}(0, I) \tag{70}$$

$$\hat{z}_{s|t} := \sqrt{\alpha_s} \hat{z}_{0|t} + \sqrt{1 - \alpha_s} \epsilon_\theta^{(t)}(z_t) \tag{71}$$

Note that DDIM is based on discrete timesteps in its standard formulation, so we originally cannot take the intermediate real-valued timestep $s$. However, the main obstacle to continuous-time extension lies in the backward path, because the model is only trained with the predefined timesteps. This limitation does not apply to the forward process. The forward process is governed by $\{\alpha_t\}_t$, which is typically defined as a continuous function over the interval $[0, T]$. Since our SA-RePaint method exclusively utilizes this forward noising mechanism, we can naturally extend the formulation to continuous time by evaluating the function $\alpha_\bullet$ at any real-valued $s$.

**(ii) Merging**   This is the same as the case of EDM-based formulation:

$$\hat{z}_{s|t}^{\text{merged}} = m^{\text{valid}} y_s + (1 - m^{\text{valid}}) \hat{z}_{s|t} \tag{72}$$

**(iii) Renoising**   The merged latent $\hat{z}_{s|t}^{\text{merged}}$ is further noised back to the noise level at timestep $t$. This is achievable by substituting $\hat{z}_{s|t}^{\text{merged}}$ to $z_s$ in Lemma 3 (c.f. Lemma 1 for comparison):

**Lemma 3.** *Let $z_0$ be the clean latent tensor. Let $s$ and $t$ be real numbers with $0 < s < t$. Under the DDIM (Song et al., 2020) framework, the operation* add_noise$(\cdot, s \to t)$ *to add an i.i.d. Gaussian noise on $z_s \sim \mathcal{N}(\sqrt{\alpha_s} z_0, (1 - \alpha_s)I)$ to generate a new random variable $z_t \sim \mathcal{N}(\sqrt{\alpha_t} z_0, (1 - \alpha_t)I)$ is given by*

$$z_t = \texttt{add\_noise}(z_s, s \to t) := \sqrt{\frac{\alpha_t}{\alpha_s}} z_s + \sqrt{1 - \frac{\alpha_t}{\alpha_s}} \epsilon, \quad \epsilon \sim \mathcal{N}(0, I). \tag{73}$$

*Proof.* Since $z_t$ is an affine transformation of independent Gaussian variables $z_s$ and $\epsilon$, it also follows a Gaussian distribution. We only need to determine its mean and variance. The mean is computed as:

$$E[z_t] = E\left[\sqrt{\frac{\alpha_t}{\alpha_s}} z_s + \sqrt{1 - \frac{\alpha_t}{\alpha_s}} \epsilon\right] \tag{74}$$

$$= \sqrt{\frac{\alpha_t}{\alpha_s}} E[z_s] + \sqrt{1 - \frac{\alpha_t}{\alpha_s}} E[\epsilon] = \sqrt{\frac{\alpha_t}{\alpha_s}} (\sqrt{\alpha_s} z_0) + 0 = \sqrt{\alpha_t} z_0. \tag{75}$$

Since $z_s$ and $\epsilon$ are independent, the variance is the sum of the variances:

$$\text{Var}[z_t] = \text{Var}\left[\sqrt{\frac{\alpha_t}{\alpha_s}}z_s\right] + \text{Var}\left[\sqrt{1-\frac{\alpha_t}{\alpha_s}}\,\epsilon\right] \tag{76}$$

$$= \frac{\alpha_t}{\alpha_s}\text{Var}[z_s] + \left(1-\frac{\alpha_t}{\alpha_s}\right)\text{Var}[\epsilon] \tag{77}$$

$$= \frac{\alpha_t}{\alpha_s}(1-\alpha_s)I + \left(1-\frac{\alpha_t}{\alpha_s}\right)I = \left(\frac{\alpha_t}{\alpha_s}-\alpha_t+1-\frac{\alpha_t}{\alpha_s}\right)I = (1-\alpha_t)I. \tag{78}$$

Therefore, we have shown that $z_t \sim \mathcal{N}(\sqrt{\alpha_t}z_0, (1-\alpha_t)I)$. $\qquad\square$

### O.3   DEDUCE THE PER-PIXEL ALPHA MAP $\mathcal{A}$

Our goal is seamless blending between $y_s$ and $\hat{z}_{s|t}$ at a suitable timestep $s$. By the same reasoning as in the main paper, we want to make $\alpha_s$ dynamically adjustable and extend it to a spatial map $\mathcal{A}$.

We again adopt local pixel variance as a quantitative measure to evaluate seamless blending. Let $p$ be a pixel location on which $y$ is valid, and we temporarily focus on a local window around $p$. Then, we define the optimal $\alpha_s$ on this window as:

$$\alpha_s^* := \arg\min_{\alpha_t \leq \alpha_s \leq 1} \left\|\text{Var}[\hat{z}_{s|t}] - \text{Var}[y_s]\right\|_1. \tag{79}$$

Because $p$ is in the valid region of $y$, the ground-truth clean latent $z_0$ should satisfy $z_0 = y$ within this local window. Therefore, we can rewrite the above equation as:

$$\alpha_s^* := \arg\min_{\alpha_t \leq \alpha_s \leq 1} \left\|\text{Var}[\hat{z}_{s|t}] - \text{Var}[z_s]\right\|_1 \tag{80}$$

where $z_s := \sqrt{\alpha_s}z_0 + \sqrt{1-\alpha_s}\epsilon$.

**Theorem 3.** *The optimal $\alpha_s^*$ satisfying Eq. 80 is an element of the set: $\alpha^* \in \{x_1, x_2, x_3, \alpha_t, 1\}$, where*

$$x_{1,2} = \text{clip}\left(\frac{-(AC-2B) \pm \sqrt{\max\left(0, (AC-2B^2)^2 - C^2(A^2+4B^2)\right)}}{A^2+4B^2}; \alpha_t, 1\right), \tag{81}$$

$$x_3 = \text{clip}\left(\frac{1}{2} - \frac{\text{sign}(B)\cdot A}{2\sqrt{A^2+4B^2}}; \alpha_t, 1\right). \tag{82}$$

*Here, $A$, $B$, and $C$ are given by the following:*

$$A = \text{Var}[z_0] - \text{Var}[\hat{z}_{0|t}] + \text{Var}[\epsilon_\theta^{(t)}(z_t)] - 1 \tag{83}$$

$$B = \text{Cov}[\hat{z}_{0|t}, \epsilon_\theta^{(t)}(z_t)] \tag{84}$$

$$C = 1 - \text{Var}[\epsilon_\theta^{(t)}(z_t)] \tag{85}$$

*Proof.* Expanding the respective variance equation, we get

$$\text{Var}[z_s] = \alpha_s\text{Var}[z_0] + (1-\alpha_s) \tag{86}$$

$$\text{Var}[\hat{z}_{s|t}] = \alpha_s\text{Var}[\hat{z}_{0|t}] + (1-\alpha_s)\text{Var}[\epsilon_\theta^{(t)}(z_t)] \tag{87}$$

$$+ 2\sqrt{\alpha_s(1-\alpha_s)}\,\text{Cov}[\hat{z}_{0|t}, \epsilon_\theta^{(t)}(z_t)] \tag{88}$$

Therefore,

$$\left\|\text{Var}[\hat{z}_{s|t}] - \text{Var}[z_s]\right\|_1 = \left\|A\alpha_s + C - 2B\sqrt{\alpha_s(1-\alpha_s)}\right\|_1 \tag{89}$$

$$A = \text{Var}[z_0] - \text{Var}[\hat{z}_{0|t}] + \text{Var}[\epsilon_\theta^{(t)}(z_t)] - 1 \tag{90}$$

$$B = \text{Cov}[\hat{z}_{0|t}, \epsilon_\theta^{(t)}(z_t)] \tag{91}$$

$$C = 1 - \text{Var}[\epsilon_\theta^{(t)}(z_t)] \tag{92}$$

Here we define a functions $L(x) = \left(Ax + C - 2B\sqrt{x(1-x)}\right)^2$ and seek for $L(x)$'s minimizer in the range $[\alpha_t, 1]$. This solution is also a minimizer of Eq. 89. Since $L'(x) = 0$ if the minimizer exists in $(\alpha_t, 1)$,

$$L'(x) = 2 \underbrace{\left(Ax + C - 2B\sqrt{x(1-x)}\right)}_{=:f(x)} \cdot \underbrace{\left(A - \frac{B(1-2x)}{\sqrt{x(1-x)}}\right)}_{=:g(x)} \tag{93}$$

$$f(x) = 0 \implies x = \frac{-(AC - 2B) \pm \sqrt{(AC - 2B^2)^2 - C^2(A^2 + 4B^2)}}{A^2 + 4B^2}, \tag{94}$$

$$\text{if } \begin{cases} (AC - 2B^2)^2 - C^2(A^2 + 4B^2) \geq 0 \\ \text{sign}(Ax + C) = \text{sign}(B) \end{cases} \tag{95}$$

$$g(x) = 0 \implies x = \frac{1}{2} - \frac{\text{sign}(B) \cdot A}{2\sqrt{A^2 + 4B^2}} \tag{96}$$

Adding the possibility that we cannot find the minimizer in the open range $(\alpha_t, 1)$, we ultimately get the following solution candidates:

$$x_{1,2} = \text{clip}\left(\frac{-(AC - 2B) \pm \sqrt{\max\left(0, (AC - 2B^2)^2 - C^2(A^2 + 4B^2)\right)}}{A^2 + 4B^2}; \alpha_t, 1\right) \tag{97}$$

$$x_3 = \text{clip}\left(\frac{1}{2} - \frac{\text{sign}(B) \cdot A}{2\sqrt{A^2 + 4B^2}}; \alpha_t, 1\right) \tag{98}$$

$$x_4 = \alpha_t \tag{99}$$

$$x_5 = 1 \tag{100}$$

We select the best $x$ as the one that minimizes Eq. 89. $\qquad\qquad\square$

Note that there can be two solutions satisfying $L(x_1) = L(x_2) = 0$ at the same time. In this case, we choose

$$\alpha_s^* = \frac{-(AC - 2B) + \text{sign}(AC - 2B) \cdot \sqrt{\max\left(0, (AC - 2B^2)^2 - C^2(A^2 + 4B^2)\right)}}{A^2 + 4B^2} \tag{101}$$

as the final solution for numerical stability. Indeed, we can show similarly to the SVD case that $A = B = C = 0$ under an ideal prediction, which means that the denominator is zero. Therefore, the above solution is considered to be numerically more stable because the absolute value of the numerator is smaller than the other.

We determine $\alpha_s$ pixelwise, and finally get the 2D map $\mathcal{A}$. This is used in place of $\alpha_s$ for the noising, merging, and renoising process described in Sec. O.2.

## O.4 IMPLEMENTATION DETAILS

We use the `StableDiffusionImg2ImgPipeline` implemented by `diffusers` library. The total number of denoising steps is 50, where the actual denoising starts from the 18th step by noising the input masked image latent. We introduce SA-RePaint after the 25th step; until then, we set $\mathcal{A} = 1$ for stability.

As for the SA-RePaint specific operations, we again need access to $\text{Var}[z_0]$ for calculating the coefficient $A$. We approximate it by exploiting the $qk$-similarity. Contrary to the SVD case, we don't have access to the full image to draw the reference variance from. Therefore, we apply attention masking to refer only to valid pixels. The variance amplifier $\lambda_{\text{var}}$ (c.f. Eq. 10) is set to be $0.5$, which resulted in better scores and qualitative results than $\lambda_{\text{var}} \geq 1$.

Lastly, we found that $\alpha_s$ must be consistent across channels; otherwise, many high-frequency artifacts appeared in the final inpainted images. As a simple remedy, we average the alpha map $\mathcal{A}$ in a channel dimension.

Table 18: Quantitative results of SD2 inpainting with/without SA-RePaint.

| Method | Medium | | | | Thick | | | |
|---|---|---|---|---|---|---|---|---|
| | LPIPS ↓ | FID ↓ | CLIP-I ↑ | DINO-S ↑ | LPIPS ↓ | FID ↓ | CLIP-I ↑ | DINO-S ↑ |
| RePaint ($\alpha_s = 1$) | 0.184 | 3.229 | 0.923 | **0.894** | 0.207 | 3.735 | 0.913 | **0.879** |
| RePaint ($\alpha_s = \alpha_{t_{i+1}}$) | 0.178 | 2.819 | 0.929 | 0.890 | 0.198 | 3.143 | 0.922 | 0.877 |
| SA-RePaint ($\lambda_{var} = 1$) | 0.178 | **2.741** | 0.928 | 0.887 | 0.197 | **2.929** | 0.923 | 0.873 |
| SA-RePaint ($\lambda_{var} = 0.5$) | **0.175** | 2.752 | **0.930** | 0.891 | **0.194** | 2.967 | **0.924** | 0.878 |

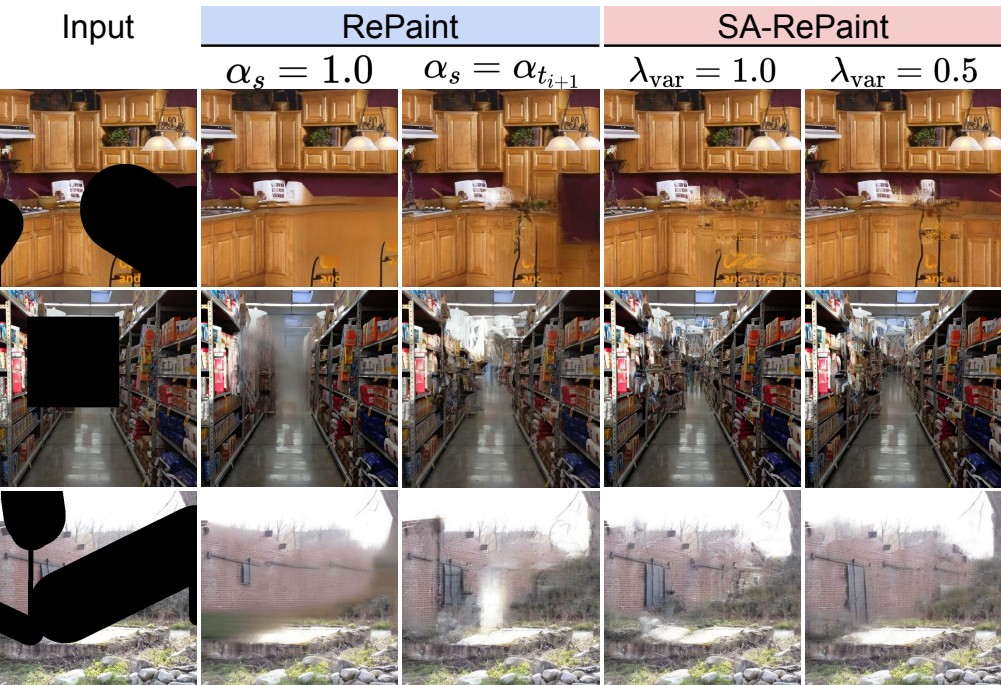

Figure 21: Qualitative comparison between RePaint and SA-RePaint under the image inpainting task with Stable Diffusion 2. Our proposed SA-RePaint maintains both the overall consistency and textural fidelity, where the latter is further controllable by $\lambda_{var}$.

## O.5 EVALUATION

We follow the evaluation scheme of LaMa (Suvorov et al., 2022). We randomly sample 30,000 images with a cropped size of 512x512 from the Places365 dataset (Zhou et al., 2017) and define two types of masks: medium and large. We utilize BLIP2 (Li et al., 2023) to generate a caption for each image, which is then fed to SD2 during image inpainting as a text prompt. We report four evaluation metrics: LPIPS, FID, CLIP Image Similarity (CLIP-I), and DINO Similarity (DINO-S).

From Table 18, we clearly observe the superiority of SA-RePaint over both conventional RePaint implementations. Furthermore, the comparison between $\lambda_{var} = 1.0$ and $0.5$ reveals a trend: a higher $\lambda_{var}$ yields better FID scores but at the cost of overall coherence (measured by LPIPS, CLIP-I, DINO-S). This finding is consistent with the observation in Sec . J.2.

As shown in Fig. 21, conventional RePaint fails to maintain a proper balance between structural consistency and textural fidelity. In contrast, our SA-RePaint achieves this balance automatically, confirming that the SA-RePaint algorithm is general and applicable to a wide range of tasks where RePaint can be employed.

Table 19: Quantitative comparison on DAVIS video inputs. Top: Comparison on standard metrics. Bottom: VBench evaluation. Gray rows are training-based methods.

| Method | Input-Faithfulness | | | Fidelity ↓ | | | Camera Pose (Median) ↓ | | |
|---|---|---|---|---|---|---|---|---|---|
| | PSNR↑ | SSIM↑ | LPIPS↓ | FID | KID ×10³ | FVD | ATE | RRE | RTE |
| Trajectory Attention | 15.48 | 0.510 | 0.452 | 41.76 | 4.789 | 733.5 | 0.468 | 0.675 | 0.204 |
| Trajectory Crafter | 22.08 | 0.743 | 0.182 | 35.90 | 4.532 | 428.2 | **0.324** | **0.517** | 0.151 |
| Diffusion As Shader | 13.84 | 0.417 | 0.486 | 37.64 | 2.756 | 599.5 | 0.941 | 0.573 | 0.582 |
| NVS-Solver | 19.42 | 0.618 | 0.280 | **27.62** | **1.281** | 419.6 | 1.058 | 0.475 | 0.454 |
| Ours | **27.51** | **0.830** | **0.106** | 29.38 | 1.455 | **380.5** | 0.377 | 0.574 | **0.149** |

| Method | Subject Consis. ↑ | Background Consis. ↑ | Temporal Flicker ↑ | Motion Smooth. ↑ | Overall Consis. ↑ | Aesthetic Quality ↑ | Imaging Quality ↑ |
|---|---|---|---|---|---|---|---|
| Trajectory Attention | **91.23** | **93.18** | **96.30** | **98.44** | 23.82 | 48.51 | 58.88 |
| Trajectory Crafter | 88.10 | 91.97 | 93.07 | 97.14 | **24.68** | 49.72 | 60.53 |
| Diffusion As Shader | 89.72 | 92.58 | 94.86 | 97.40 | 24.57 | **50.64** | 58.56 |
| NVS-Solver | 89.40 | 91.48 | 91.59 | 95.93 | 24.19 | 49.35 | **66.85** |
| Ours | 87.90 | 91.11 | 92.88 | 96.91 | 24.10 | 50.00 | 62.90 |

# P    EXPERIMENTS WITH VIDEO INPUTS

## P.1    SETTINGS

Our pipeline readily extends to video inputs with one key modification: we use VideoDepthAnything (Chen et al., 2025b) for depth estimation instead of DepthAnythingV2. This change prevents depth oscillation and texture flickering in the rendered images. Since VideoDepthAnything produces a different depth scale than DepthAnythingV2, which can result in exaggerated camera motion, we empirically halve the magnitude of the camera motion for rendering. The rendering process is similar to our single-image approach: each frame of the input video is independently unprojected into a 3D point cloud based on its estimated depth. Each resulting point cloud is then rendered from its corresponding target camera pose.

To evaluate camera pose accuracy, we employ ViPE (Huang et al., 2025) in place of GLOMAP. We chose ViPE for its robustness in estimating camera poses in scenes with dynamic objects. Specifically, we first temporally concatenate the reversed ground-truth video with the generated video. Because the generated video shares the same first frame as the ground-truth video, this creates a continuous camera path, allowing the concatenated video to be treated as a single sequential input. We then feed this video into ViPE to extract camera parameters for all frames. From these parameters, we calculate the relative camera pose between each pair of corresponding frames, one from the ground-truth and one from the generated video. Finally, we compare this calculated relative pose against the predefined camera motion used for generation.

Note that ViPE may fail to process videos if the scene is ambiguous for pose estimation (e.g., textureless or highly dynamic scenes). To mitigate the impact of such outliers, our evaluation metric is calculated as follows. For each video clip, we first compute the mean of the framewise camera pose errors. We then report the median of these per-video mean errors as our final score. The evaluation was conducted on the DAVIS dataset, using the same scenes as in our single-image experiments. We excluded the TSED metric from our evaluation, as it assumes a static scene and is therefore not applicable in this dynamic context.

## P.2    RESULTS

Figs. 22 and 23 provide a qualitative comparison between the different methods. Trajectory Attention (Xiao et al., 2025) and Diffusion As Shader (Gu et al., 2025) tend to fail in highly dynamic scenes because the tracking point map they use as auxiliary input becomes uninformative in later frames. Similar to the image-input case, NVS-Solver (You et al., 2025) struggles to align with the rendered images. In contrast, our method, despite also being a zero-shot approach like NVS-Solver, maintains significantly higher consistency with the rendered images. Furthermore, its fidelity is comparable to that of Trajectory Crafter (YU et al., 2025).

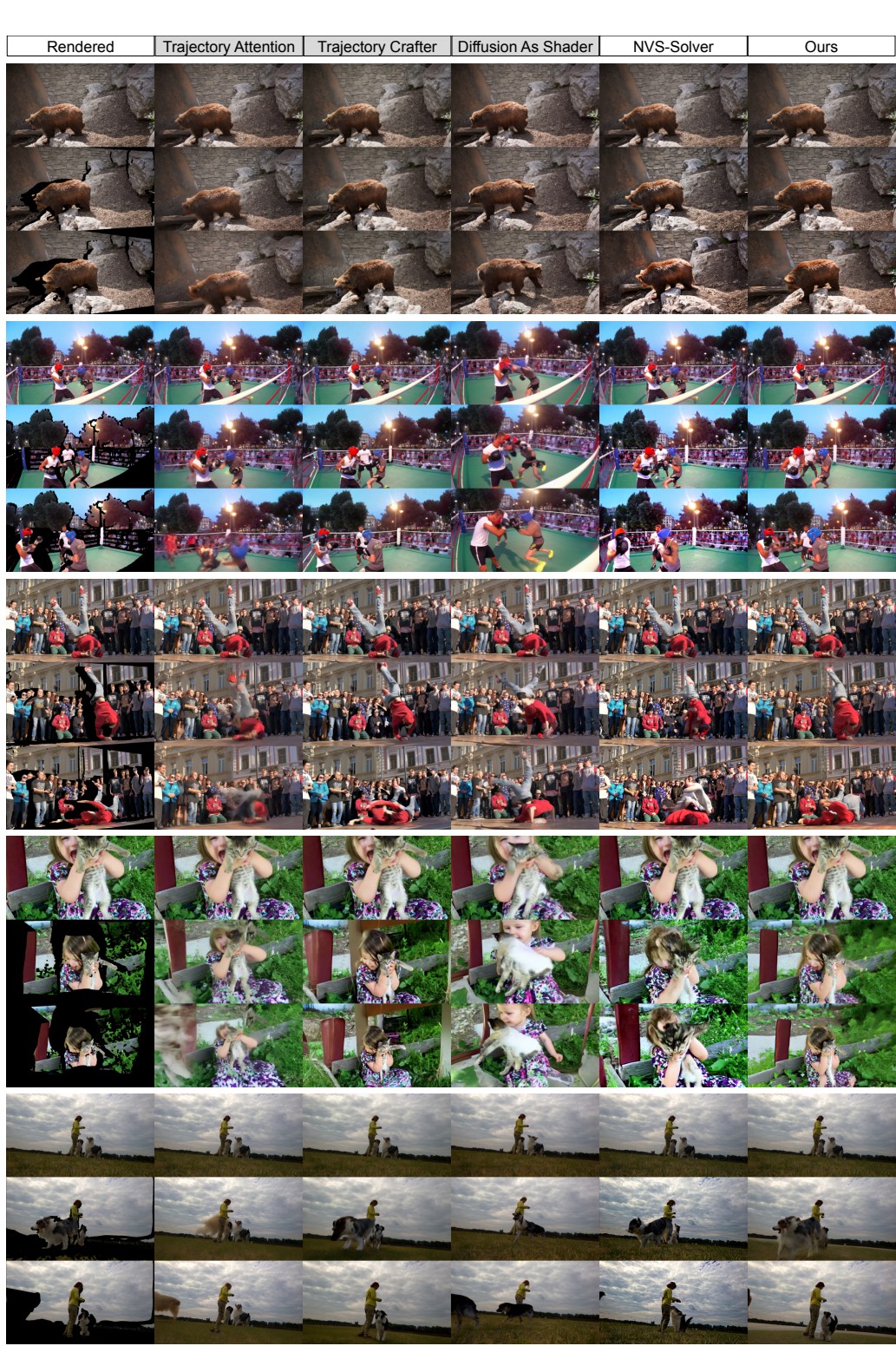

Figure 22: Qualitative comparison on DAVIS video inputs.

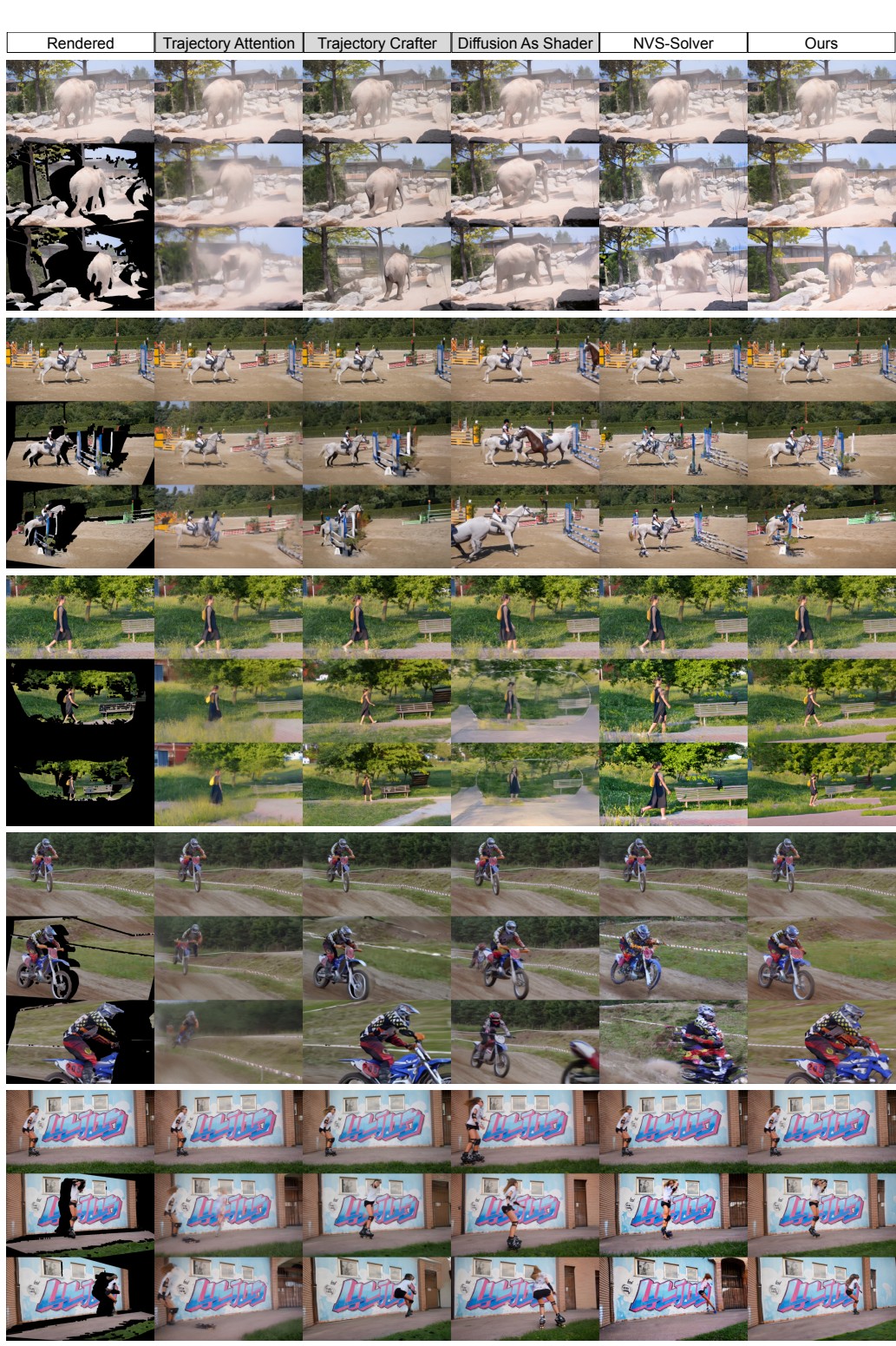

Figure 23: Qualitative comparison on DAVIS video inputs.

From Table 19, we see that our method achieves the best scores for the faithfulness metrics and the second-lowest camera pose errors, while achieving high FID/KID scores. However, the VBench scores in the table below are not as competitive. We attribute this to a specific failure mode of the other methods, especially Trajectory Attention, Diffusion As Shader, and NVS-Solver. When they fail to align with the rendered images, they tend to generate frames that are similar to the first frame. This behavior produces near-static videos, which artificially inflate their inter-frame consistency scores.

## Q    LIMITATIONS

By design, our method prioritizes strict faithfulness to the rendered geometry, which inherently limits its ability to model view-dependent effects like dynamic shadows or reflections. While masking such a region can offer a partial remedy, explicitly modeling these phenomena within a faithfulness-centric framework remains an open research direction. Furthermore, our reliance on rendered content as a conditional signal presents a natural trade-off: larger camera motions reduce the available guidance, potentially compromising geometric consistency. Addressing this challenge, perhaps by integrating semantic priors for plausible extrapolation, constitutes a promising avenue for future work.

