# OpenReview forum: "Prioritizing Faithfulness: Efficient Zero-Shot Novel View Synthesis with Adaptive Latent Modulation"
_ICLR.cc/2026/Conference — Submitted to ICLR 2026_

### Official Review · Reviewer_WhGA · 2025-10-25

**Soundness:** 3
**Presentation:** 2
**Contribution:** 2
**Rating:** 4
**Confidence:** 3

**Summary:**

The paper introduces a training-free method for generative novel view synthesis. The proposed method is based on a video diffusion model and introduces two new components compared to prior art: A inference-time optimization for homographies that are used to warp latents of different video frames, as well as a spatially varying noise level used in the inpainting process. The introduced changes are motivated by apparent 'spurious motion', as well as 'drifting synthesis' artefacts in related methods, where the former refers to subjects not remaining multi-view consistent even though having been observed from a novel view, while the latter refers to image (part) generations not following the camera motion.

**Strengths:**

The paper introduces a few relatively simple fixes that seem to be nonetheless effective for the task of generative inpainting.
The proposed spatially-adaptive RePaint variant seems novel and is well-motivated. Using the cross-attention of the video diffusion model as guidance signal to provide cross-frame correspondences is a neat trick in the absence of a clean, generated RGB video.

The paper is overall well-structured and provides an extensive amount of analysis and details on the proposed components, which facilitates reproducing the results.

**Weaknesses:**

**Evaluation and Comparisons to Prior Work**

(1)
The paper lacks comparison against multiple relevant related works. For example: InvisibleStitch [1], a method that additionally uses depth inpainting, which also resolves the issue of the proposed homography warping being not depth-aware or Stable Virtual Camera [2]. This makes me question whether all relevant related work has been cited and compared against.

(2)
The generated novel views shown in the qualitative results are in general less convincing than the prior work Trajectory Crafter, which is also reflected in the quantitative analysis. However, the reported "Faithfulness" scores are better for the proposed method, which makes me wonder how these scores were computed. A more thorough explanation on "valid" regions used for these metrics would be commendable.

(3)
While the paper provides a comparison to prior works wrt. geometric consistency using TSED, it would be commendable to include MEt3R [3], which is a more robust metric.


**Homography Estimation**

(4)
The proposed homography estimation can only be computed on the masked co-visible image parts. However, for these parts, the estimated homography should generally be close to Identity, as the initial latent that is noised and subsequently denoised is the depth-warped input image. Related to that, Figure 3b is misleading. The $z_{0|t}$ is denoised from the (already) depth-warped $y$ (as explained in Sec. 4), not the non-warped input image.

**Limited Applicability to More Recent Video Diffusion Models**

The proposed method does not directly extend to more recent video diffusion models that employ a VAE which introduces temporal compression, as the inpainting masks can then not be directly applied to the latents anymore.


**Paper Writing and Figures**

The paper writing is often not easy to follow. E.g., the description of different RePaint variants, which is an important pre-requisite for the remainder of the paper, is not easy to parse, especially the indexing used.

The figure quality is generally not great, often pixelated. I would recommend the authors to include figures with increased resolution in a revised version. Also, the section labels in Fig. 3b do not match with the paper sections.

[1] Engstler, Paul, et al. "Invisible stitch: Generating smooth 3d scenes with depth inpainting." 2025 International Conference on 3D Vision (3DV). IEEE, 2025.

[2] Zhou, Jensen, et al. "Stable virtual camera: Generative view synthesis with diffusion models." arXiv preprint arXiv:2503.14489 (2025).

[3] Asim, Mohammad, et al. "Met3r: Measuring multi-view consistency in generated images." Proceedings of the Computer Vision and Pattern Recognition Conference. 2025.

**Questions:**

Following up on the above mentioned weaknesses, I would like to see clarifications regarding the following points:

(1)
Is the diffusion model backbone used in the different implementations of related works comparable to the used one in this work? Same goes for the depth estimation model used to compute the initial 3D information.

Could you please include comparisons of your method against all relevant, openly available SOTA methods? E.g., InvisibleStitch or Stable Virtual Camera.

(2)
Please detail how the faithfulness scores were computed. Which region was considered "valid"?

(3)
How does the geometric consistency of generated novel views compare to prior art when evaluated using MEt3R?

(4)
How strong is the warping that is usually introduced through the computed homographies? Could be shown through some proxy metric like a histogram of the introduced area change / IoU of warped and original image space.

---

> ### Author Response · Authors · 2025-11-23
>
> We truly appreciate your thoroughly careful reading of our manuscript, as well as your comprehensive feedback. We have addressed your concerns and provided detailed responses to each point below.
>
> ---
> **W1: Comparison with Prior Works**
> > The paper lacks comparison against multiple relevant related works. For example: InvisibleStitch [1], a method that additionally uses depth inpainting, which also resolves the issue of the proposed homography warping being not depth-aware or Stable Virtual Camera [2]. This makes me question whether all relevant related work has been cited and compared against.
>
> We thank the reviewer for pointing out these relevant works. We acknowledge that they should be cited and have updated the "Related Works" section accordingly. While we did initially investigate these methods, we excluded them from the main comparison for specific technical reasons detailed below.
>
> - InvisibleStitch: This method (and similar ones like CamTrol) operates by iteratively performing RGBD inpainting and 2D-to-3D point lifting on a per-frame basis. Specifically, it renders the accumulated point cloud to the new view, inpaints the missing regions, lifts them back to 3D, and merges them with the existing geometry. The critical flaw in this iterative "render-inpaint-lift" loop is error accumulation. If the inpainting fails or generates inconsistent geometry in a single frame, that error is permanently baked into the 3D point cloud and propagates to all subsequent renderings without any correction mechanism. In our experiments (including tests with an unofficial CamTrol implementation: https://github.com/LAARRRY/CamTrol), we found this approach to be extremely vulnerable to temporal consistency, resulting in severe artifacts that could not match the quality of video diffusion-based baselines. (We have included visual examples of these failure cases in the supplementary materials under the `rebuttal/other_baselines` folder.) While we have added a discussion of these methods in the "Related Works" section to acknowledge their contribution, we excluded them from the main quantitative tables as they do not meet the baseline quality required for a meaningful comparison.
>
> - Stable Virtual Camera: Our primary research goal is to achieve high faithfulness to the input rendered image guidance. However, Stable Virtual Camera operates without any explicit 3D guidance during the generation process, meaning that it doesn’t use rendered images as a condition. Consequently, it belongs to a different category of methods from ours that cannot, by design, guarantee the pixel-level alignment with the source view's projection that we aim to evaluate. Comparing our explicit rendering guidance method against their implicit approach on faithfulness metrics would be structurally unfair, while comparing on fidelity alone would disregard our main contribution to faithfulness retention. Therefore, we focused our comparison on methods that share the same "render-and-inpaint" scheme (TrajectoryCrafter, etc.) to ensure a rigorous evaluation of our specific contribution. For the same reason, ReCamMaster is also excluded from the comparison.

---

> ### Author Response · Authors · 2025-11-23
>
> **W2/Q2: Definition of Faithfulness**
> > The generated novel views shown in the qualitative results are in general less convincing than the prior work Trajectory Crafter, which is also reflected in the quantitative analysis. However, the reported "Faithfulness" scores are better for the proposed method, which makes me wonder how these scores were computed. A more thorough explanation on "valid" regions used for these metrics would be commendable.
>
> This is deeply related to our main claim, so let us give a detailed explanation. First, the valid region means the areas where the point cloud rendering provides valid information, i.e, the rendered pixels are not black. For the faithfulness score calculation, we mask out the void (black) pixels and compute PSNR/SSIM/LPIPS only on the remaining regions. This directly reflects our philosophy of prioritizing strict adherence to the input visual information (rendered guidance).
>
> One might argue that this differs from the conventional way of evaluating the faithfulness metrics, where they are computed on the entire images, including invalid regions. There are several reasons we intentionally avoided this calculation scheme.
>
> - Metric Entanglement: Evaluating the entire image without differentiating valid/invalid regions conflates two distinct objectives: (1) faithfulness to the conditioning input, and (2) the realism of hallucinated content in occluded areas (fidelity). Since our primary goal is to address the lack of faithfulness in prior work, isolating the valid region is crucial.
>
> - Ambiguity of Inpainting: For invalid regions, there are infinite "correct" ways to inpaint plausible textures. Standard reference-based metrics (PSNR/SSIM/LPIPS) penalize valid generations simply for not matching a specific ground truth, which is unreasonable for generative tasks.
>
> - Dataset Limitations: Evaluating on full images requires multi-view ground truth. However, available datasets are often limited to static scenes or lab settings, failing to cover the casual, dynamic scenarios like those in DAVIS that we target (where the issues such as “spurious motion” and “drifting synthesis” become prominent). Even real-world datasets like DyCheck suffer from misalignment issues that make rigorous pixel-wise evaluation difficult (see Appendix J).
>
> Therefore, we chose DAVIS as the main dataset and regard the rendered images as the ground truth that the model must preserve.
>
> There may be a possible concern that this metric could be trivialized by simply pasting the rendered pixels onto the output. While this would indeed maximize faithfulness, it would severely degrade fidelity and temporal consistency due to boundary artifacts and lack of harmony with the inpainted region. By monitoring fidelity metrics like FID and VBench alongside faithfulness, we ensure that our method achieves a meaningful balance of preserving input content without sacrificing perceptual quality.
>
> Nevertheless, to ensure a fair comparison, we conducted an additional evaluation on the full image domain using the DyCheck dataset. As shown in Table 9 (Appendix L), our zero-shot method achieves performance highly competitive with the training-based TrajectoryCrafter even under this standard full-image protocol. This result confirms that our high scores are not merely artifacts of the metric definition but reflect an excellent balance between faithfulness and fidelity.
>
> Finally, we strongly recommend viewing the supplementary videos. Assessing the balance between faithfulness, fidelity, and geometric consistency is difficult from static frames alone.
>
> ---
> **W3/Q3: MEt3R Evaluation**
> > While the paper provides a comparison to prior works wrt. geometric consistency using TSED, it would be commendable to include MEt3R [3], which is a more robust metric.
>
> We sincerely thank the reviewer for this valuable suggestion. We have conducted additional evaluations using MEt3R on all three datasets (DAVIS, Mannequin Challenge, and Tanks and Temples) and updated the tables in the Experiments section and Appendix H. As shown in the "M3R" column, the results are largely consistent with the TSED scores, reinforcing the validity of our previous analysis about geometric consistency.

---

> ### Author Response · Authors · 2025-11-23
>
> **W4/Q4: Homography Deformation**
> > The proposed homography estimation can only be computed on the masked co-visible image parts. However, for these parts, the estimated homography should generally be close to Identity, as the initial latent that is noised and subsequently denoised is the depth-warped input image. Related to that, Figure 3b is misleading. The $z_{0|t}$ is denoised from the (already) depth-warped  (as explained in Sec. 4), not the non-warped input image.
>
> We respectfully clarify a potential misunderstanding regarding the nature of the intermediate latent $z_{0|t_i}$​ and the role of the diffusion prior. Even though the depth-warped images are used as a condition in homography deformation and SA-RePaint, the diffusion model's output in the subsequent denoising steps strongly **tends to revert to a static, non-warped video.** This phenomenon is exactly what we referred to as the "inherent static bias" of the pre-trained video diffusion model in Section 4.1. Because the model prefers generating static content (or content spatially aligned with the original source view) over the forced camera motion, the predicted latent $z_{t_{i+1}}$ in the next denoise step often deviates from the depth-warped condition.
>
> Accordingly, the estimated homography is **far from identity**, especially at the former denoising steps where this bias is strongest. We must consistently apply homography deformation at every step (up to 30/50) to actively counteract this bias and steer the generation toward the target camera motion. We have added videos comparing the intermediate results with and without homography in the supplementary material (`rebuttal/homography` folder), which clearly demonstrate this effect. Figure 8 also illustrates the deformation process across different steps.
>
> Regarding Figure 3b, with the above understanding, we believe the current depiction is appropriate. While it is true that $z_{0|t_i}$​ is technically derived from a latent that was deformed in the previous step, explicitly tracking the cumulative deformations over the course of denoising in the notation would lead to excessively cluttered expressions (e.g., $z^{\mathrm{deformed}, \mathrm{deformed},\ldots, \mathrm{deformed}}$). Since we never refer to or reuse the non-warped latents outside the context of the homography estimation block, the notational omission of the “deformed” symbol will be valid.
>
> As for the homography strength, please refer to the supplementary videos in the `rebuttal/homography` folder. This will immediately clarify that homography deformation is effectively enforcing camera-controlled scene motion, and the dynamics that the deformation gradually converges to identity as the denoising step increments.

---

> ### Author Response · Authors · 2025-11-23
>
> **W5: Applicability to DiT Models**
> > The proposed method does not directly extend to more recent video diffusion models that employ a VAE which introduces temporal compression, as the inpainting masks can then not be directly applied to the latents anymore.
>
> We appreciate the reviewer raising this important technical point regarding models with temporal compression VAEs. While it is true that temporal compression introduces complexity in mapping pixel-space masks to the latent space, we respectfully clarify that our method is not fundamentally limited by this architecture and can indeed be extended to such models.
>
> We are now testing our method with the Wan2.2-TI2V-5B pipeline for its low-cost inference capability (Note that inference cost is also one of our main concerns addressed in the paper). Its VAE compression ratio is 4x16x16, so we independently batch 4 consecutive rendering mask frames into a chunk and resize them by the above factor, obtaining a latent mask. The rendered RGB images are, after prefilling, directly fed to the VAE encoding without any intervention, and we get the conditioning latent. To our surprise, this straightforward strategy yields effective conditioning: applying our method with this latent mask immediately produced camera-controlled scene motion with high adherence to the rendered guidance.
>
> However, we observed some degradation in inpainting quality. By investigating the cause, we found that this could be resolved by adjusting the hyperparameters: specifically, terminating the homography deformation at a much earlier denoising step and reducing the strength of the attention key weighting (Section 4.2.3). With these modifications, we are observing promising results that balance faithfulness and fidelity. We plan to update the paper once the best configuration is finalized and the experiments are complete. For immediate reference, we have included preliminary results in the supplementary material (under the `rebuttal/wan` folder).
>
> Additionally, regarding the versatility of our method, we have also verified the effectiveness of SA-RePaint on Stable Diffusion 2 for image inpainting. The results are provided in Appendix M. We apologize if you have already reviewed this section; we mention it here primarily because we missed including a direct reference to Appendix M in the main paper.
>
> ---
> **W6: Paper Writing and Figures**
> > The paper writing is often not easy to follow. E.g., the description of different RePaint variants, which is an important pre-requisite for the remainder of the paper, is not easy to parse, especially the indexing used.
> The figure quality is generally not great, often pixelated. I would recommend the authors to include figures with increased resolution in a revised version. Also, the section labels in Fig. 3b do not match with the paper sections.
>
> We sincerely thank the reviewer for the feedback on the presentation. We acknowledge that the current explanation is dense and not easy to comprehend. Regarding the indexing style $z_{t_i}$​​, while we agree it is complex, we believe it is mathematically inevitable to introduce SA-RePaint seamlessly. Specifically, SA-RePaint transitions $z_{0|t}$ back to any continuous noise level Σ, which necessitates the explicit juxtaposition of Σ and t in the form $z_{Σ∣t}$​ . Therefore, for notational consistency, it will be desirable to use "t"-based indexing $z_{t_i}$ elsewhere instead of "i"-based indexing $z_i$. That said, we recognize there is room to improve the structural flow of the explanation. We view readability as an ongoing priority and will continue to polish the writing structure in future updates to ensure the method is as accessible as possible.
>
> We also appreciate the reviewer pointing out the issue with the figure resolution. We have identified that an unintentional downsampling occurred during the PDF compilation. We have fixed this issue and also replaced all figures with high-resolution vector or raster images in the revised version. We also corrected the mismatched section labels in Figure 3b. Thank you so much for pointing this out.

---

> ### Author Response · Authors · 2025-11-23
>
> **Q1: Backbone Comparison**
> > Is the diffusion model backbone used in the different implementations of related works comparable to the used one in this work? Same goes for the depth estimation model used to compute the initial 3D information.
>
> [Depth Estimation Model]
>
> To ensure a strictly fair comparison in the main inpainting part, we unify the depth estimation backbone across all methods in our experiments. Specifically, we utilize DepthAnythingV2 to generate the initial depth maps and subsequent point cloud renderings for all comparison methods; that is, all baselines use the same rendered images as a condition for inpainting. Feeding them the same rendered guidance derived from DepthAnythingV2 eliminates any discrepancy arising from the initial 3D information. As a side note:
>
> - TrajectoryAttention and NVS-Solver don’t suffer from any domain gap since they also adopt the same depth estimation model.
>
> - TrajectoryCrafter also doesn’t suffer from any issue: although it uses DepthCrafter, it postprocesses the scale so that it matches that of DepthAnythingV2.
>
> - DiffusionAsShader (DaS) is the exception, adopting DepthPro. However, DaS is designed to accommodate any trajectory map for enabling multiple tasks in a single model, so this domain gap will not be significant.
>
> [Diffusion Model Backbone]
>
> There is indeed a difference in the diffusion backbones. Our method, TrajectoryAttention, and NVS-Solver employ Stable Video Diffusion (SVD). In contrast, TrajectoryCrafter and DiffusionAsShader are built upon CogVideoX, a more recent and generally more capable video generation model. **This puts our method at a disadvantage**, as SVD is an older generation model compared to CogVideoX. However, we believe this comparison highlights a key strength of our approach: **despite using a less advanced backbone, our method achieves performance competitive with, or even superior to, methods using CogVideoX.** This underscores the effectiveness of our proposed mechanism to maximize faithfulness to the conditioning signal, which compensates for the backbone's generational gap and proves that rigorous geometric guidance is as critical as the base model's raw capability, a less explored strategy in previous works.

---

> ### Comment · Reviewer_WhGA · 2025-11-24
>
> **W1: Comparison to Prior Work**
>
> I thank the authors for their explanations. However, I would still like to encourage the authors to evaluate their method against the most prominent approaches in the task of generative novel view synthesis and not just a certain subset. Including results for InvisibleStitch, Stable Virtual Camera, and other relevant baselines would provide a more comprehensive view of the method’s strengths and limitations and prevent potential bias from a too narrowly defined comparison set.
>
> **W2: Definition of Faithfulness**
>
> While you argue that you strictly prioritize faithfulness in your method, I believe this is not really reflected well in the evaluation. I agree that measuring the faithfulness to the condition image only makes sense on the valid, i.e., co-visible, image regions. However, as far as I understand your proposed faithfulness metric, you are using the same depth estimation model to run your model and to evaluate it, i.e., to find the valid image regions.
> In my eyes, this does not make sense. Instead, it would be much more meaningful if you were to use ground truth camera trajectories for which, in the best case, depth and camera poses are already given. In case this ground truth is absent, you can still use VGGT or any other few-view feed-forward reconstruction model to find the co-visible image regions in the ground truth image trajectory.
> This would ensure a fair comparison and also enables a better way for comparison against methods like the ones I mentioned in W1.
>
> **W3: MEt3R evaluation**
>
> Thanks for running this evaluation.
>
> **W4: Homography Deformation**
>
> Thank you for this clarification. I suggest you to revisit the according paragraphs and extend / refine the explanation of the prevailing 'inherent static bias' following the explanation you provided here. To stress this again, it would be good to show the "magnitude" of this homography warping (see my initial questions) quantitatively, as this observation is one of the key motivations for your method and also seems important for determining good hyperparameters.
>
> **W5: Applicability to DiT Models**
>
> Thanks for trying out your method on diffusion models with temporal compression. Please include a comparison against RePaint and SD-RePaint in this experiment, as it seems a bit worrying that stopping the homography estimation early and reducing the strength of the attention key weighting, and thus essentially removing major parts of the proposed novel contributions, leads to better results.
>
> **W6: Paper Writing and Figures**
>
> Thank you for acknowledging that the current explanation is not easy to comprehend. Please stay true to your word and revise the formulations in the updated manuscript.
> Thank you for updating the figure quality, much better now.
>
> **Q1: Backbone Comparison**
>
> Why did you choose to not use CogVideoX in your method if it is a "more recent and generally more capable video generation model"?

---

> ### Author Response · Authors · 2025-12-03
>
> **W1: Comparison to Prior Work**
> > I thank the authors for their explanations. However, I would still like to encourage the authors to evaluate their method against the most prominent approaches in the task of generative novel view synthesis and not just a certain subset. Including results for InvisibleStitch, Stable Virtual Camera, and other relevant baselines would provide a more comprehensive view of the method’s strengths and limitations and prevent potential bias from a too narrowly defined comparison set.
>
> We appreciate your persistent encouragement. Following your suggestion, we have conducted additional experiments comparing our method with Invisible Stitch and Stable Virtual Camera (SEVA) using the Mannequin Challenge dataset under the Real Camera Motion setting, which enables comparing our method with SEVA legitimately (cf. answer to W2). The detailed qualitative and quantitative results are provided in Appendix M (Table 12 and Figure 13). For your convenience, we duplicate the quantitative table below:
>
>
> | Method | Input-Faithfulness <br> PSNR↑ / SSIM↑ / LPIPS↓ | GT-Faithfulness <br> PSNR↑ / SSIM↑ / LPIPS↓ | Fidelity ↓ <br> FID / KID' |
> | :--- | :---: | :---: | :---: |
> | Invisible Stitch | 23.18 / 0.804 / 0.215 | 18.63 / 0.633 / 0.306 | 62.89 / 13.79 |
> | Stable Virtual Camera | N/A / N/A / N/A | 13.69 / 0.497 / 0.393 | 33.59 / 4.331 |
> | **Ours** | **30.51** / **0.925** / **0.056** | **19.03** / **0.663** / **0.226** | **26.18** / **1.865** |
>
>
> However, after conducting these experiments, we reaffirmed our decision to report these results in the Appendix rather than the main section. We found that establishing a strictly fair comparison with the primary baselines is technically difficult due to the accumulation of various configuration mismatches and inherent methodological constraints. We summarize the observed obstacles below:
>
> - Invisible Stitch: Our evaluation confirms that its iterative per-frame "render-inpaint-lift" loop leads to severe error accumulation, as evidenced by its poor Fidelity scores. However, we cannot conclude that this is the sole reason because of other unresolvable architectural differences: (1) depth completion module needs to be replaced by Prior Depth Anything to align with other baselines, (2) our experimental setting requires large camera motion while Invisible Stitch is mainly targeted to panoramic scene generation, (3) PyTorch3D-based point cloud rendering likely exhibits dot or seam-like artifacts, and determining the suitable point size for fix is highly scene-dependent, (4) keyframe selection criteria for 3DGS training and the 3DGS training configurations leaves lots of room for parameter tuning.
>
> - Stable Virtual Camera (SEVA): We have found that the method is very susceptible to scale drift in the single-image setting. This is also reported in the original paper, and their proposed linear probing of the suitable scaler makes fair comparison difficult since we can never guarantee the best scaling value. After all, it is a manual heuristic preprocessing, and should be unnecessary in theory.
>
> Given these distinct operational paradigms and the difficulties in standardizing the evaluation conditions, we believe presenting these results in the Appendix as a "reference comparison" successfully provides the comprehensive view you requested without compromising the rigor of the main benchmarking.

---

> ### Author Response · Authors · 2025-12-03
>
> **W2: Definition of Faithfulness**
> > While you argue that you strictly prioritize faithfulness in your method, I believe this is not really reflected well in the evaluation. I agree that measuring the faithfulness to the condition image only makes sense on the valid, i.e., co-visible, image regions. However, as far as I understand your proposed faithfulness metric, you are using the same depth estimation model to run your model and to evaluate it, i.e., to find the valid image regions. In my eyes, this does not make sense. Instead, it would be much more meaningful if you were to use ground truth camera trajectories for which, in the best case, depth and camera poses are already given. In case this ground truth is absent, you can still use VGGT or any other few-view feed-forward reconstruction model to find the co-visible image regions in the ground truth image trajectory. This would ensure a fair comparison and also enables a better way for comparison against methods like the ones I mentioned in W1.
>
> We sincerely appreciate this critical insight. First, we respectfully argue that this design of using the same depth estimator to run a model and to evaluate it, i.e., comparing the rendered input images and generated output images, is **necessary to evaluate the model's controllability**. In a practical scenario, a user inputs an image to the pipeline and gets a sequence of rendered images, followed by their inpainted results. The main issue we point out here is that the rendered inputs and inpainted outputs don’t match in geometry, color, or texture. Even if the mismatch leads to a correct modification (e.g., making the rendered images brighter or enhancing the texture),  such deviation is perceived as a failure to respect the user's input since they don’t know the true ground truth. Rather, we claim that granting models this freedom of correction can increase the likelihood of failures.
>
> In addition, enforcing strict adherence to the “input” images facilitates a clear separation of concerns. The rendered images may be distorted, and being faithful to them may result in a collapsed novel-view video. Then it’s not the diffusion model’s task to correct it; it’s the depth estimation model that needs improvement. This responsibility clarification can not only expedite pipeline development but also make it highly modular: we can readily upgrade the depth estimation or inpainting components independently without worrying about domain shifts or hidden dependencies in the other component.
>
> However, we also acknowledge your point that evaluation against the true target is more reliable and important for NVS benchmarking. Although we initially set it aside due to the lack of multi-camera datasets capturing dynamic natural scenes where drifting is likely to occur, based on your suggestion, we have revised our evaluation protocol to reconcile our objective with standard practices. To start, we distinguish and report two complementary metrics:
>
> 1.  **Input-Faithfulness:** The adherence to the *projected input rendering* (our focus: measures controllability and drift).
> 2.  **GT-Faithfulness:** The similarity to the *ground-truth target view* (your suggestion: measures reconstruction accuracy).
>
> Both are measured on the region visible from the input image. Next, we updated the dataset: we employ Mannequin Challenge and DL3DV (newly added) to evaluate both metrics. Since these datasets capture static scenes with smooth camera trajectories, following your advice to utilize reconstruction priors, we applied Depth Anything 3 [???] to obtain pseudo-GT camera poses and depth maps. We use these maps to render novel views, and the inpainted results are compared against the target GT views.
>
> We obtained interesting results, which are summarized in the Experiments section of the main paper and Appendix H. We extract a part of Table 2 for your convenience:
>
> | Method | Input-Faithfulness <br> PSNR↑ / SSIM↑ / LPIPS↓ | GT-Faithfulness <br> PSNR↑ / SSIM↑ / LPIPS↓ |
> | :--- | :---: | :---: |
> | Traj. Attention | 21.12 / 0.735 / 0.215 | 18.29 / 0.617 / 0.281 |
> | Traj. Crafter | 22.77 / 0.793 / 0.164 | 18.71 / 0.645 / 0.269 |
> | NVS-Solver | 20.51 / 0.720 / 0.223 | 16.91 / 0.572 / 0.300 |
> | **Ours** | **28.95** / **0.893** / **0.074** | **18.97** / **0.652** / **0.241** |
>
> The first finding is a positive correlation between Input-Faithfulness and GT-Faithfulness across all methods, indicating that depth estimation accuracy is sufficient for the render-and-inpaint scheme to work correctly. The second is that our method also achieves the highest GT-Faithfulness among the other baselines. This empirical evidence strongly validates our core claim: prioritizing strict adherence to the input guidance (Input-Faithfulness) is an effective strategy for improving true NVS accuracy (GT-Faithfulness), and should thus be highlighted more within the community. We hope this dual-metric evaluation and the supporting results address your concerns.

---

> ### Author Response · Authors · 2025-12-03
>
> **W4: Homography Deformation**
> > Thank you for this clarification. I suggest you to revisit the according paragraphs and extend / refine the explanation of the prevailing 'inherent static bias' following the explanation you provided here. To stress this again, it would be good to show the "magnitude" of this homography warping (see my initial questions) quantitatively, as this observation is one of the key motivations for your method and also seems important for determining good hyperparameters.
>
> In Appendix E Figure 8, we have provided a line graph showing the evolution of deformation magnitude over different denoising steps. This clearly illustrates that the homography doesn’t readily converge to the identity. Although we could not fully elaborate on ‘how inherent static bias cancels out homography deformation’, we added a note that the homography convergence is gradual, not rapid. Thank you so much for your valuable feedback.
>
>
> **W5: Applicability to DiT Models**
> > Thanks for trying out your method on diffusion models with temporal compression. Please include a comparison against RePaint and SD-RePaint in this experiment, as it seems a bit worrying that stopping the homography estimation early and reducing the strength of the attention key weighting, and thus essentially removing major parts of the proposed novel contributions, leads to better results.
>
> We appreciate your suggestion. We provide a comprehensive description and experiments in Appendix N. Here we copy the result table (comparison with other baselines on the DAVIS dataset) for convenience:
>
> | Method | Input-Faithfulness <br> PSNR↑ / SSIM↑ / LPIPS↓ | Fidelity ↓ <br> FID / KID' / FVD | Camera Pose ↓ <br> ATE / RRE / RTE | Geometry <br> T.25↑ / T.50↑ / M3R↓ |
> | :--- | :---: | :---: | :---: | :---: |
> | Traj. Crafter | 24.24 / 0.811 / 0.119 | 32.03 / 1.906 / 704.7 | 0.139 / 0.047 / 0.071 | 0.499 / 0.958 / 0.032 |
> | Ours (SVD) | 29.58 / 0.864 / 0.074 | 28.14 / 0.816 / 705.2 | **0.051** / 0.022 / **0.027** | 0.672 / **0.964** / 0.033 |
> | **Ours (Wan2.2)** | **32.67** / **0.942** / **0.054** | **26.35** / **0.486** / **699.2** | 0.066 / **0.021** / 0.032 | **0.856** / 0.961 / 0.033 |
>
> We are excited to report the score boost in both faithfulness and fidelity. Especially, the fidelity is qualitatively much higher than our SVD baseline, now matching the quality of Trajectory Crafter. We strongly encourage you to check the supplementary folder under `rebuttal/wan`, where we compare the qualitative results of TrajectoryCrafter, ours (SVD), and ours (Wan2.2).
>
> We also conducted an ablation study to confirm the effectiveness of our devised modules: homography deformation and SA-RePaint. As shown in Table 17 and Figure 20, we notice a similar trend in score changes. Homography deformation surely contributes to camera pose accuracy and drift prevention, whereas SA-RePaint is instrumental for texture enhancement without losing global structural consistency.
>
> However, we admit that these trials are still not fully verified. We sometimes observe unsatisfactory drifting synthesis, which is reflected in the higher camera pose error than our SVD baseline, or obvious artifacts. We will address these issues in our future work.

---

> ### Author Response · Authors · 2025-12-03
>
> **Q1: Backbone Comparison**
> > Why did you choose to not use CogVideoX in your method if it is a "more recent and generally more capable video generation model"?
>
> Our decision to use Stable Video Diffusion (SVD) comes from two reasons: fair comparison and efficiency.
> 1. Fair comparison: Our primary baseline for comparison is NVS-Solver. It is the most relevant work to ours in that it satisfies three critical criteria at the same time: it is (1) zero-shot, (2) incorporates a basic faithfulness-enforcement mechanism, and (3) operates purely with a single video diffusion model (without relying on auxiliary foundation models) during inpainting. Deviating from these criteria fundamentally hinders our goal of simultaneously achieving efficiency and faithfulness by design. Consequently, the NVS-Solver architecture best offers the necessary extensibility for our purpose, making the comparison with NVS-Solver paramount to demonstrating the effectiveness of our devised methodologies. Adopting SVD, the same backbone used in NVS-Solver, is therefore a critical prerequisite to ensure a fair comparison.
> 2. Efficiency: Please note that constructing a resource-efficient framework is a core objective of our study. Recent advancements in video diffusion models, including CogVideoX, predominantly follow a trend of scaling up model size to pursue higher quality. Initiating our development with such massive models would significantly diminish the feasibility of achieving our efficiency criterion from the very outset. In contrast, SVD offers a distinct advantage in terms of lower inference costs. Given this inherent efficiency, combined with its adoption by other comparison baselines, selecting SVD as the foundational backbone represents a highly rational and strategic design choice.

---

### Official Review · Reviewer_Y4mM · 2025-10-30

**Soundness:** 3
**Presentation:** 3
**Contribution:** 3
**Rating:** 8
**Confidence:** 4

**Summary:**

Based on the video diffusion model, the authors propose a training-free novel view synthesis method. Compared to previous state-of-the-art work, they introduce homography optimization and Spatially Adaptive RePaint, demonstrating their effectiveness on datasets.

**Strengths:**

1. It is a training-free approach that leverages pre-trained large-scale video diffusion models, promising potential for improved performance as video generation techniques advance.
2. The generated videos exhibit high visual fidelity and competitive quality compared to other methods.

**Weaknesses:**

1. There is a lack of comparison with Gaussian Splatting.

**Questions:**

Although the primary contribution of the paper is in fidelity comparison, the background in the images of Figure 6 appears overly blurred. It would be better to replace these images with ones that better validate the experimental results' competitive fidelity.  (The figures in the appendix would be much better)

---

> ### Author Response · Authors · 2025-11-23
>
> Thanks for your constructive feedback on our work. Please find our response to your comments below:
>
> ---
> **W1: Comparison with 3DGS**
> > There is a lack of comparison with Gaussian Splatting.
>
> We appreciate the suggestion. While Gaussian Splatting (3DGS) sets the standard for reconstruction fidelity, we did not include it as a baseline for the following reason:
>
> Our target task of Single-Image-Based Camera-Controlled Novel View Synthesis requires two capabilities: (1) faithfulness to the visible source content, and (2) generative fidelity to hallucinate realistic textures for large occluded regions. While single-view 3DGS methods excel at (1), they fundamentally lack the generative prior required for (2). When the camera moves significantly and reveals unseen areas, 3DGS-based methods typically produce floaters, stretched artifacts, or holes, whereas diffusion-based methods can synthesize plausible content.
>
> Our research goal is to bridge this gap: incorporating the strict geometric consistency of explicit 3D representations (i.e., rendered point clouds) faithfully into the high-fidelity generation of video diffusion models. Since our primary contribution lies in enhancing the faithfulness of generative models, the most appropriate baselines are other diffusion-based NVS methods (TrajectoryCrafter, etc). Comparing against 3DGS would largely be a comparison of "Generation vs. Reconstruction," which diverts from our focus on controlling generative models.
>
> ---
> **Q1: Figure Quality**
> > Although the primary contribution of the paper is in fidelity comparison, the background in the images of Figure 6 appears overly blurred. It would be better to replace these images with ones that better validate the experimental results' competitive fidelity. (The figures in the appendix would be much better)
>
> We appreciate the reviewer's feedback regarding the visual quality in Figure 6. We carefully re-evaluated our figure selection and would like to explain why this specific example was chosen. Among the test cases, this scene most effectively highlights the misalignment from the rendered images present in the baseline methods compared to the strict faithfulness preserved by our approach. Although visually pleasing, replacing it with other examples would obscure this critical comparative insight, which is central to our paper's claim. However, to directly address the concern about background blurriness, we have updated Figure 6 with higher resolution in the revised manuscript to ensure the details are presented as clearly as possible while maintaining the comparative value of the scene.

---

### Official Review · Reviewer_QBDc · 2025-10-31

**Soundness:** 3
**Presentation:** 3
**Contribution:** 3
**Rating:** 6
**Confidence:** 3

**Summary:**

This paper introduces a training-free pipeline for NVS task from a single source image, which focuses on the faithfulness to the input while achieving efficiency. The method follows a render-then-inpaint manner: it first lifts the source image to a 3D point cloud, renders views along a specified camera trajectory. Then it inpaints disoccluded regions with SVD via a modified RePaint technique.

The key contributions include: (1) test-time latent homography deformation, an optimization that aligns latent predictions with rendered images to prevent drifting synthesis and ensure motion coherence, and (2) SA-RePaint, which derives a per-pixel noise map to balance structural consistency and texture fidelity by matching local variances.

The authors claim that the proposed method outperforms existing methods in faithfulness and camera accuracy with competitive visual quality.

**Strengths:**

The paper proses to rebalance priorities in NVS of faithfulness and efficiency, which is often sidelined in favor of fidelity in generative approaches. The technical details are clearly derived (e.g., the closed-form solution for $\Sigma$ in Theorem 1) and illustrative figures effectively convey the pipeline and trade-offs.

Overall, this paper is well-written. This work also presents a new perspective for generative NVS scenario where faithfulness is critical and efficiency is also demanded.

**Weaknesses:**

Given this is a training-free method for generative NVS, one weakness could be the assumption of the method.
The core method *Test-time Latent Homography Deformation* assumes largely planar or global motions (homography deformation), which may not handle complex parallax, non-rigid, and large-motion scenes well.

**Questions:**

1. How does the method perform on inputs with significant depth variations or dynamic elements, given homography's global nature? Although a brief discussion of the limitation in given in the appendix, some examples of failure cases would help evaluate the limitations of current method.

2. Since the proposed method relies on latent-space manipulations during inference and treats the video diffusion model as a black box, it appears naturally extendable to DiT architectures. Have the authors experimented with applying it to more recent DiT-based video models?

---

> ### Author Response · Authors · 2025-11-23
>
> Thanks for your appreciation of the originality, effectiveness, and readability of our work. Here is our response to your questions:
>
> ---
> **W1: Limitation of Homography**
> > Given this is a training-free method for generative NVS, one weakness could be the assumption of the method. The core method Test-time Latent Homography Deformation assumes largely planar or global motions (homography deformation), which may not handle complex parallax, non-rigid, and large-motion scenes well.
>
> We acknowledge the concern regarding the planar assumption of homography. However, we argue that this design does not hinder the modeling of depth parallax for the following reasons, supported by our empirical observations and the diffusion model's behavior:
>
> - As visualized in Figure 8, the video content is generated in a coarse-to-fine manner. At these early stages where Homography Deformation is active (up to step 30), the latent features represent only global structures and rough motion, where fine-grained depth parallax is not yet established. Therefore, planar deformation is sufficient to guide the "gist" of the camera movement without conflicting with detailed geometry. Since we cease the homography deformation in the latter steps, the generation of fine geometry structures doesn’t interfere with the homography at all.
>
> - Crucially, we observe that as the denoising process progresses, the estimated homography matrix naturally converges toward the identity. This indicates that once the global camera motion is successfully induced in the early steps, the video diffusion model itself begins to consistently generate the subsequent motion and appropriate parallax effects based on its learned priors. In other words, the homography deformation serves as an initial guidance for the camera trajectory. By the time fine parallax details start to be formed (e.g., around step 27/50 and beyond), the deformation becomes minimal or is explicitly disabled (after step 30), allowing the model to freely hallucinate correct 3D parallax without being constrained by the planar assumption.
>
> - The datasets we used (DAVIS, Mannequin Challenge, and Tanks and Temples) all contain highly diverse depth variation, meaning that the virtual camera motion inevitably induces noticeable depth parallax effects. However, upon manually inspecting all our experimental results, we couldn’t identify any videos that failed to model depth parallax effects.
>
> We have added videos comparing the intermediate results before and after applying homography. Please refer to the supplementary material under the `rebuttal/homography` folder.
>
> ---
> **Q1: Potentioal Failure Cases**
> > How does the method perform on inputs with significant depth variations or dynamic elements, given homography's global nature? Although a brief discussion of the limitation in given in the appendix, some examples of failure cases would help evaluate the limitations of current method.
>
> As discussed above, it is difficult to find failure cases directly induced by the homography deformation itself. While this speaks to the robustness of our approach, there is an alternative explanation for scenarios with extreme depth variations: the guidance signal from the rendered images effectively disappears before the homography can cause artifacts.
>
> Consider a "selfie" scenario (a subject 1-2 meters from the camera with a distant background), which typifies significant depth variation. If the camera moves significantly along an orbital trajectory around the subject, most background pixels are immediately occluded by the foreground subject and then reappear from behind in subsequent frames. However, as explained in Appendix G, we discard these disoccluded pixels from the rendered images as they are considered unreliable. Consequently, under such large camera motion, the rendered images become mostly black (invalid), resulting in a loss of camera-control guidance. Any potential geometric artifacts from homography are possibly overshadowed by this more dominant failure mode. This point is indeed a limitation of our current framework, which is to be addressed in future work.

---

> ### Author Response · Authors · 2025-11-23
>
> **Q2: Applicability to DiT Models**
> > Since the proposed method relies on latent-space manipulations during inference and treats the video diffusion model as a black box, it appears naturally extendable to DiT architectures. Have the authors experimented with applying it to more recent DiT-based video models?
>
> We thank the reviewer for this insightful question. Indeed, the model-agnostic nature of our approach makes it theoretically compatible with DiT-based architectures, and this is exactly an area we are currently exploring. Since efficiency is one of the key concerns in the paper, we target Wan2.2-TI2V-5B as the new baseline for its affordability. In our initial experiment, simply integrating our methods with the same parameter settings was not successful, in that the inpainting quality significantly dropped. Upon further investigation, we found that homography deformation can be halted at a much earlier denoising step. Moreover, the attention key weighting introduced in Section 4.2.3 needs to be weakened. After these adjustments, we are seeing promising camera-controlled generation results. It seems that Wan2.2 is more sensitive to the latent manipulation than SVD. We are now investigating the best configuration and preparing for experiments, after which we will update the paper. The current exemplary result is updated in the supplementary material under the `rebuttal/wan` folder.
>
> Additionally, regarding the versatility of our method, we have also verified the effectiveness of SA-RePaint on Stable Diffusion 2 for image inpainting. The results are provided in Appendix M. We apologize if you have already reviewed this section; we mention it here primarily because we missed including a direct reference to Appendix M in the main paper.

---

### Official Review · Reviewer_5Z8V · 2025-11-01

**Soundness:** 3
**Presentation:** 3
**Contribution:** 3
**Rating:** 4
**Confidence:** 3

**Summary:**

This work introduces a training-free NVS method aimed at improving faithfulness and geometric consistency in diffusion-based novel view synthesis. It introduces a Latent Homography Deformation module to enforce content coherence and a Spatially Adaptive RePaint mechanism to address the structure-texture trade-off. The proposed method achieves competitive results in both quantitative and qualitative evaluations on several benchmark datasets.

**Strengths:**

* This work addresses the challenging faithfulness issues that typically exist in NVS methods relying on diffusion models and manages to mitigate them through a training-free solution.


* The idea of applying homography warping in the latent space to address “drifting synthesis” issues is interesting and sound.


* The concept of applying per-pixel noise levels is mathematically well illustrated.


* The manuscript is well structured and easy to follow.

**Weaknesses:**

* **The ablation study seems confusing.** As reported in Tab. 2, compared to the baseline, the introduced components bring only minor improvements, or even worse results, in terms of faithfulness. This contradicts the motivation of “prioritizing faithfulness.” A more thorough analysis would help clarify this issue.


* **The improvements over other comparison models seem to mainly stem from the enhanced baseline.** As shown in Tab. 1, the proposed method performs significantly better than other state-of-the-art models. However, Tab. 2 shows that the final model performs even worse than the baseline. In this case, the superiority of the method may purely come from an improved baseline. It would be more convincing to apply the proposed modules to other baselines, such as Trajectory Crafter, to verify whether these introduced components genuinely contribute to the performance gains.


* **The effectiveness of the “prefill” module is not analyzed.** As mentioned in L208, the re-projected image is first prefilled with a classical inpainting method before being fed into the VAE. It would be helpful to show how important this step is. Moreover, applying the same prefill step to other comparison methods, such as Trajectory Crafter, could help ensure a fairer comparison.

**Questions:**

For the comparison figures (e.g., Fig. 6 and Fig. 7), it would be better to include the Ground Truth images to confirm whether the proposed method indeed maintains faithfulness better than others.

---

> ### Author Response · Authors · 2025-11-23
>
> We sincerely appreciate your recognition of our manuscript and proposed methods. Please find below our clarification to each of your concerns:
>
> ---
> **W1: The ablation study seems confusing**
> > As reported in Tab. 2, compared to the baseline, the introduced components bring only minor improvements, or even worse results, in terms of faithfulness. This contradicts the motivation of “prioritizing faithfulness.” A more thorough analysis would help clarify this issue.
>
> We clarify that the baseline achieves the highest "faithfulness" score precisely because it focuses exclusively on preserving the valid regions, often by sacrificing global consistency and fidelity in the inpainted regions. Since the faithfulness metrics are computed only on the valid (non-black) regions (cf. the answer to Q1 below), a naive baseline that simply copies the rendered pixels without explicit care to harmonize them with the inpainted areas will naturally score high on these specific metrics. However, as visible in the supplementary videos in the `ablation` folder, this results in severe artifacts (e.g., blurry backgrounds, geometric drifting) in the final output, especially in the inpainted region.
>
> Our proposed components are designed to balance this trade-off. Their goal is to achieve a level of perceptual quality and geometric consistency (FID, KID, FVD, VBench) in the inpainted regions comparable to other state-of-the-art methods, while maintaining the highest possible faithfulness. Bearing this in mind, the slight numerical decrease in faithfulness compared to the naive baseline is generally inevitable but fully acceptable, as our full model strikes a much better overall balance between faithfulness and fidelity.

---

> ### Author Response · Authors · 2025-11-23
>
> **W2: The improvements over other comparison models seem to mainly stem from the enhanced baseline**
> > As shown in Tab. 1, the proposed method performs significantly better than other state-of-the-art models. However, Tab. 2 shows that the final model performs even worse than the baseline. In this case, the superiority of the method may purely come from an improved baseline. It would be more convincing to apply the proposed modules to other baselines, such as Trajectory Crafter, to verify whether these introduced components genuinely contribute to the performance gains.
>
> We appreciate this suggestion to verify our components on other baselines. However, we respectfully argue that applying our proposed modules to methods like TrajectoryCrafter or NVS-Solver is methodologically ill-suited, as they rely on fundamentally different mechanisms (fine-tuning or test-time optimization). Our modules: homography deformation and SA-RePaint are specifically designed to address the issues inherent in an “efficient zero-shot” regime, particularly to overcome the challenges of native RePaint.
>
> To recapitulate, native RePaint achieves the highest faithfulness due to its exclusive focus on the valid rendered area, and so results in extremely poor inpainting fidelity. In light of this, homography deformation addresses the drifting synthesis of the inpainted area, while SA-RePaint ensures better harmonization between valid and invalid regions in terms of texture. Neither of them directly addresses the faithfulness since it’s already heightened enough by the RePaint operation, and they are rather designed to compensate for its defects without significantly compromising this high faithfulness.
>
> We now see that it will not be that effective to apply these two components to other baselines for the following reasons:
>
> - TrajectoryCrafter is trained to balance the camera controllability and generative fidelity, so drifting synthesis rarely occurs in this model. Therefore, homography deformation will not have a noticeable effect. On the other hand, it sometimes exhibits spurious motion, i.e., the faithfulness issue. In this case, SA-RePaint may give a benefit. However, the potentially better faithfulness effect should come from the native RePaint for the reason duly described above; so we expect that the faithfulness score by using native RePaint will be slightly higher compared to using SA-RePaint, similarly to Table 2, offering no new insight.
>
> - NVS-Solver is also a zero-shot method, and it suffers from both drifting synthesis and spurious motion. Therefore, homography deformation could remedy the former (at the expense of a slight decrease in faithfulness). However, the integration of SA-RePaint is more subtle. Originally, NVS-Solver applies test-time optimization for faithfulness to the rendered images. Our SA-RePaint (or the native RePaint) also aims for this faithfulness in an optimization-free manner. Since they are fundamentally different in methodology, there is no guarantee that they will yield a synergistic effect (E.g., test-time optimization may cancel out the outcome of SA-RePaint). Even if the result were positive, the faithfulness scores when using SA-RePaint would be slightly inferior to those obtained when using the native RePaint, which is as expected by design.
>
> In summary, Homography Deformation and SA-RePaint are not designed to simply boost the metrics of the original RePaint. Instead, their role is to rebalance the trade-off between faithfulness and fidelity, demonstrating their true value specifically within the efficient zero-shot inference setting.

---

> ### Author Response · Authors · 2025-11-23
>
> **W3: The effectiveness of the “prefill” module is not analyzed**
> > As mentioned in L208, the re-projected image is first prefilled with a classical inpainting method before being fed into the VAE. It would be helpful to show how important this step is. Moreover, applying the same prefill step to other comparison methods, such as Trajectory Crafter, could help ensure a fairer comparison.
>
> We appreciate the reviewer bringing this to our attention. Following the advice, in the revised paper (Appendix J, Figure 11), we demonstrate the qualitative impact of the prefill module. As shown, omitting this step results in noticeable gray border artifacts at the boundaries between valid and inpainted regions. This validates our claim in Section 4 that prefilling prevents black void pixels from "contaminating" valid pixels during VAE encoding, and is a crucial step for ensuring high fidelity in conditional generation. Regarding the fairer comparison, we provide quantitative results of applying the same prefill step to other baselines in Table 8 (+ indicates the addition of the prefill step). While baselines like NVS-Solver show improvements, they still significantly lag behind our method. This confirms that our performance advantage stems from the core design of our approach, rather than being solely attributed to this pre-processing technique.
>
> ---
> **Q1: Clarification about Ground Truth**
> > For the comparison figures (e.g., Fig. 6 and Fig. 7), it would be better to include the Ground Truth images to confirm whether the proposed method indeed maintains faithfulness better than others.
>
> We appreciate the suggestion. However, we would like to respectfully mention that no complete "ground truth" images exist for the DAVIS dataset in the context of our task (novel view synthesis from a single image with depth warping), as we do not have multi-view synchronized RGBD captures for these dynamic scenes.
>
> In our main experiments (Table 1 and Fig. 6/7), the "ground truth" for faithfulness evaluation is defined as the valid (non-void) regions of the rendered input itself. Our primary goal is to faithfully preserve these visible pixels from the source view.
>
> Although we can evaluate our results against a complete target image without differentiating valid and invalid regions by using appropriate datasets, we avoid it on purpose because it would mix two different evaluation criteria: (1) how well the model preserves the visible source pixels (faithfulness), and (2) how realistically the model hallucinates the occluded areas (fidelity). We believe these should be evaluated separately, and so the latter is measured by FID or VBench scores.
>
> Nevertheless, we understand the reviewer's possible concern that evaluating against a complete ground-truth image is standard practice in related literature. To address this, we conducted a new experiment using the DyCheck dataset, which provides multi-view RGBD ground truth. We have added these results to Appendix L.
>
> As shown in Table 9, our method still achieves competitive performance with TrajectoryCrafter, further supporting the high accuracy of our approach. However, as noted in the appendix, we remain skeptical about the validity of this evaluation protocol due to the misalignment between the rendered and ground-truth images, as well as the aforementioned entangled evaluation of faithfulness and fidelity.

---

### Author Response · Authors · 2025-12-03

We are grateful to all the reviewers for your critical and insightful comments. Given that our perspective on faithfulness differs from the convention, this review process was highly constructive in refining our problem formulation into a rigorous evaluation protocol. During the rebuttal, we conducted more than 30 additional experiments to verify our hypotheses. Below, we briefly summarize our major (and extended) contributions:

**Refined Definition of Faithfulness:** Our central claim is that the model should strictly retain faithfulness “to the input rendered images.” This distinction clarifies the confusion with the conventional metric measured against full target images. We formally termed the former **Input-Faithfulness** and the latter **GT-Faithfulness**. Our experiments demonstrated a positive correlation between them, showing that prioritizing Input-Faithfulness directly benefits GT-Faithfulness, thereby shedding light on this often-overlooked aspect of generation control.

**Versatility of Homography Deformation and SA-RePaint:** While our primary implementation is built on Stable Video Diffusion, we have verified that our proposed modules are extensible to other generative architectures. Notably, we successfully applied SA-RePaint to EDM, DDIM (Appendix O), and Flow-matching models (Appendix N), demonstrating its broad applicability across different generative frameworks.

We hope these new perspectives will inspire further exploration in the field. Once again, we sincerely thank the PCs, ACs, and reviewers for your invaluable feedback.

---

### Meta-Review · Area_Chair_6kQt · 2025-12-15

**Summary:**

Firstly, the reviewers raised several central concerns regarding the paper’s main motivation. Reviewer 5Z8V pointed out that the ablation study is confusing, noting that the proposed components, Latent Homography Deformation and SA-RePaint, show limited or even negative improvements in faithfulness metrics compared to a naive baseline. Reviewer 5Z8V also expressed concerns about the unexamined prefill step and the absence of ground-truth-based visual comparisons.

Secondly, reviewers challenged the fairness and completeness of the comparison and evaluation protocol. Reviewer WhGA raised broader concerns regarding missing related-work baselines, the definition of valid regions for faithfulness evaluation, the limited effectiveness of homography estimation on co-visible regions, and the clarity and resolution of several figures. In particular, the faithfulness metrics (e.g., PSNR, SSIM, LPIPS) were computed only on selected “valid regions,” and reviewers requested a more rigorous justification for this selective evaluation. Reviewer WhGA and Reviewer Y4mM also noted the absence of comparisons with modern, high-performing NVS approaches, especially Gaussian Splatting–based methods and their variants. In addition, Reviewer 5Z8V observed that the qualitative comparison figures (e.g., Fig. 6 and Fig. 7) do not include ground-truth images, limiting the interpretability of the results.

Thirdly, reviewers questioned the generalizability and robustness of the proposed technical modules. Reviewer QBDc and Reviewer WhGA emphasized that the proposed Test-time Latent Homography Deformation relies on a global planar motion assumption, which may struggle to handle scenes with strong parallax, non-rigid motion, or complex dynamic content, thereby limiting the method’s general applicability.
Reviewer WhGA further noted the limited effectiveness of homography estimation restricted to co-visible regions. Reviewer QBDc requested additional discussion on the applicability of the method to more recent diffusion architectures (e.g., DiT-based models) and asked for explicit failure case analysis. Reviewer Y4mM also observed that some qualitative results appear overly blurred, which contradicts the claim of achieving both high faithfulness and high visual fidelity.

Finally, Reviewer WhGA pointed out that the clarity and resolution of several figures were insufficient, making accurate qualitative assessment difficult.

**Reviewer Concerns:**

**Addressed Concerns**

Reviewer 5Z8V raised concerns about the unexamined prefill step. The authors added a discussion showing that omitting the prefill step leads to noticeable gray border artifacts near image boundaries, thereby clarifying its practical role.

Reviewer QBDc and Reviewer WhGA questioned the assumptions underlying Homography Deformation. The authors clarified that homography deformation is applied only during the early denoising steps. At this coarse stage, latent features primarily capture global structure, for which planar guidance is sufficient. The deformation is subsequently disabled, allowing later denoising steps to freely generate fine geometry based on the diffusion model’s learned priors.

Reviewer Y4mM expressed concern about the lack of comparison with Gaussian Splatting methods. The authors provided a detailed explanation for why comparisons with 3D Gaussian Splatting were not included.

Reviewer Y4mM also commented on some qualitative examples appearing blurry or low-resolution. The authors explained why these specific examples were selected.

Reviewer WhGA proposed an evaluation using the MEt3R metric, which the authors subsequently conducted.

**Outstanding Concerns**

Reviewer QBDc and Reviewer WhGA questioned the generalizability of the proposed modules to DiT-based diffusion models. The rebuttal did not explicitly address how the method would transfer to or integrate with increasingly dominant DiT architectures.

Reviewer 5Z8V remained concerned about the ablation studies. The authors’ explanation did not directly address the observation that improvements over comparison methods appear to stem mainly from an enhanced baseline rather than the proposed modules.

Reviewer 5Z8V also raised concerns about the absence of ground-truth images in qualitative comparisons. The authors were asked to include ground-truth references; however, without explicit confirmation that these were added, the qualitative evaluation remains less convincing.

Reviewer QBDc requested an analysis of potential failure cases, which was not provided in the rebuttal.

Reviewer WhGA questioned the definition of the “valid region” used in the faithfulness metric. While the authors justified their choice as isolating input-faithfulness, they did not reconcile this evaluation with conventional full-image faithfulness metrics, leaving the comparability of the results unclear.

**Reviewer Scores:**

Reviewer 5Z8V: 4, no change, as the main concern on the ablation study is not solved.

Reviewer QBDc: 6, change to 4, as some concerns of this reviewer is not solved.

Reviewer Y4mM:8, no change, as the concern is solved

Reviewer WhGA: 4, no change, as this reviewer had a round of discussion with the authors but didn’t change the score.

---

### Decision · Program_Chairs · 2026-01-26

Reject